# Mapping the susceptibility of rain-triggered lahars at Vulcano island (Italy) combining field characterization, geotechnical analysis and numerical modelling

Valérie Baumann[1], Costanza Bonadonna[1], Sabatino Cuomo[2], Mariagiovanna Moscariello[2], Sebastien Biass[3], Marco Pistolesi[4], Alessandro Gattuso[5]

[1] Department of Earth Sciences, University of Geneva, Rue des Maraîchers 13, 1205 Geneva, Switzerland

[2] Laboratory of Geotechnics, University of Salerno, Via Giovanni Paolo II 132, 84081 Fisciano Salerno, Italy

[3] Earth Observatory of Singapore, Nanyang Technological University, Singapore, Singapore

[4] Dipartimento di Scienze della Terra, Università di Pisa, Pisa, Italy

[5] Istituto Nazionale di Geofisica e Vulcanologia, Sezione Palermo, Italy

**Abstract**. The characterization of triggering dynamics and of remobilised volumes is crucial to the assessment of associated lahar hazard. We propose an innovative treatment of the cascading effect between tephra fallout and lahar hazards based on probabilistic modelling that also accounts for a detailed description of source sediments. As an example, we have estimated the volumes of tephra-fallout deposit that could be remobilised by rainfall-triggered lahars in association with two eruptive scenarios that have characterized the activity of La Fossa cone (Vulcano, Italy) in the last 1000 years: a long-lasting Vulcanian cycle and a subplinian eruption. The spatial distribution and volume of deposits that could potentially trigger lahars were analysed based on a combination of tephra-fallout probabilistic modelling (with TEPHRA2), slope-stability modelling (with TRIGRS), field observations and geotechnical tests. Model input data were obtained from both geotechnical tests and field measurements (e.g. hydraulic conductivity, friction angle, cohesion, total unit weight of the soil, saturated and residual water content). TRIGRS simulations show how shallow landsliding is an effective process for eroding pyroclastic deposits on Vulcano. Nonetheless, the remobilised volumes and the deposit-thickness threshold for lahar initiation strongly depend on slope angle, rainfall intensity, and grainsize, friction angle, hydraulic conductivity and cohesion of source deposit.

**Keywords**: lahar triggering, slope stability model, shallow landslides, cascading hazards, probabilistic tephra modelling, La Fossa volcano

## 1 Introduction

Lahars, an Indonesian term to indicate volcanic debris flows and hyper-concentrated flows with various amount of volcanic solid content, can cause loss of life and damage to infrastructure and cultivated lands and represent one of the most devastating hazards for people living in volcanic areas (Pierson et al., 1990, 1992; Janda et al., 1996; Scott et al., 1996, 2005; Lavigne et al., 2000a; Witham, 2005; De Bélizal et al., 2013). The most destructive lahars were caused by break out of crater lakes or volcano dammed lakes (e.g., Mt. Kelud in Indonesia; Thouret et al., 1998) or by the interaction of hot pyroclastic density currents (PDCs) with glacial ice and snow at ice-capped volcanoes (e.g., Nevado del Ruiz in Colombia; Pierson et al., 1990). However, the most common lahars are those generated by heavy rainfalls on tephra fallout and PDC deposits emplaced on volcano slopes (e.g., Casita Volcano, Nicaragua (Scott et al., 2005); and Panabaj, Guatemala (Charbonnier et al., 2018)). For example, torrential rainstorms on loose pyroclastic deposits produced by the 1991 eruption of Pinatubo (Philippines) have generated hundreds of secondary lahars for years after the end of the eruption (e.g., Janda et al., 1996; Newhall and Punongbayan, 1996). Despite their relatively small volumes, over 6 years, these lahars have remobilised 2.5 km$^3$ of the 5.5 km$^3$ of primary pyroclastic material, inundated 400 km$^2$ of villages and fields and more than 50,000 people

were evacuated (Vallance and Iverson, 2015).

Many studies exist that make use of various analytical and numerical models to describe the potential inundation area of lahars (e.g. Procter et al. 2012; Cordoba et al. 2015; Caballero et al. 2016; Mead and Magill 2017; Charbonnier et al. 2018). The associated outcomes are fundamental to the development of risk-reduction strategies; nonetheless, all inundation models require the determination of the volume of potentially remobilised material that is often approximated due to the lack of information. In fact, the identification of lahar source areas and lahar initiation mechanisms is crucial to the evaluation of lahar recurrence and magnitude. With "lahar source areas" we refer to areas with pyroclastic material that can be remobilised to form lahars; these areas are normally located on steep slopes (> 20°), and at the head of channels draining the volcano flanks. The generation of rain-triggered lahars may be influenced by several factors such as the amount of rainfall, deposit stratigraphy, slope gradient, vegetation cover and the physical characteristics of pyroclastic deposits (e.g., thickness, permeability, pore pressure and grainsize distribution). Two main mechanisms have been identified for the triggering of lahars by rainfall on pyroclastic material: sheet and rill erosion due to Hortonian overland flow caused by deposit saturation (e.g., Collins and Dunne, 1986; Cuomo et al., 2015) and infiltration of slope surface by rainfall that can generate shallow landslides (e.g., Iverson and Lahusen, 1989; Manville et al., 2000; Crosta and Dal Negro, 2003; Zanchetta et al., 2004; Volentik et al., 2009; Cascini et al., 2010). Infiltration occurs when the rainfall intensity is lower than the hydraulic conductivity, while overland runoff occurs when rainfall intensity is greater than infiltration capacity, which is also related to capillary suction for unsaturated soils (Cuomo and Della Sala, 2013). Overland runoff is enhanced by the emplacement of very fine ash layers (< 0.125 mm) that reduces the infiltration capacity (Collins and Dunne, 1986; Leavesley et al., 1989; Pierson et al., 2013; Cuomo et al., 2016). Poor infiltration capacities of fresh pyroclastic deposits have been measured, for example, at Mt. St. Helens 1980, USA (Collins and Dunne, 1986; Leavesly et al., 1989; Major et al., 2000; Major et al., 2005), Mt. Uzen 1990-1995, Japan (Yamakoshi and Suwa, 2000; Yamamoto, 1984) and Chaitén 2008, Chile (Pierson et al., 2013). The combination of a large supply of loose pyroclastic deposits and intense rainfall episodes, therefore, increases the likelihood of lahars. This is the case on Vulcano island (Italy; Fig. 1), where the deposits of the 1888-1890 eruption, the last eruption of La Fossa volcano, are still remobilised during the rainy season (e.g., Frazzetta et al., 1984; Dellino and La Volpe, 1997; De Astis et al., 2013; Di Traglia et al., 2013). The initiation mechanism of recent lahars has been studied in detail by Ferrucci et al. (2005), but the initiation mechanism during and just after long-lasting eruptions and in association with short sustained eruptions (such as the eruptions that have characterized the activity of the volcano of the last 1000 years; Di Traglia et al., 2013) has not yet been characterized.

In order to determine the volume of potentially remobilised material, Volentik et al. (2009) and Galderisi et al. (2013) have already combined lahar triggering modelling with probabilistic assessment of tephra deposition based on a static hydrological model (Iverson, 2000) and assuming total saturation of the deposit. In addition, Tierz et al. (2017) have compiled a probabilistic lahar hazard assessment through the Bayesian belief network "Multihaz" based on a combination of probabilistic hazard assessment of both tephra fallout and PDCs and a dynamic physical model for lahar propagation. Even though these three examples were pioneering in assessing the effect of cascading hazards, the associated description of lahar triggering was overly simplified (i.e., fundamental aspects such as hydraulic conductivity and friction angle were not taken into account) and the soil characteristics as well as the intensity and duration of the rainfall were not considered. In our paper we build on these first studies to show the importance of the application of physically based models in combination with the characterization of pyroclastic material for the determination of deposit instability. Our goal is to accurately predict the volume of tephra fallout that could be remobilised by a rainfall-triggered shallow landslide in association with various eruptive conditions at La Fossa cone in order to compile a rain-triggered lahar susceptibility map. To achieve this task, we combine the shallow landslide model TRIGRS (Baum et al., 2002) with both probabilistic modelling of tephra fallout (for eruptions of different duration and magnitude) and field and geotechnical characterisations of tephra-fallout deposits (i.e., grainsize, hydraulic conductivity, soil suction, deposit density).

First, we describe the physical characteristics (e.g., grainsize, hydraulic conductivity, angle of friction) of selected tephra-fallout deposits associated with both a long-lasting Vulcanian eruption (i.e., the 1888-90 eruption) and a subplinian eruption (Pal D eruption of the Palizzi sequence; Di Traglia et al., 2013). Second, we characterize the lahar deposits associated with the 1888-90 eruption, which provide insights into lahar source areas, flow emplacement mechanisms and inundation areas of future lahars. The physical characteristics of tephra-fallout deposits are used in combination with a probabilistic modelling of tephra fallout (Biass et al., 2016) to estimate the unstable areas based on the shallow landslide model TRIGRS (Baum et al., 2002). This approach provides the first integrated attempt to quantify the source volume of lahars as a function of probabilistic hazard assessment for tephra fallout (with TEPHRA2), numerical modelling of lahar triggering (with TRIGRS), field observations (including primary tephra-fallout deposits, geology, geomorphology and precipitations), and geotechnical tests of source deposits. Finally, we propose a new strategy to map the syn-eruptive lahar susceptibility as a critical tephra-fallout deposit thickness resulting in unstable conditions, which could represent a valuable tool for contingency plans. Here the term syn-eruptive is used in the sense of Sulpizio et al. (2006) to indicate lahars originated during volcanic eruptions or shortly after, while post-eruptive lahar events are generated long (i.e. few to several years) after an eruption. This is also in agreement with the current definition of primary (syn-eruptive) or secondary (post-eruptive or unrelated to eruptions) provided by Vallance and Iverson (2015) and Gudmunsson (2015). To sum up, this study explores new strategies for volcanic multi-hazard assessment and offers an innovative treatment of the cascading effects between tephra fallout and lahar susceptibility.

## 2 Study area

### 2.1 Eruptive history

The island of Vulcano, the southernmost island of the Aeolian archipelago, consists of several volcanic edifices whose formation overlapped in time and space beginning 120 ka ago. The most recently active volcano is the La Fossa cone, a 391 m-high active composite cone that began to erupt 5.5 ka ago (Frazzetta et al., 1984) and whose erupted products vary in composition from latitic to rhyolitic, with minor shoshonites (Keller, 1980; De Astis et al., 1997; Gioncada et al., 2003). The stratigraphy of La Fossa cone has been described in detail in several studies (Keller, 1980; Frazzetta et al., 1983; Di Traglia et al., 2013; De Astis et al., 2013). The last period of eruptive activity (younger than 1 ka) has been recently divided into two main eruptive clusters, further separated into eruptive units (Di Traglia et al., 2013; **Fig. 1**). The activity of Palizzi and Commenda eruptive units (PEU and CEU, respectively) is grouped into a single eruptive period (Palizzi-Commenda Eruptive Cluster, PCEC) lasting approximately 200 years (11th to 13th century). The following Pietre Cotte cycle, the post-1739 AD and the 1888–1890 AD activity form the Gran Cratere eruptive cluster (GCEC, 1444-1890 AD; Di Traglia et al., 2013). The stratigraphic sequence of PEU displays a large variety of eruptive products and a wide spectrum of magma compositions, including cross-stratified and parallel-bedded ash layers (e.g., Pal A and Pal C in Di Traglia et al., 2013), pumiceous fallout layers of rhyolitic (Pal B) and trachytic composition (Pal D), several lava flows intercalated in the sequence (e.g., the rhyolitic obsidian of Commenda and the trachytic lava flows of Palizzi, Campo Sportivo and Punte Nere), and, finally, several ash layers and widely dispersed PDC deposits (CEU) associated with the hydrothermal eruption of Breccia di Commenda Eruptive Unit (Gurioli et al., 2012; Rosi et al., 2018) which closes the PCEC. The GCEC includes the recent products of the Pietre Cotte eruptive unit (ash and lapilli layers from Vulcanian activity), rhyolitic pumiceous fallout layers and the rhyolitic AD 1739 Pietre Cotte lava flow. The uppermost part of the GCEC is represented by the products of the AD 1888-1890 eruption, consisting of latitic spatters, trachytic and rhyolitic ash and lapilli layers and the characteristic breadcrust bombs. Historical chronicles (Mercalli and Silvestri, 1891; De Fiore, 1922), archeomagnetic data (Arrighi et al., 2006; Zanella et al., 1999; Lanza and Zanella, 2003) and stratigraphic investigations (Di Traglia et al. 2013; De Astis et al. 1997, 2013) concur in indicating that in the past 1000 years at least 20 effusive and explosive eruptions have occurred. Among the explosive eruptions, the Vulcanian cycles represent the most important events in terms of recurrence (at least five

long-lasting episodes corresponding to annual frequencies of $5 \times 10^{-3}$ /year) and erupted volumes, followed by rarer short-lived, higher intensity events, strombolian explosions and phreatic eruptions (Table 1). In this work, we will focus on the tephra fallout associated with two main eruptive styles of the past 1000 years: a long-lasting Vulcanian eruption such as that of 1888-90 (e.g., Di Traglia et al., 2013, Biass et al., 2016) and a subplinian eruption, which emplaced deposits such as the Palizzi D sub-unit. For simplicity, we refer to the deposits of the last 1888-1890 long-lasting Vulcanian event as the "1888-90 eruption" and to the fallout deposit of the subplinian Palizzi D eruption as "Pal D" primary deposits.

## 2.2 Climate

Vulcano island has a typical semi-arid Mediterranean climate (De Martonne, 1926) with annual rainfall between 326 mm and 505 mm, falling mostly during autumn and winter seasons **(Fig. 2a)**. Based on Arnone et al. (2013), rainfall trends in Sicily island can be classified in three intensity-based categories: light precipitation (0.1-4 mm d$^{-1}$), moderate precipitation (4-20 mm d$^{-1}$) and heavy–torrential precipitation (>20 mm d$^{-1}$). Heavy–torrential precipitation occurred three times in 2010 and 2011, two times in 2012 and one time in 2013 and 2014 (INGV Palermo); the associated rainfall duration can last 2-3 hours to 3 days and occur normally in September, October, November and December, and, more rarely, in May **(Fig. 2b)**. Such observation agrees with the observations of De Fiore (1922) for the period just following the 1888-90 eruption, indicating that the weather pattern in the region has been pretty constant. These meteorological conditions associated with poor vegetation coverage (Valentine et al., 1998), steep slopes **(Fig. 1d)** and the presence of layered, fine-grained tephra (lapilli and ash), favour the remobilisation of volcanic deposits (Ferrucci et al., 2005; Di Traglia, 2011). Wind patterns inferred from the ECMWF ERA-Interim dataset (Dee et al., 2011) for the period 1980-2010 show a preferential dispersal towards SE at altitudes lower than 10 km above sea level (a.s.l.), which above shift towards E (Biass et al., 2016).

## 2.3 Recent lahars

Both syn-eruptive and post-eruptive lahar events induced the progressive erosion of the tephra deposits which covers La Fossa cone. The tephra-fallout deposit associated with the most recent Vulcanian eruption (1888-90) has been almost completely removed from the upper slopes and accumulated at the foot of the cone, where the stratigraphic sections show a succession of lahar deposits with thicknesses between 0.1 and 1 m (Ferrucci et al., 2005). All the tephra sequence of the Gran Cratere eruptive cluster (including the 1888-90 Vulcanian eruption) lies on top of thinly stratified reddish, impermeable ash layers (Varicolored Tuffs or "Tufi Varicolori"; Frazzetta et al., 1983; Capaccioni and Coniglio, 1995; Dellino et al., 2011) which concluded the Breccia di Commenda phase. In most parts of the cone, the dark-grey tephra of the Vulcanian cycles is almost completely eroded and the "Tufi Varicolori" tuffs are exposed. A rill network developed on the impermeable fine-grained tuffs that conveys water to a funnel shape area, where a main gully defines a drainage basin (Ferrucci et al., 2005). The main gullies start where the loose grey interdigitated tephra-fallout and lahar deposits crop out **(Fig. 1)**. Lahar volumes and travel distance strongly depend on both the availability of pyroclastic material in the source area and on the characteristics of the rainfall events (intensity and duration). Ferrucci et al. (2005) estimated volumes between 20 and 50 m$^3$ for 3 recent lahars based on levees and terminal lobe deposits geometry on the NW sector of La Fossa cone. We describe, as an additional example, a small lahar that occurred on September 2017 (just before our main field survey) on the NW cone flank covering part of the La Fossa crater trail (supplementary material). However, the occurrence of larger syn-eruptive lahars ($10^3$-$10^4$ m$^3$) reaching the Porto di Levante and Porto di Ponente areas have been also reported (Di Traglia, 2011). Much of the material of the La Fossa ring plain has been transported by lahars during the post-1000 AD period; in fact, the Porto di Levante and Porto di Ponente plains were progressively filled up with 2-3 m of reworked tephra-fallout deposits during this time interval (Di Traglia et al., 2013). In 1921 the loose grey 1888-1890 tephra-fallout deposit was already largely eroded and the "Tufi Varicolori" tuffs were already exposed (De Fiore, 1922). Di Traglia et al. (2013) also reported post-eruptive lahar deposits on the flank of the volcano with increasing thickness towards the base. Deposit

thickness then decreases on the ring plain in Porto area where they also show finer grainsize and lamination. The study of recent lahars located on the NW flank of La Fossa cone indicated that the deposits were emplaced by saturated slurries in which grain interaction dominated the flow dynamics (Ferrucci et al., 2005). The use of empirical ratio clay/(sand+silt+clay) (Vallance and Scott, 1997) and the physically based calculation of Nmass (Iverson and Vallance, 2001) suggest non-cohesive debris flows and transport dominated by granular flows (Ferrucci et al., 2005).

## 3 Methods

### 3.1 Field sampling

The field characterization of both tephra-fallout primary deposits in the lahar initiation zones and lahar deposits was carried out during two field campaigns in 2017 and 2018. Eight undisturbed samples of tephra-fallout primary deposit were collected for geotechnical tests from four outcrops on the La Fossa cone and Palizzi valley for the 1888-90 and the Pal D units (yellow circles in Fig. 1b and 1d). In particular, Pal D tephra-fallout primary deposit was sampled at locations V3 (1 sample) and V4 (1 sample) (Fig. 1b), while the 1888-90 tephra-fallout primary deposit was sampled at locations V1 (4 samples) and V2 (2 samples) (Fig. 1d). In addition, five samples for grainsize analysis were collected vertically every 6 cm at two 1888-90 tephra-fallout primary deposit outcrops (yellow circles V1 and V2 in Fig. 1d). We consider the tephra-fallout primary deposit of V1 and V2 as representative of the lahar initiation zone on the S and NW flanks of La Fossa cone, respectively. In contrast, the lahar source area associated with the Pal D primary deposit is either covered by the new eruptive products or eroded. As a result, Pal D had to be sampled at the base of the cone (V3 and V4 in Fig. 1b). Deposit sampling for geotechnical tests was performed by inserting a steel tube with a height of 30 cm and a diameter of 10 cm into the ground (see Appendix A, Fig. A1). A basal support was then inserted, and the tube extracted from the deposit with a minimum disturbance of the internal stratigraphy. The tube was then covered on both ends to preserve the deposit for further laboratory analysis (see Section 3.2). Due to the sampling apparatus, most geotechnical tests could only be carried out on the top 30 cm of each deposit location. As a comparison, in V1 the 30 cm tube was inserted after having eliminated the top 30 cm of deposit to analyse the central part of the outcrop. Given the characteristics of the deposit, we consider 30 cm of sampling as representative for the main characteristics of both the 1888-90 (which is thinly laminated across the entire section) and the Pal D (which is mostly massive) deposits. Soil suction measurements were carried out in situ on the 1888-90 tephra-fallout deposit with a soil moisture probe ("Quick Draw" Model 2900FI) (Fig. 1 and Appendix A). The saturated hydraulic conductivity was estimated in the field with a single-ring permeameter on Pal D primary deposit (V3 and I3 in Fig. 1) and on 1888-90 primary deposits (I1 and I2, near V2 location, in Fig. 1) (see Appendix A, Fig. A3). The field description and sampling of syn-eruptive lahar deposits associated with the 1888-90 eruption were performed on the NW volcano flanks, in the Palizzi valley and in the Porto plain (red squares in Fig. 1; V5 to V12). Eleven samples of lahar matrices were sampled on the S cone flank (V5), on the NW cone flank (V8-V12), in the Palizzi valley (V6) and in the Porto Plain (4 samples in V7).

### 3.2 Laboratory analyses

Grainsize analyses were carried out at the University of Geneva for three tephra-fallout sections (11 samples) and for 11 lahar-deposit matrix samples (fractions between -6 and 10 φ). The phi (φ) scale is a sediment particle size scale diameter calculated as the negative logarithm to the base 2 of the particle diameter (in millimetres) (Krumbein, 1938). Samples were mechanically dry sieved at half-φ intervals for the coarser fraction between 16 mm and 0.25 mm. The laser granulometry technique (CILAS 1180 instrument) was used for fractions smaller than 0.25 mm. Deposit density of five samples of the 1888-90 primary deposits and for one samples of the Pal D tephra-fallout deposit were also determined at the University of Geneva weighing a given volume of sample material measured with a graduated cylinder.

Natural water content and shear strength were measured on undisturbed samples at the University of Salerno. The natural water content ($w_n$) was evaluated at several depths (from 0.06 m to 0.3 m) for six samples of the 1888-90 primary deposits and for four samples of the Pal D primary deposit. The shear strength of primary deposits was measured through direct shear tests performed in conventional direct shear apparatus in the Laboratory of Geotechnics at University of Salerno. Tests on undistured specimens of 1888-90 primary deposit were performed at both natural water content and in fully saturated condition. The natural water content ($w_n$) and the degree of saturation were evaluated before and after the tests. The modified Kovacs model of Aubertin et al. (2003) was used to obtain the Soil Water Retention Curve (SWRC) from the grainsize data of the source area (i.e., D10 and D60, the diameter corresponding to 10% and 60% of the grainsize distribution; see supplementary material, Table S2), and the liquid limit. SWRC relates the water content to soil suction. The tests were interpreted in terms of shear stress and the vertical effective stress as defined by Bishop (1959), referring to the "effective saturation degree" ($S_{re}$) following Eq. (1):

$$\sigma'_{ij} = \sigma_{ij} - u_a\delta_{ij} + S_r(u_a - u_w)\delta_{ij}, \tag{1}$$

where $\sigma_{ij}$ (kPa) is the total stress tensor, $u_a$ (kPa) is the pore air pressure, $u_w$ (kPa) is the pore water pressure, $u_a$ - $u_w$ is the matric suction and $S_r$ (%) is the degree of saturation.

The saturated hydraulic conductivity $K_s$, (m s$^{-1}$) for the 1888-90 primary deposit was measured through laboratory tests at the University of Salerno. The test was carried out on a reconstructed specimen with height of 140 mm and diameter of 39.4 mm obtained through water pluviation technique, ensuring specimen saturation. A constant water volume (5 ml) was forced to go through the specimens by applying a difference of pore pressure between top and bottom, while the time was measured. The test was repeated 5 times and each time $K_s$ was evaluated.

The saturated soil diffusivity $D_0$ (m$^2$ s$^{-1}$) was evaluated for both deposits using the soil water retention relationship of Rossi et al. (2013) as a function of the saturated hydraulic conductivity and the parameters ($h_b$, $\lambda$) of Brooks and Corey (1962, 1964) following Eq. (2):

$$D_0 = \frac{h_b k_{sat}}{\lambda(100 \cdot n - \theta_s)}, \tag{2}$$

where $\theta_s$ (%) is the soil water content at the saturation, $h_b$ (kPa) the bubbling pressure and $\lambda$ (m$^2$ g$^{-1}$) the pore size index distribution. The parameters $h_b$ and $\lambda$ were estimated interpolating the data of SWRC.

### 3.3 Probabilistic tephra-fallout modelling

In order to best describe the cascading effect between tephra deposition and lahar triggering susceptibility in a context of multi-hazard assessments, tephra-fallout deposits considered in our analysis are those probabilistically modelled by Biass et al. (2016). Based on the stratigraphy of the last 1000 years of La Fossa (Di Traglia et al., 2013), Biass et al. (2016) defined three eruption scenarios for tephra fallout including: i) a long-lasting Vulcanian eruption scenario (plume heights: 1-10 km a.s.l.; total mass: 1.9-140 $\times$ 10$^9$ kg; duration: 30 days-3 years); ii) a VEI 2 (Volcanic Explosivity Index, Newhall and Self, 1982) subplinian eruption scenario (plume heights: 5-12 km a.s.l.; mass: 0.6-6 $\times$10$^9$ kg; duration: 0.5-6 h); and iii) a VEI 3 subplinian eruption scenario (plume heights: 8-17 km a.s.l.; mass: 6-60$\times$ 10$^9$ kg; duration: 0.5-6 h). Note that although no VEI 3 eruption is observed in the stratigraphy of the last 1000 years of activity, evidences of VEI 3 eruptions are found in the older history of La Fossa. All three scenarios were simulated probabilistically using the TEPHRA2 model (Bonadonna et al., 2005) through the TephraProb software (Biass et al., 2016b). Probabilistic isomass maps were computed for various probability thresholds, which express the distribution of tephra load for a fixed probability of occurrence within a given eruption scenario (Biass et al., 2016b). For the long-lasting Vulcanian scenario, various probabilistic isomass maps were computed to express the cumulative tephra fallout at a given time after eruption onset. Note that these cumulative maps ignore remobilisation of the primary deposit between single Vulcanian explosions. These probabilistic isomass maps were

converted in probabilistic isopach maps using deposit densities of 1200 kg m$^{-3}$ and 600 kg m$^{-3}$ for the Vulcanian and subplinian scenarios, respectively (Biass et al., 2016).

## 3.4 TRIGRS model

Based on the physical characteristics of the tephra-fallout deposits (high permeability) and on the high intensity of rainfall events, we assume that the most probable rain-triggered lahar initiation mechanism on La Fossa cone is shallow landsliding. Shallow landsliding is produced by an increase of water pore pressure due to rainfall infiltration on tephra deposits which causes a slope failure. Several slope stability models have been used to predict lahar initiation processes as shallow landslides in volcanic areas (e.g. Cascini et al., 2010; Frattini et al., 2004; Crosta and Dal Negro, 2003; Sorbino et al. 2007, 2010; Cascini et al., 2011; Cuomo and Iervolino, 2016; Cuomo and Della Sala, 2016; Mead et al., 2016, Baumann et al., 2018). Among those, the Fortran Program TRIGRS (Baum et al., 2002) can be used for computing transient pore pressure and the related changes in the factor of safety due to rainfall infiltration. Here, TRIGRS is used to investigate the timing and location of shallow landslides in response to rainfall in large areas (e.g., Baumann et al., 2018). Baum et al. (2002) extended the method of Iverson (2000) by implementing the solutions for complex time sequence of rainfall intensity, an impermeable basal boundary at infinite depth and optional unsaturated zone above the water table. TRIGRS model is applicable for unsaturated initial conditions, with a two-layer system consisting of a saturated zone with a capillary fringe above the water table overlain by an unsaturated zone that extends to the ground surface. The unsaturated zone acts like a filter that smooths and delays the surface infiltration signal at depth. The model uses the soil-water characteristic curve for wetting of the unsaturated soil proposed by Gardner (1958) and approximates the infiltration process as a one-dimensional vertical flow (Srivastava and Yeh, 1991, Savage et al., 2004). The reader is referred to the vast literature published on the application of this model for more details (e.g., Baum et al. 2002, 2008, Savage et al., 2003; Salciarini et al., 2006, Cuomo and Iervolino, 2016). Briefly, the infiltration models in TRIGRS for wet initial conditions are based on Iverson's (2000) linearized solution of the Richards equation and its extension to that solution (Baum et al., 2002; Savage et al., 2003, 2004). The solution is valid only where the transient infiltration is vertically downward and the transient lateral flow is relatively small.

Following Iverson (2000), slope stability is calculated using an infinite-slope stability analysis. Incipient failure of infinite slopes is described by an equation that balances the downslope component of gravitational driving stress against the resisting stress due to basal Coulomb soil friction and the influence of groundwater (Iverson, 2000). The Factor of Safety (FS) is calculated at a depth Z by Eq. (3):

$$FS\ (Z,t) = \frac{tan\phi'}{tan\beta} + \frac{c' - \psi\ (Z,t)\gamma_w tan\phi'}{\gamma_s Zsin\beta cos\beta} \tag{3}$$

where $c'$ (kPa) is the effective soil cohesion, $\phi'$ (deg) is the effective friction angle, $\psi(Z,t)$ is the ground water pressure head $\psi$ (kPa) as a function of depth $Z$ (m) and time $t$ (s), $\beta$ (deg) is the slope angle, $\gamma_w$ (kN m$^{-3}$) is the unit weight of groundwater and $\gamma_s$ (kN m$^{-3}$) is the unit weight of soil. The pressure head $\psi$ (Z, t) in (3) is obtained from various formula depending on the particular condition modelled. FS is calculated for pressure heads at multiple depths (Z). The slope is predicted to be instable where FS< 1, in a state of limiting equilibrium where FS=1 and stable where FS>1. Thus, the depth $Z$ of landslide initiation is where FS first drops below 1.

TRIGRS model for unsaturated initial conditions was applied on the probabilistic isopach maps described in section 3.3 for the long-lasting Vulcanian, the subplinian VEI 2 and the subplinian VEI 3 eruption scenarios. For each eruption scenario, probabilistic isopach maps are computed for probabilities of occurrence of 25% and 75% (Biass et al., 2016). The eruption associated with the subplinian scenario is considered to be short lived (<6 hours), and therefore, one single deposit is analysed. Instead, for the long-lasting Vulcanian scenario, deposits are computed for durations of 3, 6, 9, 12, 18 and 24 months. A 5-m resolution Digital Elevation Model (DEM) of Vulcano Island was used in our susceptibility analysis that was

computed from contour lines and spot heights reconstructed from stereo-photograms at a scale of 1:35000 collected during an aerophotogrametric flight in 1994-5 (Bisson et al., 2003). The maximum planimetric error of the contour lines reconstructed from stereo models is less than 3.5 m. The vertical error of the DEM is lower than 0.5 m in the area of La Fossa cone and <1 m in the flat area of Vulcano Porto (Bisson et al., 2003).

## 4 Results

### 4.1 Field characterization of tephra-fallout and lahar deposits

*Pal D tephra-fallout deposit*

We logged two sections of the Pal D tephra-fallout deposit at outcrops located in the Palizzi Valley (point V3 and V4, Fig. 1b). The isopach map of Di Traglia (2011) shows the associated southward dispersal (Fig. 1b). The Pal D section at V3 is a 1 m-thick, massive, grain supported and well-sorted pumice deposit between two sub-units of the Palizzi cycle (Fig. 3a): Pal C deposit at the base (alternation of black lapilli and ash) and the rhyolitic white ash of Rocche Rosse eruption from Lipari and the Breccia di Commenda at the top (Di Traglia et al., 2013; Rosi et al., 2018). The mean saturated hydraulic conductivity of the Pal D deposit measured in the field at V3 (Fig. 1b) is $6.8 \times 10^{-4}$ m s$^{-1}$.

*1888-90 tephra-fallout deposit*

Two stratigraphic sections of the 1888-90 tephra-fallout deposit were logged in the upper part of the La Fossa cone S flank (V1 in Fig. 1d; Fig. 3c) and at the base of the NW flank (V2 in Fig. 1d; Fig. 3d). The isopach map for 1888-90 primary tephra-fallout deposit shows and almost circular dispersal (Fig. 1d). The V1 section overlies several older tephra-fallout and lahar units. It is a 1 m-thick deposit consisting of an alternation of thin ash and lapilli layers overtopped by 0.2 m of reworked tephra. The whole sequence shows an inclination of 30°. The second section (V2) is a 0.5 m-thick deposit laying on the Commenda tephra sequence and is overlaid by 0.3 m of reworked tephra. Seven soil suction measurements on the 1888-90 deposits located on the La Fossa cone in the upper catchment and lower part of the cone are comprised between 15 kPa and 27 kPa (Fig. 1d and supplementary material, Table S1). The mean saturated hydraulic conductivity measured in the field varies between 6.0 and $7.5 \times 10^{-5}$ m s$^{-1}$ (Fig 1d, I1 and I2).

*1888-90 lahar deposits*

Stratigraphic sections in gullies and small channels on the NW flank of the La Fossa cone show several massive to laminated, remobilised deposits covering the 1888-90 primary tephra-fallout deposit. Unfortunately, no map exists that describes the distribution of lahars on Vulcano mostly due to the difficulty to correlate the exposed deposits across the different gullies. The thickness of each lahar layer, representing different flow pulses, varies between ~ 0.20 and 1 m. The lahar deposits are massive to thinly laminated and matrix supported, with boulders immersed in a coarse-ash and lapilli matrix. The observed boulders have diameter between 5 and 15 cm. Although the distinction among debris flow and hyper-concentrated flow emplacement mechanisms from the direct observation in the field was not possible due to the lack of information related to original water content, we speculate that most of the well-sorted, massive deposits were related to hyper-concentrated flows based on the definition of Pierson (2005).

### 4.2 Laboratory characterization of tephra-fallout and lahar deposits

*Grain-size analyses*

The Mdφ of the majority of tephra-fallout and lahar samples is in the range of 2φ and -1φ and most deposits are well sorted (σφ mostly between 1-2) (Fig. 5; associated grainsize distributions are shown in supplementary material, Fig. S3). An exception is represented by the grainsize distribution of the top 30 cm of the Pal D primary tephra-fallout deposit at section

V3 that shows an Mdφ and σφ of -3.42 and 1.55, respectively (Fig. 5). Grainsize distributions of the top 30 cm of the 1888-90 tephra-fallout deposit on the S flank of La Fossa cone (V1) show Mdφ between -0.88–0.08 and σφ between 1.46–1.80, respectively (Fig. 5). At the base of the cone on the NW sector (V2), grainsize distributions of the 1888-90 tephra-fallout deposit are slightly finer (Mdφ of 0.09 to 0.9) and with a poorer sorting (σφ of 1.49 to 2.1). The grainsize distributions of the top 30 cm of the 1888-90 tephra-fallout deposit at V1 and V2 show a predominance of coarse ash (Fig. 3). All lahar matrix samples have a low content of fine ash (i.e., 2.4–16%) and contain 60–83% of coarse ash and 2–36% of lapilli. The Mdφ vs σφ diagram shows a finer grainsize distribution for lahar matrices located in the Palizzi valley and Porto di Ponente harbour (V6 and V7 samples, respectively; Fig. 5) than for those located on the La Fossa cone (V5 sample; Fig. 5). In general, the grainsize distributions of the 1888-90 primary tephra-fallout deposits is similar to their remobilised counterparts (Fig. 5 and supplementary material, Fig. S3). Only three primary tephra-fallout deposits are coarser than lahar matrices (V1C, V1D and V3; Fig. 5)

*Natural water content*

The natural water content ($w_n$) of the 1888-90 tephra-fallout deposit varies from 2.64% to 3.65%, while Pal D tephra-fallout deposit exhibit higher $w_n$ (10.80–30.63%). The specific gravity ($G_s$) for the solid fraction was measured for both deposits and shows values of 2.57 and 2.42 for the 1888-90 and Pal D tephra-fallout deposits, respectively.

*Shear strength*

Although the four specimens of 1888-90 primary tephra-fallout deposit are consolidated at three different total stresses, all specimens exhibit a slight hardening associated to a dilative behaviour. The shear stress envelope exhibits high friction angle (φ = 42°) and nil cohesion (Table 2). For Pal D tephra-fallout deposits, direct shear tests are performed on reconstructed specimens constituted only by the size fraction smaller than 20 mm (i.e. lapilli and ash). In order to maintain the in-situ characteristics, the specimen is reconstructed using air pluviation method into the shear box, i.e. pouring the dry deposit material with a spoon from nil drop height. The specimens exhibit high porosity (n equals about 0.72). The tests are performed in dry condition and all specimens exhibit a dilative behaviour. The friction angle at peak is high, and typical of lapilli clasts (φ = 54°), while the cohesion is null (Table 2). The angle of dilatation (Ψ) is also evaluated and is about 13°. Thus, according to Taylor (1948) the friction angle is about 41°, but it will be reached at large deformation.

*Hydraulic conductivity*

The mean saturated hydraulic conductivity of the 1888-90 tephra-fallout deposit measured in the laboratory ($8.50 \times 10^{-5}$ m s$^{-1}$) is similar to that obtained during field measurements (6.0 and $7.5 \times 10^{-5}$ m s$^{-1}$). The mean hydraulic saturated conductivity of the Pal D tephra-fallout deposit could not be measured in the laboratory because the deposit grainsize (Mdφ = -3.4φ) is too coarse for the apparatus. As a result, we use two values for the modelling: one from the literature derived for coarse-grained volcanic soils and one obtained from field measurements. A value of $1 \times 10^{-2}$ m s$^{-1}$ (Table 2) is inferred from the hydraulic conductivity measured in the field on 2011 Córdon Caulle (Chile) eruption lapilli deposits (Baumann et al., 2018) and from the lower pumice deposit from Vesuvius (Crosta and Dal Negro, 2003). These values are significantly higher with respect to our field measurements of $6.8 \times 10^{-4}$ m s$^{-1}$. These two end members are hereafter referred to as *high* (i.e. $1 \times 10^{-2}$ m s$^{-1}$) and *low* (i.e. $6.8 \times 10^{-4}$ m s$^{-1}$) hydraulic conductivities and will be used to explore a variety of deposit conditions.

*Soil Water Retention Curve*

The SWRC of the 1888-90 primary tephra-fallout deposit exhibits air entry value equal to 1 kPa, while the water content at saturation ($\theta_s$) is 0.47 and the residual water content ($\theta_r$) is 0.04. The SWRC of Pal D deposit exhibits air entry value lower than 1 kPa and slightly high-water content at saturation ($\theta_s = 0.72$) and low residual water content ($\theta_r = 0.03$) (Table 2). The

data are interpolated using the equations from both Gardner (1958) and Brooks and Corey (1962, 1964).

*Saturated soil diffusivity*

The saturated soil diffusivity of the 1888-90 and Pal D tephra-fallout deposits are one order of magnitude higher than their saturated soil conductivity (Table 2). The 1888-90 tephra-fallout deposit shows a value of $3.28 \times 10^{-4}$ m$^2$ s$^{-1}$, while the Pal D primary tephra-fallout deposit a value of $6.59 \times 10^{-3}$ m$^2$ s$^{-1}$.

**4.3 Modelling**

Based on the local weather pattern (Section 2.2), TRIGRS simulations were run using one high intensity rainfall scenario of 6.4 mm h$^{-1}$ over 5 hours (i.e., total of 32 mm). As mentioned in section 2.2, such heavy-torrential precipitations have occurred twice in 2011 causing widespread floods in the Porto flood-plain and can be considered amongst the most intense scenarios to occur on Vulcano based on available data (Fig. 2). Following Arone et al. (2013), we consider this scenario as a heavy-torrential scenario and we used it to investigate the maximum unstable tephra-fallout volume. For the Pal D tephra-fallout deposit, we used two different hydraulic conductivities in order to consider both end-members as described in laboratory analyses (Table 2).

TRIGRS simulations assume that: i) a water table is located at the bottom of the tephra-fallout sequence (lower boundary); and ii) the tephra-fallout sequence lies on an impermeable layer. These assumptions are supported by the exposure of the consolidated and impermeable Tufi Varicolori unit on the upper part of the La Fossa cone (Frazzetta et al., 1983; Capaccioni and Coniglio, 1995; Dellino et al., 2011). Although lahars have been initiated all around the La Fossa cone in the past, we analyse here the slope and the stability condition of the tephra deposits in two selected potential lahar source areas (Fig. 6, black lines). The first NW source area represents a direct threat to the populated Porto village, while the second S source area is downwind of the prevailing wind and presents the highest probabilities of tephra accumulation (Section 1.2; Biass et al., 2016). The percentage of slope angle ranges (Fig. 6) for the NW and S lahars source area respectively are 18% and 22% for a slope angle between 6° and 30°; 38% and 46% for slope angle between 30° and 35°; 31% and 24% for slope angles between 35° and 41°; and 12% and 7 % for slope angles bigger than 41°. Comparing the slope angle distribution for both areas, we observe that the percentage of steep slopes is higher in the NW area. Finally, we selected two upper catchments with similar surface area and reshaped the S upper catchment to have the same size of 4,665 m$^2$ in order to facilitate the comparison of remobilised volumes associated with different eruptive conditions (Fig. 6). The catchment boundaries were defined with a semiautomatic tool in ArcGIS, using the flow direction raster and defining a pour point for the catchment, then we obtained the contributing area above the pour point, which was defined as the lahars source area.

**4.3.1 Deposition and remobilisation scenarios**

*Vulcanian eruption scenarios*

Figure 7 shows the probabilistic isopach maps (for a probability of occurrence of 25%) and the instability maps for eruption durations of 6, 12, 18 and 24 months. For an eruption duration of 6 months, only 4.8% percent of the NW and 6% of the S lahar source areas, respectively, are unstable due to the small tephra-fallout deposit thickness (between 6 and 12 cm) (Fig. 7A). For an eruption duration of 12 months, the unstable areas are significantly higher: 69% for the NW and 81% for the S lahar source areas, respectively (tephra-fallout deposit thickness between 14 and 22 cm; Fig. 7B and Table 4). For an eruption duration of 18 months the percentage of unstable areas for the NW is also very high (89%) and reached a value of 66% for the S area (Fig. 7C). The percentage of unstable area decreases for an eruption duration of 24 months, with 52% for the NW and 22% for the S, where the tephra-fallout deposit accumulation is more than 35 cm in the case of S source area (Fig. 7D, Table 4). Figure 8 shows the unstable volumes as a function of eruption durations described above calculated for the 2 single upper catchments with the same area (4,665 m$^2$) located in the NW source area (NW catchment) and in the S

source area (S catchment; Fig. 6). The largest unstable volume is reached for an eruption duration of 18 months, with a volume of 1,105 m$^3$ for the S upper catchment and 990 m$^3$ for the NW upper catchment in the case of 25% probability of occurrence scenario (Table 5).

*Subplinian eruption scenarios*

The same two lahar source areas were used for investigating the instability of the subplinian deposits. Four probabilistic isopach maps were considered (VEI 2 and VEI 3 with 25% and 75% probability of occurrence) and combined with hydraulic conductivities both measured in the field and derived from literature (i.e. $K_s = 1 \times 10^{-2}$ m s$^{-1}$ and $K_s = 6.8 \times 10^{-4}$ m s$^{-1}$). Using the highest hydraulic conductivity, a VEI 2 eruption with a 25% probability of occurrence results in 79% and 14% of

10 unstable areas in the NW and S flank, respectively (Table 4). A 75% probability of occurrence increases unstable areas to 99% and 97%. On the contrary, considering a VEI 3 eruption with 25% probability of occurrence, only the tephra-fallout deposit located on slope > 48° is unstable (2% and 0.02%). A 75% probability shows a 99% of unstable area in the NW and 58% in the S area. Using the lowest hydraulic conductivity, almost all the deposit resulting from a VEI 2 is unstable (99% of the NW source area and 97% of the S source area) regardless of the probability of occurrence (Table 4). In the case of VEI 3

scenario with a 25% probability of occurrence, a high percentage of the NW source area is unstable (97%) whereas only 25% of the S source area is unstable (Table 4). The unstable tephra-fallout volumes calculated for the subplinian scenarios for the two single upper catchments NW and S show that the largest volume (2,455 m$^3$) resulted for the NW upper catchment and the subplinian VEI 3 (25%) scenario with a $K_s = 6.8 \times 10^{-4}$ (i.e. hydraulic conductivity measured in the field) (Table 5, Fig. 9).

**4.3.2 Parametrization of unstable area based on variable tephra-fallout thickness**

In order to characterize the minimum tephra-fallout deposit thickness necessary to trigger lahars on Vulcano during or just after either a Vulcanian cycle or a subplinian eruption, we carried out TRIGRS simulations using the characteristics of the 1888-90 and of the Pal D tephra-fallout deposits and increasing deposit thicknesses from 0.1 to 1.1 m, with an interval of 0.05 m. In these simulations, the deposit thickness was considered constant over the whole NW and SE source areas. The

25 same rainfall scenario was applied. The percentage of unstable areas for the NW and S source areas (Fig. 10 shows that the tephra-fallout deposit thickness generating the largest instability for the 1888-90 eruption tephra-fallout deposit is between 20 and 30 cm. For Pal D tephra-fallout deposits using the lowest hydraulic conductivity, the unstable percentage area decreases rapidly with an increase in deposit thickness, with virtually the entire deposit being stable beyond a 45 cm thickness. In contrast, when the Pal D tephra-fallout deposit is simulated with high hydraulic conductivity, almost all of the

30 lahar source area is unstable (98%) for deposit thickness between 10 and 65 cm, after which the fraction of unstable area decreases with increasing deposit thickness.

In order to investigate the combination of multiple parameters (FS, deposit thickness, pore pressure, slope, rainfall intensity), we have carried out dedicated simulations for a smaller area (100 pixel only on the NW source area) (Fig. 11 for the 1888-90 eruption and Fig. 12 for Pal D). Three slope angles (38°, 35.4° and 30.1°) and two rainfall intensities have been considered

(6.4 mm h$^{-1}$ with a duration of 5 hours and 15.5 mm h$^{-1}$ with a duration of 3 hours). In particular, the rainfall intensity of 15.5 mm h$^{-1}$ represents the worst rainfall scenario for 2017 (rainfall event recorded at Lentia station the 11 November 2017, with a total of 46.5 mm). In summary, Table 6 shows how a lapilli-rich tephra-fallout deposit with a low hydraulic conductivity is unstable for slopes >30∘ regardless of the associated thickness (deposit features of Pal D eruption); nonetheless, Fig. 10 shows that the deposit becomes stable for thickness values >65 cm. In fact, a constant FS (dashed lines in Fig. 12b,d) is the

result of the upper boundary of pressure head, which is physically limited at the beta-line (Iverson, 2000; Baum et. al., 2008). The total pore pressure cannot be above the values denoted by a water table at the ground surface (beta-line) and the model calculates FS with this value, which is the worst condition for instability. In contrast, the same lapilli-rich tephra-fallout

deposit is unstable at all observed slopes only for thickness values <10-20 cm in case of high hydraulic conductivity. Finally, a tephra-fallout deposit dominated by coarse ash is unstable at slopes <35.4∘ mostly for thickness values between about 10-40 cm (deposit features of the 1888-90 eruption); for a slope >38∘ the same tephra-fallout deposit is unstable for thickness values larger than 13 cm. For the same tephra-fallout deposit the ratio of rainfall intensity and hydraulic conductivity ($I/K_s$) determines the time to reach the water table (located at the bottom of the deposit in our case study); the rate of rise of water table increases with an increase in $I/K_s$ ratio (Li et al., 2013). In the case of the 1888-90 tephra-fallout deposit, the upper critical thickness for instabilities increases with the increase of rainfall intensity and total rainfall. It is also important to note how the maximum value of total pore pressure, and, therefore, the potential for triggering lahars, is shifted toward larger values of tephra-fallout deposit thickness when rainfall intensity is increased. As an example, the maximum value of pore pressure for 38° slope angle (blue solid line in Figs 12a,c) is reached at 15 cm and 30 cm for a rainfall intensity of 6.4 mm h$^{-1}$ and 15.5 mm h$^{-1}$, respectively.

## 5. Discussion

### 5.1 Characteristics of lahar source deposits

Rain-triggered lahars associated with both tephra-fallout and PDC deposits are associated with a variety of precipitation, grainsize, hydraulic conductivity and infiltration capacity (Table 5). It is important to note that infiltration capacity and hydraulic conductivity can be considered as similar parameters for the sake of this comparison. In fact, the infiltration capacity is a measure of the rate at which soil is able to absorb water (Horton, 1945), while the hydraulic conductivity measures the ease with which water will pass through a porous media (Darcy, 1856). Infiltration capacity typically decreases through time and converges to a constant value, which is the hydraulic conductivity. Infiltration capacity is more easily measured in the field, while hydraulic conductivity is more easily measured in the laboratory. Examples of lahar generation enhanced by fine-grained deposits include Mt. St. Helens 1980 (Leavsley et al., 1989), Chaitén 2008 (Pierson et al., 2013) and Córdon Caulle 2011 (Pistolesi et al., 2015) (Table 6). In contrast, the grainsize of the Vulcano 1888-90 tephra-fallout deposit is closer to Mt. Unzen PDC and Pinatubo tephra-fallout deposit. Hydraulic conductivity associated with the Vulcano 1888-90 tephra-fallout deposit is 44 times higher than the infiltration capacity of the PDC in Shultz Creek (Mt. St. Helens). Infiltration capacity is low also in the case of Mt. Unzen but is higher for the PDCs of Mt. Pinatubo. If we also compare the lahar volumes of Mt. St. Helens 1980, Pinatubo 1990, Chaitén 2008 and Vulcano 1888-90, we observe that the volumes are in the range of million cubic meters for the three first volcanoes, while in the case of Vulcano the larger events were only in the range of thousand cubic meters. An important difference between the deposits studied in this paper and the other deposits is the climatic conditions (Table 7). In fact, Vulcano is characterized by a semi-arid, poorly vegetated regions with non-permanent streams and limited annual rainfall (500 mm), while all other cases are characterized by a forested area with permanent streams draining the volcano flanks and annual precipitation between 1000 mm and until 4300 mm.

Lahar triggering is clearly influenced by hydraulic conductivity and infiltration capacity of the primary deposits, which, in turns, are strongly related to deposit grainsize. The highest hydraulic conductivities ($1\times10^{-2}$) are associated with Md$\phi$ < -1$\phi$ (lapilli), while the lowest hydraulic conductivities and infiltration capacities (between $5\times10^{-5}$ and $1.9\times10^{-6}$) result for Md$\phi$ > 1 (coarse and fine ash), except for the PDCs at Mt. Unzen where Md$\phi$ is between -1 and 1$\phi$ (Table 6). Nonetheless. in the case of Mt. Unzen, the poor infiltration capacity is not due to fine grainsize but to the development of an impermeable crust on the top of the deposit (Yamakoshi et al., 2000). The combination of hydraulic conductivity (or infiltration capacity) and rainfall intensity influences the lahar triggering mechanism either in terms of slope failure or erosion (Cuomo and Della Sala, 2013). If hydraulic conductivity exceeds rainfall intensity only infiltration occurs, but if rainfall intensity exceeds hydraulic conductivity runoff (overland flow) occurs (Cuomo and Della Sala, 2013; Pierson et al., 2014).

The effect of grainsize on runoff has also been investigated based on laboratory experiments. As an example, Jones et al. (2017) have investigated the behaviour of two tephra-fallout samples with contrasting grainsize (a fine grained sample from Chaitén 2008 eruption (D50= 4.2 φ, fine ash) and a coarse grained sample from Mt. Kelud 2014 eruption (D50= 0.9ϕ, coarse ash)) in relation to one rainfall intensity (150 mm/h). Experiments showed that surface sealing occurred within minutes of rainfall on dry fine-grained tephra but was not evident on coarser material. The surface sealing on fine-grained tephra reduces infiltration and enhances overland flow generating downslope sediment transportation. Additionally, antecedent rainfall and, thus, increased moisture content, increased runoff rates and reduced runoff lag time; thus, low rainfall intensities with short duration could still trigger lahars when the tephra residual moisture content is high (Jones et al. 2017). More experimental investigations should be carried out considering a range of rainfall intensities and more directly relating grainsize with hydraulic conductivity. Nonetheless, these outcomes confirm that lahar triggering mechanism is strongly influenced by grainsize, and, therefore, by hydraulic conductivity, and rainfall intensity, and could be complicated by deposit local grainsize, composition and weather patterns. Interesting to note that on Vulcano some specific material formed a solid crust that made it impermeable forming an ideal surface for shallow landsliding (e.g. Tufi Varicolori), while some other material remains unconsolidated over the years (e.g. the 1888-90 deposit); this is probably related to the grainsize and composition of the pyroclastic material and to the variable fumarolic activity at the time of deposition (De Fiore 1922; Fulignati et al. 2002).

## 5.2 Short versus long-lasting eruptions

The duration of a long-lasting eruption plays an important role in the pattern of remobilisation of tephra-fallout deposits. Different unstable volumes calculated with TRIGRS were obtained for durations of Vulcanian eruptive cycles between 3 and 24 months without considering remobilisation in between the eruptive cycles. The results show that for an eruption time of 18 months and a probability of occurrence of 25% (corresponding to a tephra-fallout deposit thickness between 17 and 33 cm) the unstable areas, and, therefore, the remobilised volumes from the lahar source areas, reached a maximum (1,105 m³ for the S upper catchment and 990 m³ for the NW upper catchment). For the eruption duration of 24 months, the increase of tephra-fallout deposit thickness (between 21 and 42 cm) produced a decrease in the unstable areas (95 m³ for the S upper catchment and 861 m³ for the NW upper catchment). It is worth noting also that the thickness of the deposit affects both the driving and resisting forces along the slip surface at the bedrock contact. In addition, a higher soil thickness also increases the time for rainfall to produce significant changes in pore water pressure at the bedrock contact. Thus, a lower volume of remobilised material may occur despite a thicker deposit. The results obtained with TRIGRS showed that there is an unstable window of tephra-fallout deposit thickness, which depends on amount and duration of rainfall, slope angle and geotechnical characteristics of the deposit (Table 6). These results are in agreement with the window of potentially unstable soil thickness found by Dietrich et al. (2007) for a range of non-volcanic case studies. In their work, Dietrich et al. (2007) have adopted a slope stability model that includes shear resistance due to lateral and basal boundaries as a result of combination of cohesion (soil and root cohesion) and friction.

The morphology of the middle and the upper part of the La Fossa cone shows a strong remobilisation of the 1888-90 eruption tephra-fallout deposit. The coarse-ash grainsize range and medium permeability of the 1888-90 tephra-fallout deposits in combination with the impermeable deposits at the base of the sequence (i.e., Tufi Varicolori) make this deposit easily remobilised by rainfall through a shallow landslide initiation mechanism. Deep channels due to the continuous remobilisation of this deposit can be observed on the cone (Fig. 1). Because of the short transport distance (200-400 m) the lahar deposits on the La Fossa cone have almost the same grainsize as the 1888-90 tephra-fallout deposits (Fig. 5). The same relation between the primary pyroclastic deposits and the lahars has been described for the la Cuesta succession (Valentine et al., 1998).

Field evidence for post Pal D remobilisation and lahar deposits are not recorded in the stratigraphic record (Di Traglia,

2011). This is consistent with our modelling results with $K_s = 1 \times 10^{-2}$ m s$^{-1}$ that shows the low potential of remobilisation associated with thick lapilli deposits. In this case, a high hydraulic conductivity allows the water to rapidly migrate down to the water table with low transient pressure. Therefore, the water table rarely rises sufficiently to induce instability, which explains why thicker deposits are relatively less unstable. In fact, the thick lapilli deposits associated with both VEI 2 and 3 and a high permeability are stable even for the largest rainfall event occurring on Vulcano, e.g. VEI 3 and 25% probability of occurrence (Table 4). Studies on rainfall lahar generation in Mayon volcano, Philippines also demonstrated that coarse and high permeability pyroclastic deposits on volcano slopes remain stable in most cases (Rodolfo and Arguden, 1991). A similar case occurred at Mt. St. Helens, where rainfall-induced lahar drastically dropped when the erosion of fine ash exposed coarser and more permeable material (Collins and Dunne, 1986).

## 5.3 Initiation mechanisms of rain-triggered lahars

The tephra remobilisation model used in our study assumes rainfall-induced shallow landslides caused by the infiltration of rain on the slope surface. These shallow landslides can eventually transform into lahars depending on the availability of water, slope morphology and characteristics of tephra deposits. In particular, we studied the cases in which rainfall intensity (I) is lower than hydraulic conductivity ($Ks$) and infiltration occurs before runoff (Cuomo and Della Sala, 2013). At Vulcano, both the 1888-90 and the Pal D tephra-fallout deposits have high permeability compared to the cases of Mt. St. Helen 1980 and Chaitén 2008 fine-grained tephra-fallout deposits (Table 3). The fine-grained tephra-fallout deposits reduce infiltration capacity on basin slopes, enhancing runoff and producing larger peak flows. However, we cannot discard the mechanism of sheet and rill erosion in Vulcano, which was not simulated in this study. The physical characteristics of primary tephra-fallout deposits (e.g. high hydraulic conductivity) and the rainfall characteristics on Vulcano indicate that the main lahar initiation mechanism is most likely shallow landsliding.

The relationship between unstable areas and deposit thicknesses suggests a significant influence of the hydraulic conductivity on the model outcomes and on the resulting estimation of unstable volumes (Fig. 7 and Tables 4 and 5). Our results better explain the parameter values affecting slope instability (Fig. 10 and Tables 4 and 5). In fact, the tephra-fallout deposit thickness of 21-33 cm associated with the largest unstable volumes for the Vulcanian scenarios (18- and 24-months durations), well correlates with the thickness of 20-30 cm shown in Figure 10. Similarly, the tephra-fallout deposit thickness associated with the largest unstable volumes for the VEI 2 and 3 scenarios (with Ks derived from literature, i.e. $1 \times 10^{-2}$ m s$^{-1}$), i.e. 8-25 cm, also shows how a higher hydraulic conductivity generates lahars for lower deposit thickness. These values of tephra-fallout deposit thickness are in good agreement with the critical threshold for the lahar generation found by Sulpizio et al. (2006) for syn-eruptive lahars in the Vesuvian area (i.e. 10 cm).

For the 1888-90 tephra-fallout deposit, results suggest that cohesion leads to a critical minimum landslide depth size (lower limit deposit thickness for instability) dependent on the slope angle (Fig. 11b, d; Table 6). Using a model for natural slopes, Milledge et al. (2014) found a critical depth in cohesive soil, resulting in a minimum size for failure. For cohesionless material, such as the primary Pal D, the lower thickness limit is not defined as most small deposit thickness is unstable and become progressively stable with deposit thickness increase depending on rainfall intensity, duration and slope angle (Fig. 12b, d; Table 6). The different behaviour shown by the different tephra-fallout deposits modelled in our study could relate to the fact that rainfall-induced slope failure can occur by two mechanisms (Li et al. 2013): 1) rainfall infiltration that produces a rise of groundwater generating positive pore pressure and adds weight on the slope (Cho and Lee, 2002; Crosta and Frattini, 2003; Soddu et al., 2003); 2) rainfall that results in a propagation of wetting front causing an increase in water content and pore pressure (loss in matric suction) (Ng et al., 2001; Collins and Znidarcic, 2004; Rahardjo et al., 2007). First, in the case of subplinian tephra-fallout deposit with $K_s = 6.8 \times 10^{-4}$ m s$^{-1}$ (low conductivity), wetting front mechanism moves from the ground surface toward the bedrock, which means that the time for the perturbation to reach the bedrock contact increases with deposit thickness. As a result, for the same rainfall, the higher the thickness the more stable the slope (Fig. 10,

green curve). Second, in the case of subplinian tephra-fallout deposit with $K_s = 1 \times 10^{-2}$ m s$^{-1}$ (high conductivity) the water moves fast down to the water table with low transient pressure (a sort of drained conditions). This means that the water table should be the same provided by the same rainfall, independently from the total deposit thickness. The slope is very sensitive to the ratio of water table/total thickness and becomes stable with the increase of deposit thickness. Finally, in the case of the 1888-90 tephra-fallout deposit, the increase in deposit stability (right side of the red curve in Fig. 10) could be explained by both mechanisms described above, but the first decrease in stability (left side of the red curve) indicates that for small deposit thicknesses the pore pressure reached at the end of the rainfall is not enough to neutralize the effect of cohesion (0.5 kPa).

The results obtained with the TRIGRS model show the potential for the evaluation of transient pore-water pressure stability condition and lahar (landslide) source areas during rainfall (Godt et al., 2008), even though the role of suction in unsaturated condition, which plays a fundamental role for the pore pressure regime, is not included in the model (Sorbino et al., 2010). Matrix suction between 24 and 27 kPa were measured in 1888-90 primary tephra deposits in May 2018 (at the beginning of the dry season), but further seasonal matrix suction variation needs to be performed to evaluate the role of suction in potential unstable areas evaluation and the most critical period for slope stability (Pirone et al., 2016). Finally, our deposit-stability analysis could be largely strengthened by the validation with the volume of observed lahar deposits that, unfortunately, is difficult to obtain for the 1888-90 eruption due to complex deposit correlation.

### 5.4 Impact and risk implications

Remobilised tephra-fallout volume was calculated with TRIGRS for two different catchments with same area, one located on the NW flank and the other on S flank of La Fossa volcano, and different values were obtained for the same eruption scenarios (Figs. 8 and 9). Two main factors are responsible for these differences in volume. The first factor is that the tephra deposit is thicker for the S flank due to the prevailing wind direction to the SE and, therefore, it inhibits the formation of lahars as it requires more water to be remobilised (which is not frequently available in the Vulcano area). In fact, there is a thickness threshold for instability depending on rainfall intensity and tephra-fallout deposit properties (Figs. 11 and 12). An additional factor influencing the deposit stability is the slope morphology. Steep slope (> 35°) are more frequent on the NW flank, with 42% for the NW basin and 32% for the S basin, which explains a higher percentage of unstable area for the NW part. As a result, due both to the lower deposit thickness and to the steeper slopes, the NW flank is more likely to generate lahars than the S flank, even though lahars from the S flank can also be significant (Figs. 8 and 9). It is important to consider that one of the most populated part of the island, which is also where the key infrastructures are located (i.e., Porto Village; Galderisi et al., 2013), is directly exposed to the lahars potentially generated on the NW flank of the volcano. In contrast, the residential area of Piano located on the S of the island is protected by caldera rim that could easily block all lahars forming on the S flank. It is also important to highlight the importance of assessing the effect of compounding hazards in the case of multi-hazard environments such as volcanic eruptions. In fact, volcanic hazards are often assessed individually, while investigation of the associated cascading effects such as for tephra sedimentation and lahars should be considered (e.g. Volentik et al. 2009, Tierz et al. 2017). Our results demonstrate the effectiveness and strength of combining probabilistic tephra hazard modelling with both physically-based lahar-triggering modelling and physical and geotechnical characterisation of the pyroclastic material. The next step necessary to assess the impact of the combination between tephra sedimentation and lahar generation is lahar-inundation modelling. Clearly, each step requires dedicated studies and investigations and has some intrinsic value on its own; however, the combination of all aspects has a tremendous potential for the impact assessment of communities located in volcanic areas.

## 6. Conclusions

We presented a detailed analysis of the volume of tephra-fallout deposit that could be potentially remobilised by rainfall as a result of two likely eruptive scenarios of La Fossa volcano, the main volcanic system on Vulcano island: a long lasting Vulcanian eruption (i.e., using the 1888-90 eruption as the reference event) and a short-lived eruption (VEI 2 and VEI 3; using the Pal D eruption as the reference event) (Fig. 1 and Table 1). The great novelty of this work is the assessment of compounding hazards (tephra-fallout deposits and lahar triggering) based on both numerical modelling and field and geotechnical characterization of the source deposit. In fact, volumes of tephra-fallout deposit that could be remobilised by rain-triggered lahars were analysed combining a tephra sedimentation model (TEPHRA2) and slope stability model (TRIGRS) in combination with field observations and geotechnical tests.

We have considered 12 probabilistic isopach maps with different eruption duration and probabilities of occurrence of 25% and 75% in the case of the Vulcanian eruptive scenario. We have also considered 4 probabilistic isopach maps for two short-lived eruptions of VEI 2 and 3 and same probabilities of occurrence as in the case of the Vulcanian eruptive scenario. In addition, a parametric analysis was performed with TRIGRS to determine the tephra-fallout thickness thresholds required to trigger lahars for a given rainfall event. Two basins of same area were identified on the NW and S flank of the volcano to analyse the effect of different morphology and of different accumulation related to the prevailing wind direction. The results of unstable volumes for the two basins show that:

1) for the Vulcanian scenario, the largest unstable volume is reached for an eruption duration of 18 months and a 25% probability of occurrence scenario, with a volume of 1,105 m$^3$ for the S basin and 990 m$^3$ for the NW basin (Fig. 8);

2) for the subplinian scenario, the largest unstable volume (2,455 m3) resulted for the VEI 3 (25% of occurrence) with a Ks= $6.8 \times 10^{-4}$ m s$^{-1}$ (i.e. hydraulic conductivity measured in the field) in the case of the NW basin (Fig. 9);

3) for the subplinian scenario with Ks= $1 \times 10^{-2}$ m s$^{-1}$ (i.e. hydraulic conductivity estimated from literature) the largest unstable volume (563 m$^3$) was found for the NW basin with a scenario VEI 2 (25% of occurrence) (Fig. 9).

For a tephra-fallout deposit with features associated with a Vulcanian eruption we observe an unstable window of deposit thickness suggesting that particle cohesion leads to a critical minimum landslide depth, which is dependent on the slope angle; an increase in rainfall intensity enlarges the windows of thickness instability (Table 6). In contrast, for cohesionless material such as the primary Pal D, a low thickness limit of instability is not reached, and the deposit becomes stable with thickness increase depending on rainfall intensity and slope angle (Table 6). In particular, the parametric analysis with variable tephra-fallout thickness and slope and two rainfall intensities of 6.4 mm h$^{-1}$ for 5 hours and 15. 5mm h$^{-1}$ for 3 hour shows that:

1) for a tephra-fallout deposit with features associated with a Vulcanian eruption, the thickness generating the largest instability is between 20 and 27 cm for 6.4 mm h$^{-1}$ and between 20 and 35 cm for 15.5 mm h$^{-1}$ (Fig. 11 and Table 6);

2) for a tephra-fallout deposit with features associated with a subplinian eruption with $K_s$= $1 \times 10^{-2}$ m s$^{-1}$, the unstable area decreases rapidly with an increase in deposit thickness, being all the area almost stable beyond a thickness of 32 cm (Fig. 12 and Table 6);

3) for a tephra-fallout deposit with features associated with a subplinian eruption with $K_s$= $6.8 \times 10^{-4}$ m s$^{-1}$, almost all the lahar source area results unstable (98%) for deposit thickness < 65 cm (Figs 10, 12 and Table 6);

The results modelled with TRIGRS show that shallow landsliding is an effective process for eroding both Vulcanian-type and subplinian-type (with $K_s$= $6.8 \times 10^{-4}$ m s$^{-1}$) tephra-fallout deposits in combination with high-intensity rainfall events with short duration, such as those occurring in Vulcano every year. Nonetheless, the occurrence of shallow landsliding is a complex process (e.g., Table 4 and Fig. 10) and the tephra-fallout deposit thickness threshold strongly depends on rainfall

intensity, tephra-fallout deposit characteristics and geomorphology features. Both eruptive scenarios (e.g., plume height, erupted mass, eruption duration) and prevailing wind direction are, therefore, crucial to the generation of rain-triggered lahars, having a first order control on tephra-fallout deposit thickness. Physical characteristics of tephra-fallout deposits (e.g. hydraulic conductivity, grainsize, friction angle and cohesion), geomorphological features (e.g. flank slopes), the characteristics of soil at the base of the deposits and vegetation are also important parameters to consider as they have a first order control on slope instability. We can conclude that deposit thickness and rainfall intensity alone are not sufficient to derive thresholds for lahar triggering; a comprehensive assessment of unstable volumes that could potentially trigger lahars, in fact, requires dedicated numerical simulations combined with detailed field observations and geotechnical analysis as we have shown in this study.

## Data availability

Most data are made available in main tables and supplementary material. Additional data is available upon request, based on a collaborative agreement.

## Appendix

### Appendix A: Field strategies

### Sampling of undisturbed deposit for geotechnical tests

Undisturbed tephra-fallout deposit is sampled for testing the properties in the laboratory, without disturbing structure texture, density, natural water content and stress condition (Figs A1a and b). Sampling was performed by inserting a steel tube 3 mm thick with a height of 30 cm and a diameter of 10 cm into the ground (Fig. A1a). After that, we cleaned all the deposit around the tube to extract it with a minimum disturbance. Then, a support was inserted at the base of the cylinder, and the tube was extracted from the deposit. Finally, the tube was covered on both ends with a plastic cover and plastic tape to preserve the deposit for disturbance during the transport (Fig. A1b).

### Soil suction measurement

Soil suction measurements were carried out in situ on the 1888-90 tephra-fallout deposit with a soil moisture probe ("Quick Draw" Model 2900FI) (Fig. A2b). The first step in taking a reading with the probe is to core a hole pushing the coring tool into the soil (Fig. A2a). After removing the coring tool, we have a proper sized hole to insert the probe. The second step is to insert the probe in the soil and wait approximately one minute (equalization time assessed for such soils). Finally, the suction can be read on the dial gauge (Fig. A2b). The soil suction is created by water capillary pressure that each soil particle applies into the soil. The moisture probe has a porous ceramic sensing tip at the end of the tube. The soil suction reading is obtained when a small amount of water transfers between the sensing tip of the probe and the soil.

### Saturated hydraulic conductivity measurement

The saturated hydraulic conductivity was estimated in the field with a single-ring permeameter for both deposits (Figs A3a and b). The apparatus for the measurements consists of a steel ring with diameter of 21 cm and height of 12 cm, a plastic cover with a hole to insert a Mariotte bottle (Fig. A3b). In the field, we put the ring on a horizontal plane surface on the tephra-fallout deposit. Then, the first 6 cm were pushed into the ground. Finally, we filled the Mariotte bottle with water and inserted it on the tape turned upside-down. The water first formed a 1 cm layer on the tephra-fallout deposit and then started to infiltrate into the deposit. For the infiltration rate measurements, the readings were done every minute in the case of 1888-90 eruption deposit and every 30 seconds in the case of Pal-D deposit. The duration of the measurements was 40 minutes for 1888-90 deposit and 3.2 minutes for the pal-D deposit.

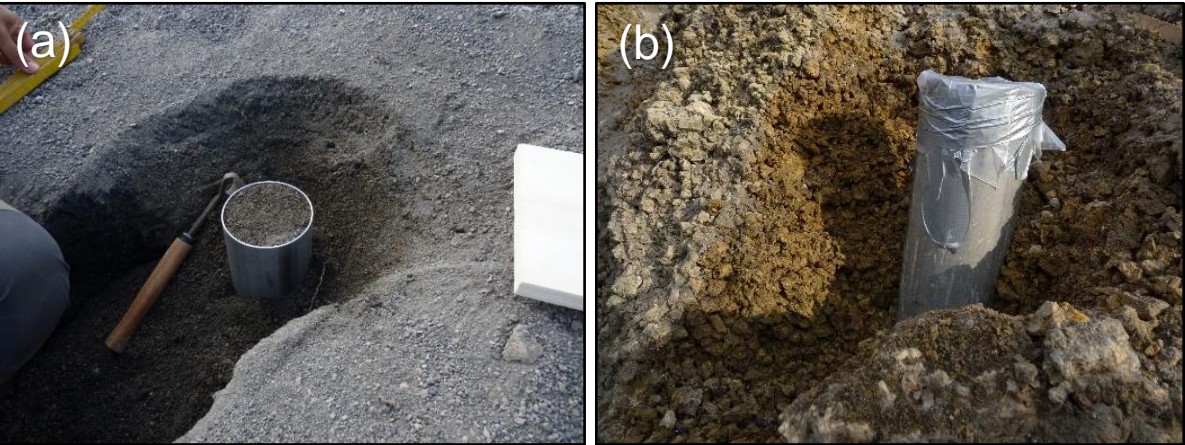

**Figure A1: a) Sampling the 1888-90 AD tephra-fallout deposit with a 30 cm steel tube. We cleaned all the deposit around the tube to extract it with a minimum disturbance. B) Sampling the Pal-D tephra-fallout deposit with a 30 cm steel tube. The tube is covered with a plastic cover and plastic tape before extracting it from the deposit with a basal support.**

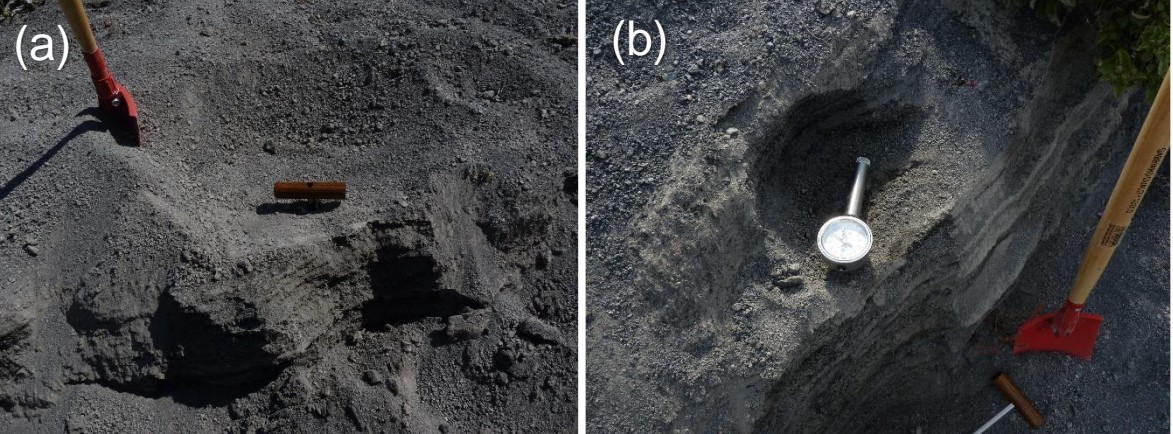

**Figure A2: a) Coring tool into the deposit before soil suction measurement b) Soil suction measurement on the 1888-90 primary**
10 **tephra-fallout deposit on the NW volcano flank.**

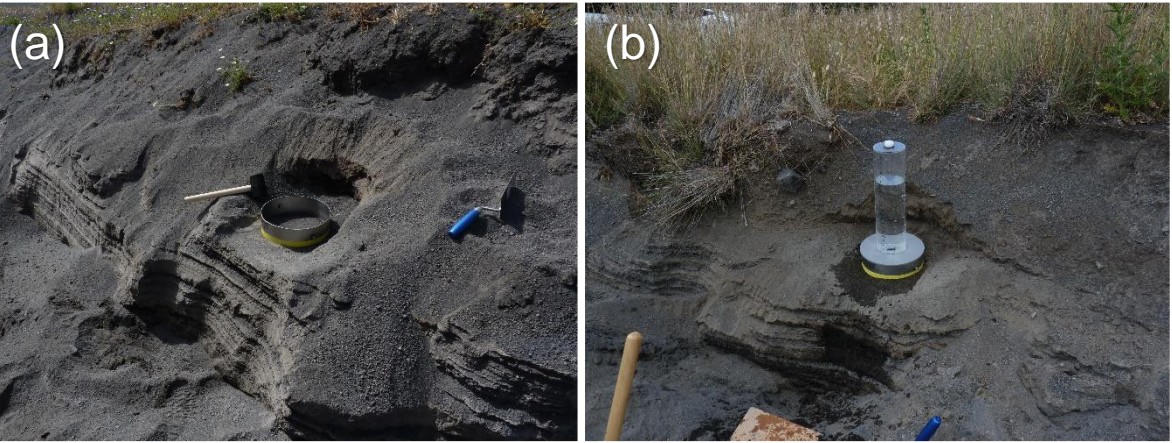

**Figure A3: a) Ring infiltrometer: 6 cm are buried into the 1888-90 tephra-fallout deposit. b) Ring infiltrometer during the infiltration measurements showing the bottle turned upside down with water infiltrating in the deposit.**

**Author contribution**

VB, CB and SC conceived the study. VB, CB, SB, MP and AG were involved the field work. VB carried out sample characterization both in the field and in the laboratory. SB carried out the tephra fallout modelling. SC and MM carried out the geotechnical tests in laboratory. VB, SC and MM carried out the slope stability modelling. VB analysed the results and compiled the figures with input from the other authors. VB and CB prepared the manuscript with contributions from all co-authors. All the authors read, reviewed and approved all versions of the manuscript.

**Competing interests**

The authors declare that they have no conflict of interest.

**Acknowledgments**

The authors are grateful to Irene Manzella, Michel Jaboyedoff and Mario Sartori for valuable discussion and thank the NHESS editor, Jenni Barclay and an anonymous reviewer for detailed and constructive comments. The work was supported by the by the Swiss National Science Foundation (#200021_163152). M.P. was supported by a project "PRA 2018-19 (Progetti di Ricerca di Ateneo)" granted by University of Pisa.

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

**Tables**

Table 1: Number of observed eruptions for the different types of activity in the last 1000 years on Vulcano island. Values of frequencies and the detailed events for each type are also reported. In red, the selected scenarios used in Section 3.3. Based on data from Di Traglia et al. (2013) and De Astis et al. (1997, 2013).

| Eruption type | Frequency and Events | |
|---|---|---|
| **Phreatic eruptions** | Eruption 1727<br>Eruption 1444<br>Commenda | |
| | Number | 3 |
| | Freq. [a$^{-1}$] | $3.0 \times 10^{-3}$ |
| **Effusive activity** | Pietre Cotte,<br>Palizzi, Commenda,<br>Vulcanello 3 (2), Vulcanello 1, Vulcanello 2, Punte Nere,<br>Campo Sportivo | |
| | Number | 9 |
| | Freq. [a$^{-1}$] | $9 \times 10^{-3}$ |
| **Strombolian activity** | Vulcanello 3, Vulcanello 2,<br>Vulcanello 1 | |
| | Number | 3 |
| | Freq. [a$^{-1}$] | $3.0 \times 10^{-3}$ |
| **Vulcanian cycles** | **1888-90**<br>Pietre Cotte (3)<br>Palizzi (1) | |
| | Number | 5 |
| | Freq. [a$^{-1}$] | $5.0 \times 10^{-3}$ |
| **Sustained eruptions** | Event in Pietre Cotte<br>Palizzi trachitic (**PAL D**)<br>Palizzi rhyolitic (Pal B) | |
| | Number | 3 |
| | Freq. [a$^{-1}$] | $3.0 \times 10^{-3}$ |

Table 2: Geotechnical parameters for the subplinian tephra-fallout deposit (Pal D) and the Vulcanian tephra-fallout deposit associated with the 1888-90 eruption (Vulc).

| unit | $K_s$ | $D_0$ * | $\varphi'$ | $\Upsilon_s$ wet | $\Upsilon_s$ dry | c´ | $\theta_s$ | $\theta_r$ | $\alpha$ | $G_s$ | n | e |
|---|---|---|---|---|---|---|---|---|---|---|---|---|
| | m s$^{-1}$ | m$^2$ s$^{-1}$ | deg | kN m$^{-3}$ | kN m$^{-3}$ | kPa | % | % | kPa$^{-1}$ | - | - | - |
| **Pal D** | $1 \cdot 10^{-2}$ | $6.59 \cdot 10^{-3}$ | 54.00 | 13.70 | 6.64 | 0.00 | 0.72 | 0.04 | 0.93 | 2.42 | 0.72 | 2.57 |
| **Vulc** | $8.50 \cdot 10^{-5}$ | $3.28 \cdot 10^{-4}$ | 40.98 | 17.00 | 13.51 | 0.00 | 0.47 | 0.03 | 0.28 | 2.57 | 0.47 | 0.87 |

c': cohesion; φ': friction angle; γ$_s$: total unit weight of the soil; K$_s$: saturated conductivity; D$_0$: saturated diffusivity; θ$_s$: saturated water content; θ$_r$: residual water content; α: Gardner parameter; Gs: specific gravity; n: porosity; e: void ratio. * Diffusivity was evaluated using the procedure proposed in the paper: Rossi et al. (2013).

Table 3: Input parameters for subplinian ($K_s$ from literature), subplinian *($K_s$ measured in the field) and the 1888-90 Vulcanian eruption scenarios used for simulation with TRIGRS.

| unit | $K_s$ (m s$^{-1}$) | $D_0$ (m$^2$ s$^{-1}$) | $\varphi'$ (deg) | $\Upsilon_s$ (kN m$^{-3}$) | c´ (kPa) | $\theta_s$ (%) | $\theta_r$ (%) | $\alpha$ (kPa$^{-1}$) |
|---|---|---|---|---|---|---|---|---|
| vulcanian | 8.50x10-5 | 1 x 10-4 | 41.00 | 17.00 | 0.5 | 0.47 | 0.029 | 0.28 |
| subplinian | 1x 10$^{-2}$ | 6.59x10$^{-3}$ | 54.00 | 13.70 | 0.00 | 0.72 | 0.04 | 0.93 |
| subplinian * | 6.8x10$^{-4}$ | 6.59x10$^{-3}$ | 54.00 | 13.70 | 0.00 | 0.72 | 0.04 | 0.93 |

c': cohesion; φ': friction angle; γ$_s$: total unit weight of the soil; K$_s$: saturated conductivity; D$_0$: saturated diffusivity; θ$_s$: saturated water content; θ$_r$: residual water content; α: Gardner parameter;

Table 4: Unstable areas (FS≤1) for long-lasting vulcanian eruption and subplinian (VEI: $K_s$ from literature; VEI*: $K_s$ measured in the field) calculated with TRIGRS for NW and S source areas (Fig. 6). Rainfall intensity 6.4 mm h$^{-1}$ with a duration of 5 h. Thickness: tephra-fallout deposit thickness from probabilistic isopach maps considered in the model. Input parameters used for the simulation are described in Table 3. VEI: Volcanic explosivity index.

**Long-lasting Vulcanian eruption scenario**

| Probability (%) | Duration (month) | NW source area | | S source area | |
|---|---|---|---|---|---|
| | | Thickness (cm) | percentage of unstable area (%) | Thickness (cm) | percentage of unstable area (%) |
| 25 | 9 | 11-14 | 34 | 13-17 | 57 |
| 25 | 12 | 14-18 | 69.5 | 16-22 | 81.6 |
| 25 | 18 | 21-26 | 89.4 | 24-31 | 66.6 |
| 25 | 24 | 26-33 | 52.3 | 30-39 | 22.9 |
| 75 | 9 | 6.5-8 | 0.08 | 7.6-10 | 0 |
| 75 | 12 | 8-10 | 7.1 | 8-12 | 10.9 |
| 75 | 18 | 9-12 | 17.9 | 11-14 | 31.1 |
| 75 | 24 | 10-12.8 | 20.7 | 11-15 | 36.8 |

**Subplinian eruption scenario**

| Probability | VEI | NW source area | | S source area | |
|---|---|---|---|---|---|
| | | Thickness (cm) | percentage of unstable area (%) | Thickness (cm) | percentage of unstable area (%) |
| 25 | 2 | 8-25 | 79.1 | 15-40 | 14.7 |
| 25 | 3 | 36-95 | 1.9 | 61-112 | 0.2 |
| 75 | 2 | 1-3 | 99.1 | 2-4 | 97.1 |
| 75 | 3 | 3-10 | 0.9 | 5-19 | 0.5 |
| 25 | 2* | 8-25 | 99 | 15-40 | 97 |
| 25 | 3* | 36-95 | 97 | 61-112 | 25 |
| 75 | 2* | 1-3 | 99 | 2-4 | 97 |
| 75 | 3* | 3-10 | 99 | 5-19 | 97 |

Table 5: Total and unstable volumes of primary tephra deposits for long-lasting vulcanian eruption and subplinian (VEI: $K_s$ from literature; VEI*: $K_s$ measured in the field) calculated with TRIGRS for NW and S upper catchments. NW and the S UP (upper catchments) have the same area (4665 m$^2$), with a mean slope of 43.5° and 40.1°, respectively. Rainfall intensity 6.4 mm h$^{-1}$ with a duration of 5 h. Thickness: tephra-fallout deposit thickness from probabilistic isopach maps considered in the model.

**Long-lasting Vulcanian eruption scenario**

| Probability (%) | Duration (months) | NW catchment | | | S catchment | | |
|---|---|---|---|---|---|---|---|
| | | Thickness (cm) | Volumes (m³) | | Thickness (cm) | Volumes (m³) | |
| | | | Total | Unstable | | Total | Unstable |
| 25 | 9 | 11-13 | 574 | 246 | 15-16 | 734 | 585 |
| 25 | 12 | 14-16 | 748 | 601 | 20-21 | 959 | 952 |
| 25 | 18 | 20-24 | 1061 | 991 | 28-30 | 1353 | 1105 |
| 25 | 24 | 25-30 | 1326 | 861 | 35-37 | 1689 | 95 |
| 75 | 9 | 6.5-7.5 | 333 | 0 | 9 | 427 | 0 |
| 75 | 12 | 9-10 | 417 | 0 | 11 | 533 | 21 |
| 75 | 18 | 9.5-11.5 | 489 | 47 | 13-14 | 627 | 231 |
| 75 | 24 | 10-12 | 507 | 74 | 13.5-14.5 | 649 | 319 |

**Plinian eruption scenario**

| Probability (%) | VEI | NW catchment | | | S catchment | | |
|---|---|---|---|---|---|---|---|
| | | Thickness (cm) | Volumes (m³) | | Thickness (cm) | Volumes (m³) | |
| | | | Total | Unstable | | Total | Unstable |
| 25 | 2 | 11-15 | 589 | 563 | 32-40 | 1663 | 0 |
| 25 | 3 | 37-58 | 2455 | 0 | 101-112 | 5112 | 0 |
| 75 | 2 | 1.2-2 | 77 | 77 | 3.2-3.6 | 187 | 185 |
| 75 | 3 | 2-8 | 245 | 244 | 13-16 | 736 | 401 |
| 25 | 2* | 11-15 | 589 | 604 | 32-40 | 1663 | 1636 |
| 25 | 3* | 37-58 | 2455 | 2455 | 101-112 | 5112 | 0 |
| 75 | 2* | 1.2-2 | 77 | 77 | 3.2-3.6 | 187 | 187 |
| 75 | 3* | 2-8 | 245 | 245 | 13-16 | 736 | 731 |

Table 6. Summary description of outcomes of Figs 11 and 12 showing the relation between tephra-fallout deposit thickness and Safety Factor (FS) based on various key parameters (i.e. tephra-fallout properties, slope angle, rainfall intensities). Unstable deposit thickness is shown in red. The ratio between rainfall intensity and hydraulic conductivity ($I/K_s$) is also shown as an indication of the time for the rainfall water to reach the bottom of the deposit.

| Tephra-fallout properties | Slope | Tephra fallout thickness (cm) | | Stability |
|---|---|---|---|---|
| | | Rainfall $I = 6.4$ mmh$^{-1}$ $D = 5$ hours | Rainfall $I = 15.5$ mmh$^{-1}$ $D = 3$ hours | |
| **1888-90** | | $I/K_s = 0.02$ | $I/K_s = 0.05$ | |
| Md$\varphi$ = - 0.90 - 1 | 38° | 0-12 | 0-12 | stable (FS > 1) |
| $K_s = 8.50 \times 10^{-5}$ m s$^{-1}$ | | **13 - 50** | **13 - 50+** | **unstable (FS <1)** |
| $\varphi' = 41°$ | | 0-10 | 0-12 | stable (FS > 1) |
| c´= 0.5 kPa | 35.4 ° | **11-40** | **11-50** | **unstable (FS <1)** |
| | | 41-50 | 50 + | stable (FS > 1) |
| | 30.1° | 0-19 | 0-19 | stable (FS > 1) |
| | | **20-27** | **20-35** | **unstable (FS <1)** |
| | | 28-50 | 36-50 | stable (FS > 1) |
| **Pal-D - high Ks** | | $I/K_s = 0.0001$ | $I/K_s = 0.0004$ | |
| Md$\varphi$ = - 3.42 | 38° | **0-21** | **0-32** | **unstable (FS <1)** |
| $K_s = 1 \times 10^{-2}$ m s$^{-1}$ | | 22-50 | 33-50 | stable (FS > 1) |
| $\varphi' = 54°$ | 35.4° | **0-18** | **0-26** | **unstable (FS <1)** |
| c´= 0 kPa | | 19-50 | 27-50 | stable (FS > 1) |
| | 30.1° | **0-13** | **0-21** | **unstable (FS <1)** |
| | | 14-50 | 22-50 | stable (FS > 1) |
| **Pal-D - low Ks** | | $I/K_s = 0.002$ | $I/K_s = 0.006$ | |
| Md$\varphi$ = - 3.42 | | | | |
| $K_s = 6.8 \times 10^{-4}$ m s$^{-1}$ | 38° | **0-50** | **0-50** | **unstable (FS <1)** |
| $\varphi' = 54°$ | 35.4° | **0-50** | **0-50** | **unstable (FS <1)** |
| c´= 0 kPa | 30.1° | **0-50** | **0-50** | **unstable (FS <1)** |

Table 7: Measured median grainsize, hydraulic conductivity and infiltration capacity on tephra-fallout (TF) and PDC deposits near volcanic vents and lahar volumes

| Eruption | Deposit type | Mdφ | P (mm) | $K_s$ (m/s) | I (m/s) | V (m³) | Data source |
|---|---|---|---|---|---|---|---|
| Vulcano (1888-90 vulcanian) | TF | -0.90 - 1 | 500 | 8.5x10-5 | ND | $10^3$-$10^4$ | this study, a |
| Vulcano (Pal D subplinian) | TF | -3.42 | 500 | 1x10-2 | ND | ND | this study |
| Cordón Caulle 2011 | TF-ash (unit III) | 2.15 | 2500-3000 | 5x10-5 | ND | ND | b |
| Cordón Caulle 2011 | TF-lapilli (unit I-II) | 0.2 | 2500-3000 | 3.9x10-2 | ND | ND | b |
| Chaiten 2008 | TF | < 3 | 2600-4300 | ND | ND | $3$-$8x10^6$ | c |
| Mt Pinatubo 1995 | PF | 0-3 | | ND | 1x10-4 | | d |
| | TF | -1.1- 1.7 | 1950 | ND | ND | $80$-$250x10^6$ | d, e |
| Mt Unzen 1990-95 | PF | -1-1 | 3100 | ND | 1.25x10-5; 5x10-6 | ND | f |
| Mt St Helen 1980 | TF | 1.73 | 1200 | ND | 1.9x10-6; 1.13x10-6 | $14x10^6$ | g, h |

Mdφ: Median grain size of deposit; P: annual precipitation; $K_s$: Hydraulic conductivity; I: Post-eruption infiltration capacity: V: lahars volume; ND = No data.

References: a) Di Traglia et al. (2013); b) Baumann et al. (2018); c) Pierson et al. (2013); d) Daag (2003); e) Janda et al. (1996); f) Yamakoshi and Suwa (2000); g) Leavesley et al. (1989); h) Pierson 1985

**Figures**

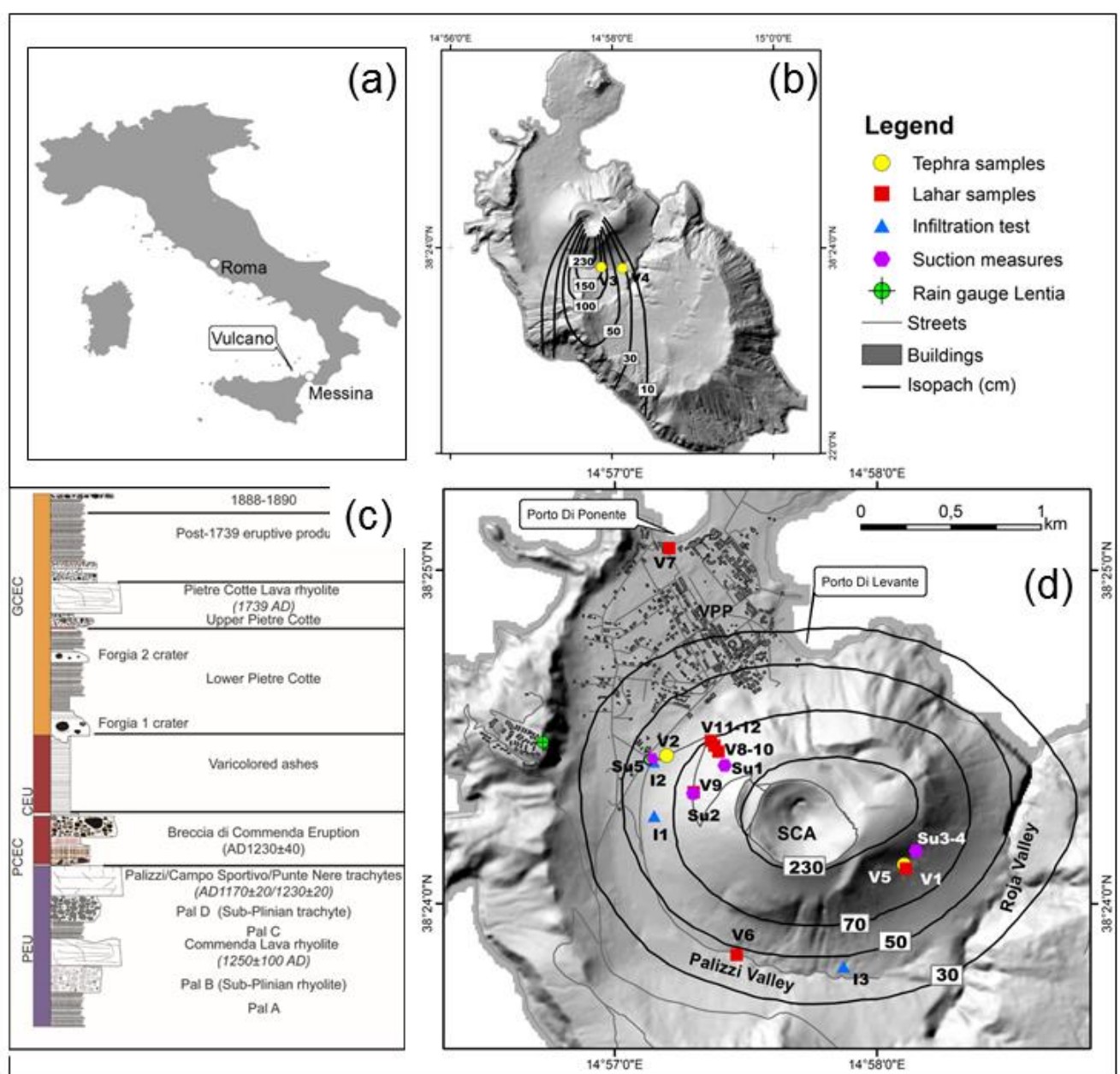

**Figure 1: Overview of the study area. A)** location of the island of Vulcano in Italy; **B)** Isopach map on shaded relief of Vulcano Island of the cumulative tephra-fallout Pal D deposit (after Di Traglia (2011); **C)** Simplified stratigraphy of the last 1000 years of La Fossa volcano based on DI Traglia et al. (2013) and De Astis et al. (1997, 2013) (see Table 1 for more details); **D)** isopach map of the cumulative 1888-90 tephra-fallout deposit (after Di Traglia, 2011) and sample locations. Sample names refer to *V* samples of tephra fallout (yellow circles) and lahars (red squares), *I* infiltration measures (blue triangle) and *Su* suction measurements (pink diamond). *SCA*: summit cone area. *VPP*: Vulcano Porto Plain. The rain gauge of Lentia is also shown (green circle).

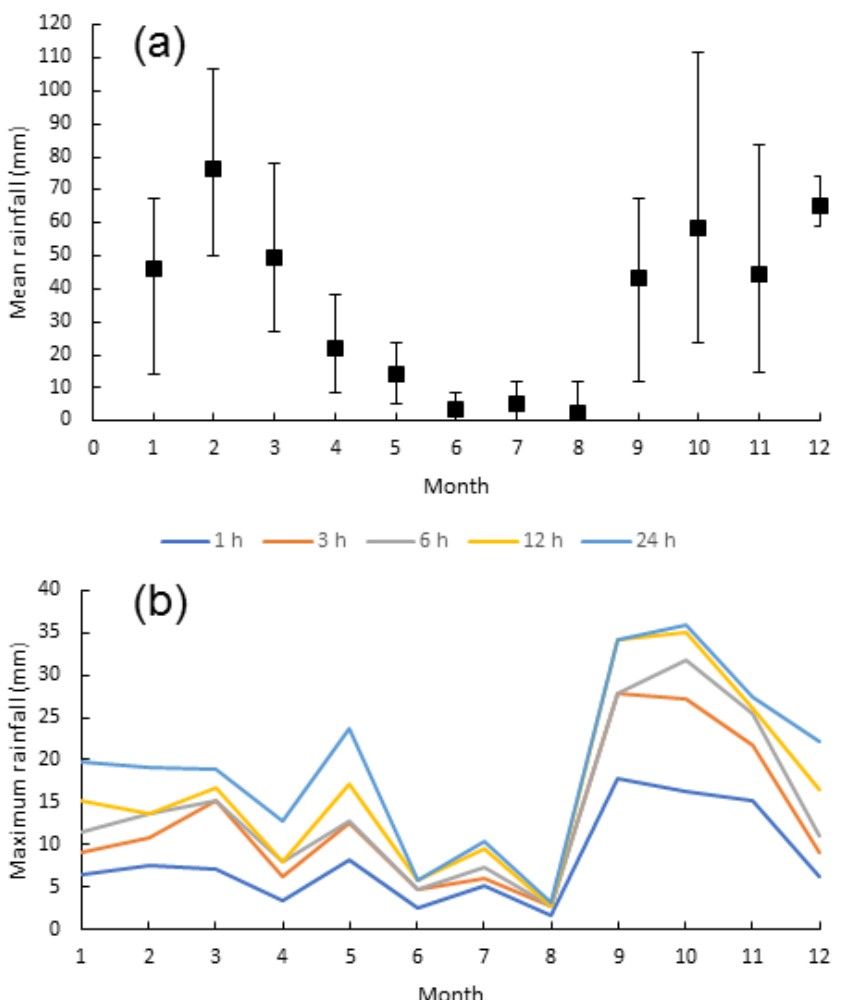

**Figure 2: a) Mean monthly rainfall (mm) observed at Lentia station (Fig. 1) between 2010 and 2014 (error bars also indicate minimum and maximum values); b) Maximum monthly rainfall within 3, 6, 12 and 24 hours (data from INGV Palermo).**

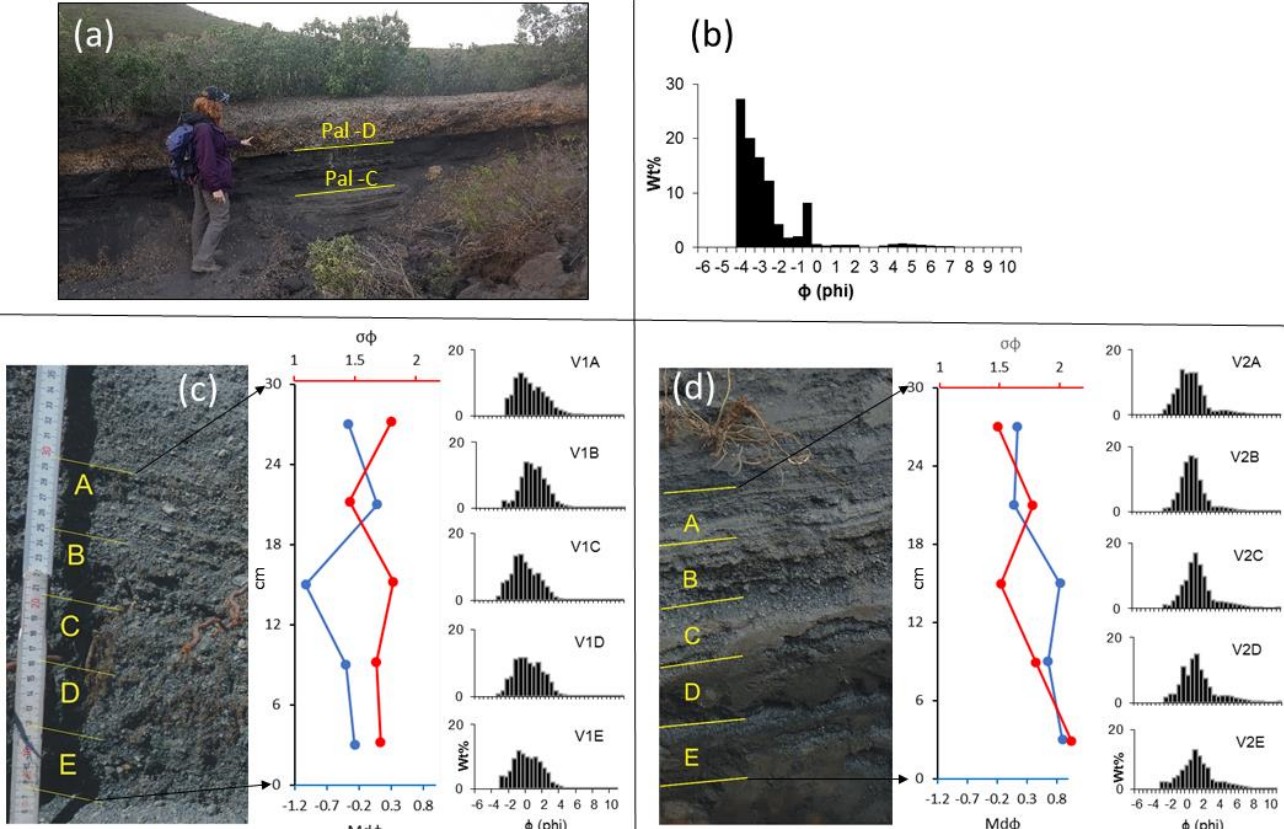

**Figure 3: a) Pal D subplinian tephra fallout deposit outcrop (V3, Fig.1) b) grainsize distribution Pal D subplinian tephra fallout deposit; c) and d) Mdφ, σφ and grainsize distribution of the first 30 cm of the tephra-fallout deposit associated with the 1888-90 Vulcanian eruption for stratigraphic sections V1 and V2 (Fig. 1) (thickness of layers analysed: A (24-30 cm); B (24-18 cm); C (18-12 cm); D (12-6cm); E (6-0 cm).**

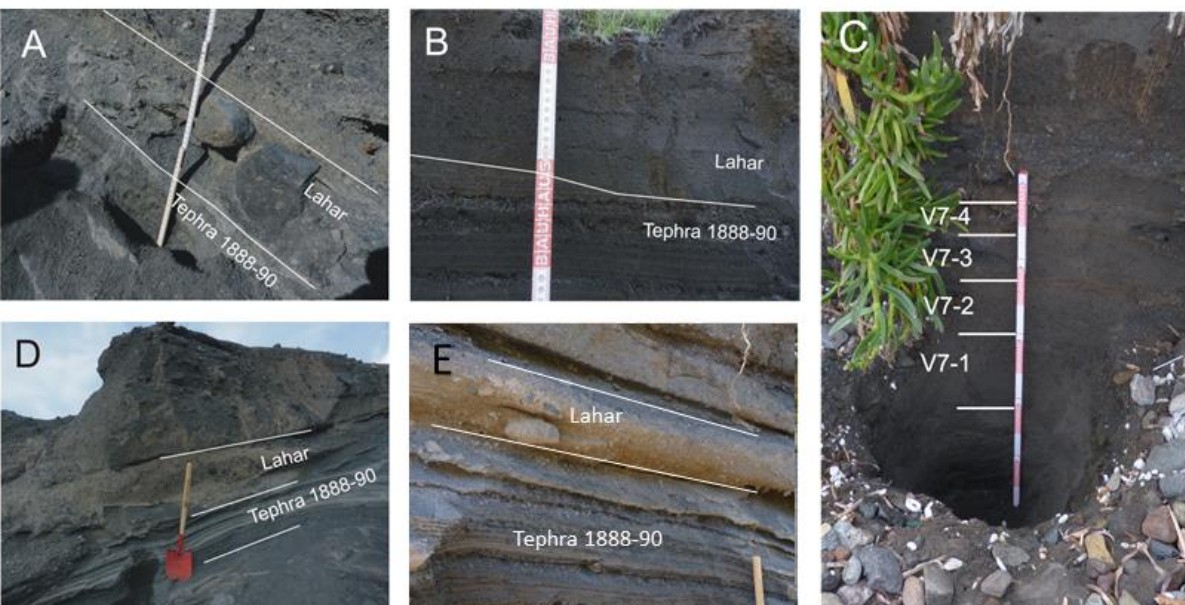

**Figure 4: A)** Lahar levee deposit (sample V5 in Fig. 1) above the 1888-90 tephra-fallout, located in a channel cut on the S of La Fossa cone. **B)** Lahar deposit (sample V6) above the 1888-90 tephra-fallout in the Palizzi valley. **C)** Profile observed at the beach side in Porto di Ponente (1) is a 14 cm bed of coarse ash, with a 1cm thick grey fine ash grey layer, this is the primary 1888-90 AD tephra-fallout; Layer V7-1 is a 26 cm lahar deposit of coarse ash to fine lapilli inversely graded; Layer V7-2 is a 11 cm fine lapilli lahar deposit with a 1 cm soil on the top; V7-3 is a 10 cm lahar deposit of coarse to fine ash with a soil on the top and 6 cm lahar deposit (V7-4) of coarse to fine ash with the recent soil on the top. **D)** Several lahar deposits located in a channel on the NW of La Fossa cone: the first deposit above 1888-90 was sampled (V9 in Fig. 1). **E)** Lahar deposits located in a channel on the NW of La Fossa cone on the top of the 1888-90 tephra-fallout deposit.

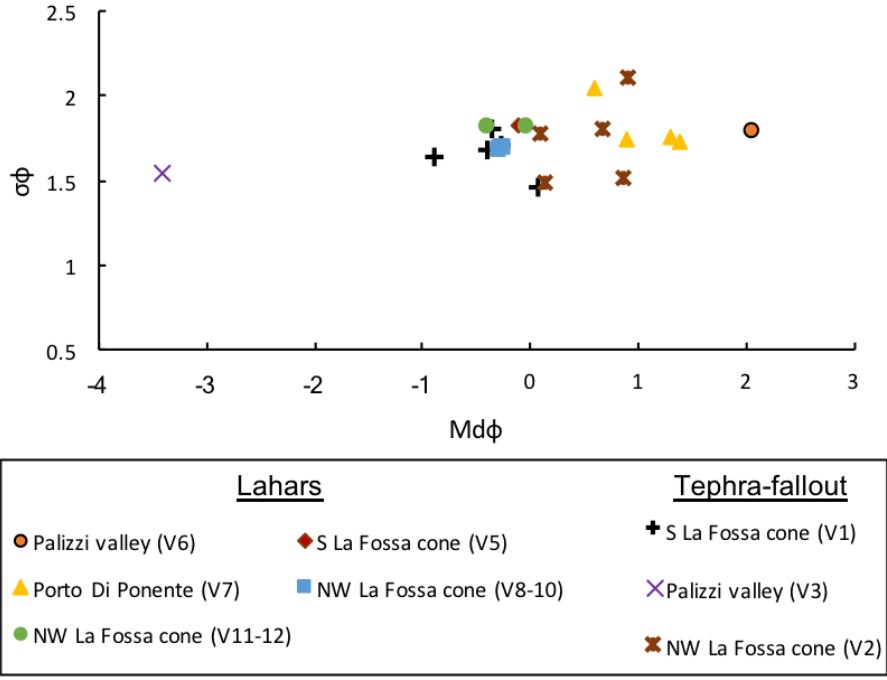

**Figure 5: Mdφ vs σφ diagram for the lahar matrix and tephra-fallout deposits. Porto Di Ponente (orange triangle) correspond to samples: V7-1, V7-2, V7-3, V7-4 S (Fig 4C); La Fossa cone V1 (black cross): V1A, V1B, V1C, V1D, V1E (Fig. 3C). NW La Fossa cone V2 (brown star): V2A, V2B, V2C, V2D, V2E (Fig. 3D).**

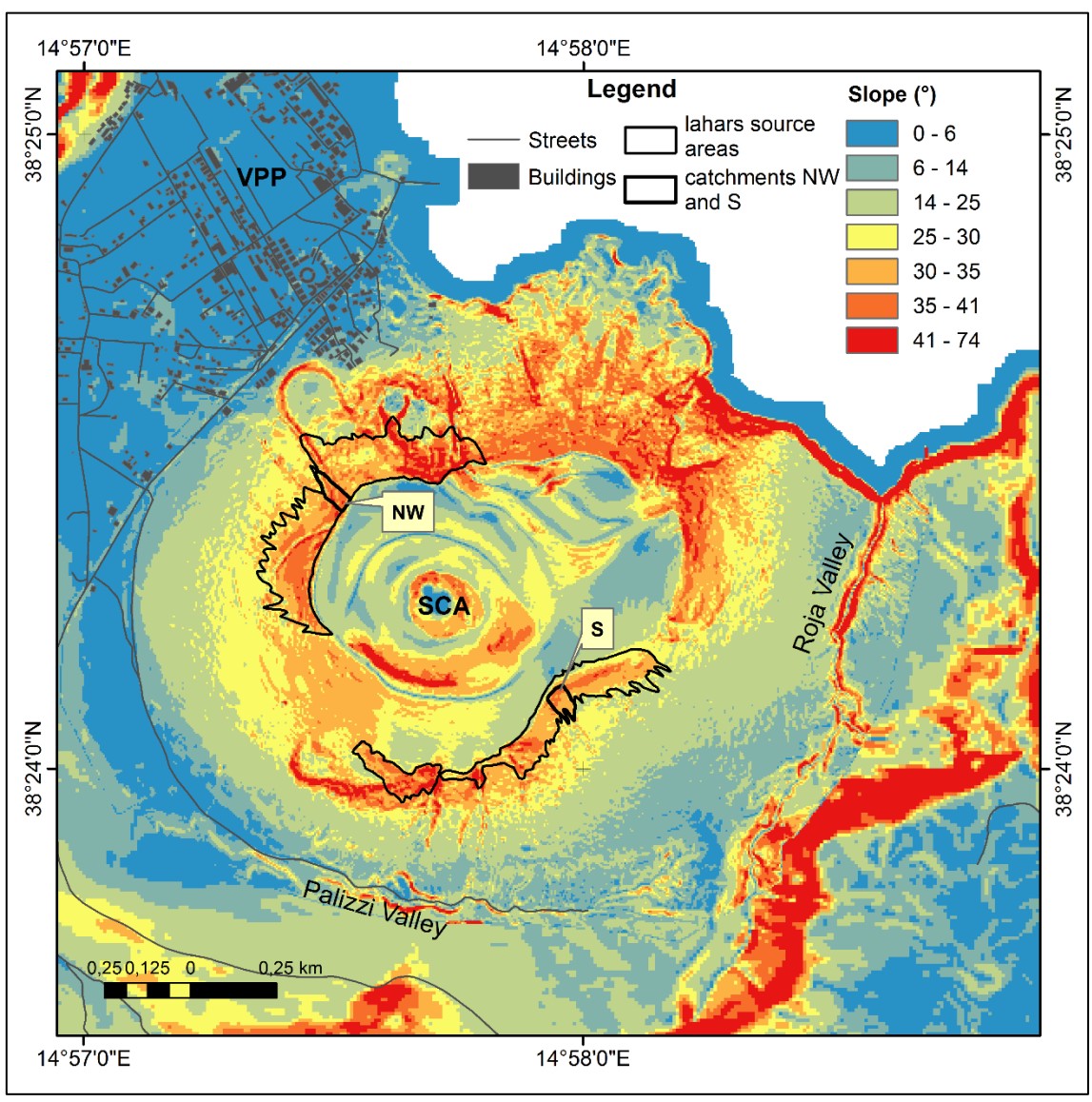

**Figure 6: Slope map for the La Fossa cone and surrounding areas. North West and South lahar source areas are indicated with a black contour. The NW and S upper catchment are indicated with a black contour.** *SCA*: **summit cone area;** *VPP*: **Vulcano Porto Plain**

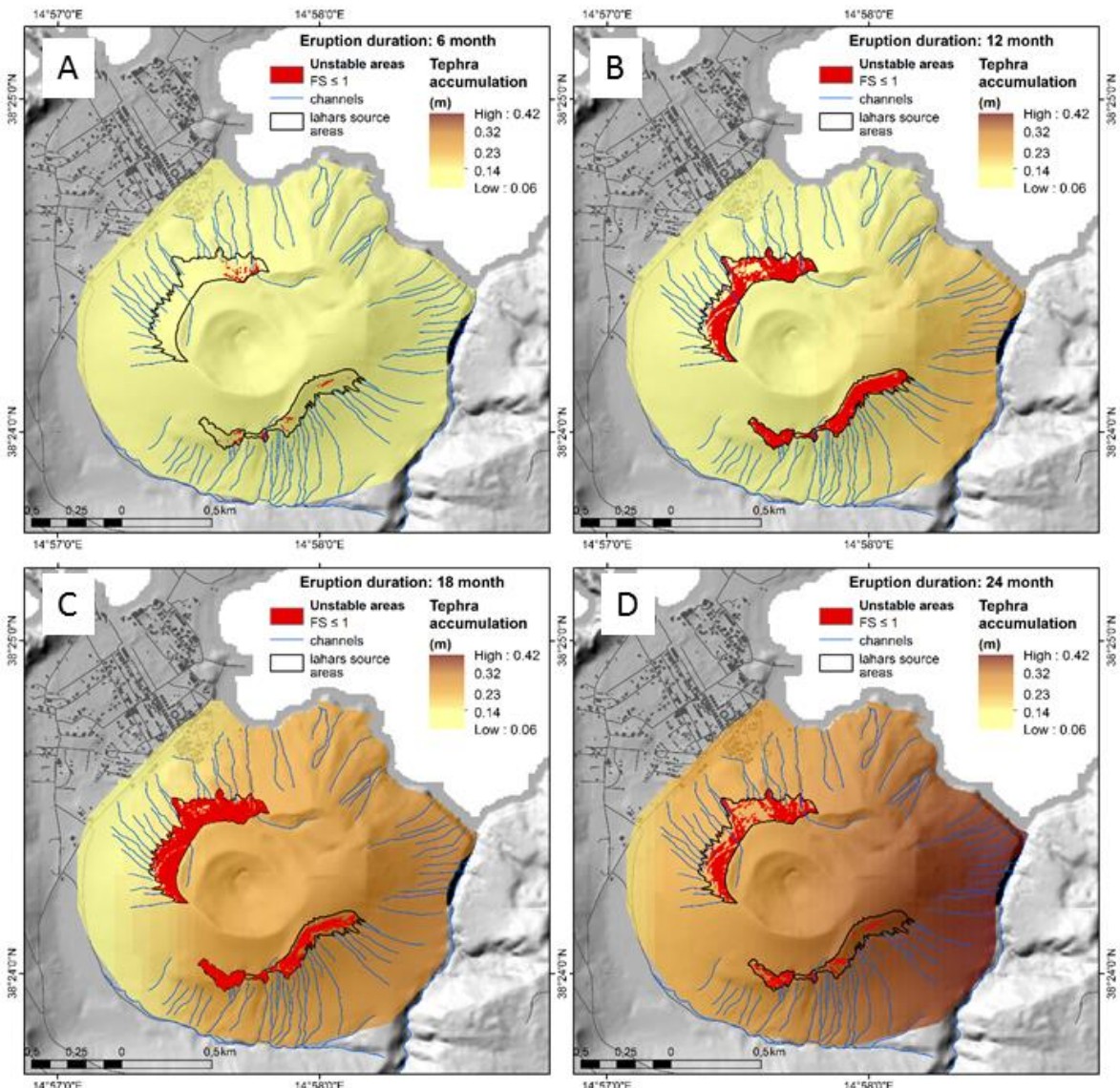

**Figure 7: Probabilistic isopach maps (converted from the probabilistic isomass maps of Biass et al. (2016) based on deposit density) and corresponding instability maps compiled with TRIGRS for a Vulcanian eruption with: A) an eruption duration of 6 months and a probability of occurrence of 25%; B) an eruption duration of 12 months and a probability of occurrence of 25%; C)) an eruption duration of 18 months and a probability of occurrence of 25%. D) an eruption duration of 24 months and a probability of occurrence of 25%. The rainfall intensity is 6.4 mm h$^{-1}$ with a duration of 5h for all the scenarios and the parameters for the 1888-90 Vulcanian deposits are listed in Table 3.**

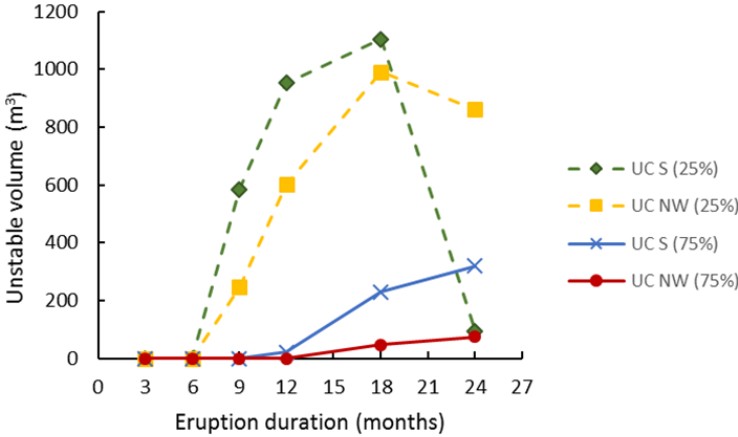

**Figure 8: Unstable tephra-fallout volume for the S and NW upper catchments obtained with TRIGRS for eruption durations of 3, 6, 9, 12, 18 and 24 and for probabilities of occurrence of 25% and 75%. UC: upper catchment (see Fig. 7b).**

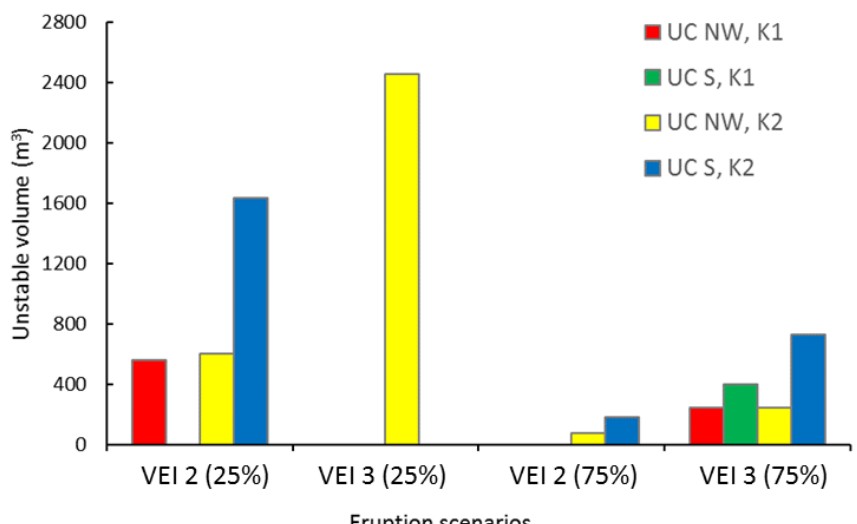

**Figure 9: Unstable tephra-fallout volume for the S and NW upper catchments obtained with TRIGRS for: 1) the subplinian scenarios VEI 2 and VEI 3 with a $K_s = 1 \times 10^{-2}$ m s$^{-1}$ (from literature) and VEI2, VEI3 with a $K_s = 6.8 \times 10^{-4}$ m s$^{-1}$ (as measured in the field) and the probabilities of occurrence 25% and 75%.**

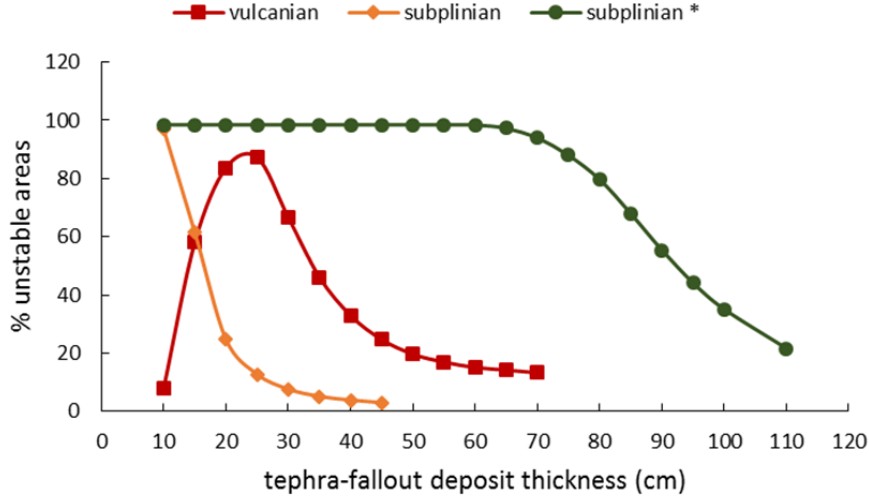

**Figure 10: Percentage of unstable area for the NW and S lahar source areas simulated with TRIGRS for tephra-fallout deposit thicknesses between 0.1-1.1 m and a rainfall intensity of 6.4 mm h$^{-1}$ with a duration of 5 hours and parameters for: Vulcanian**
10     **tephra-fallout deposits (red squares); subplinian tephra-fallout deposits with $K_s$ = 1x10$^{-2}$ m s$^{-1}$ (value from literature; orange diamonds) and subplinian tephra-fallout deposits with $K_s$= 6.8x10$^{-4}$ m s$^{-1}$ (value measured in the field; green circles).**

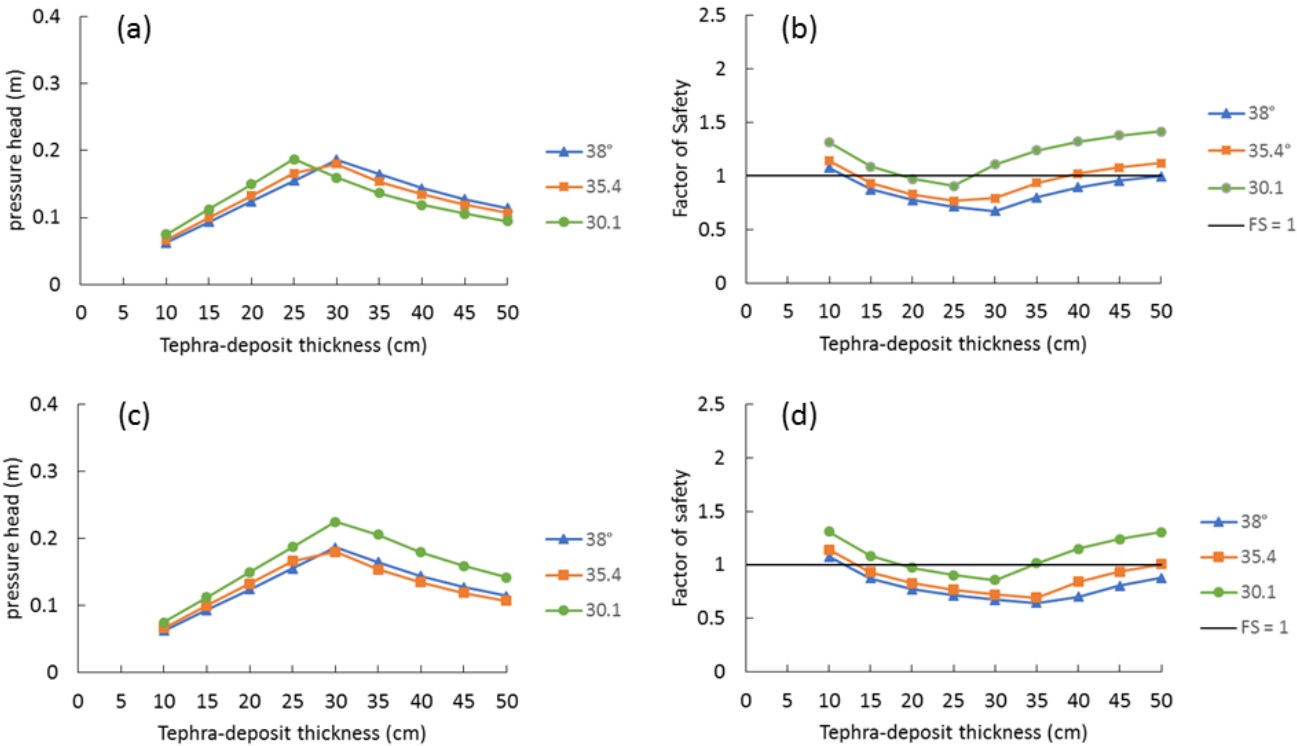

**Figure 11: Total pressure head and factor of safety versus tephra-fallout deposit thicknesses between 0.1 and 0.55 m for Vulcanian tephra-fallout deposits (Table 3) and a rainfall intensity of: A) and B) 6.4 mm h$^{-1}$ with a duration of 5 hours (I/K$_s$ = 0.02); C) and D) 15.5 mm h$^{-1}$ with a duration of 3 hours (I/K$_s$ = 0.05) (see also Table 6).**

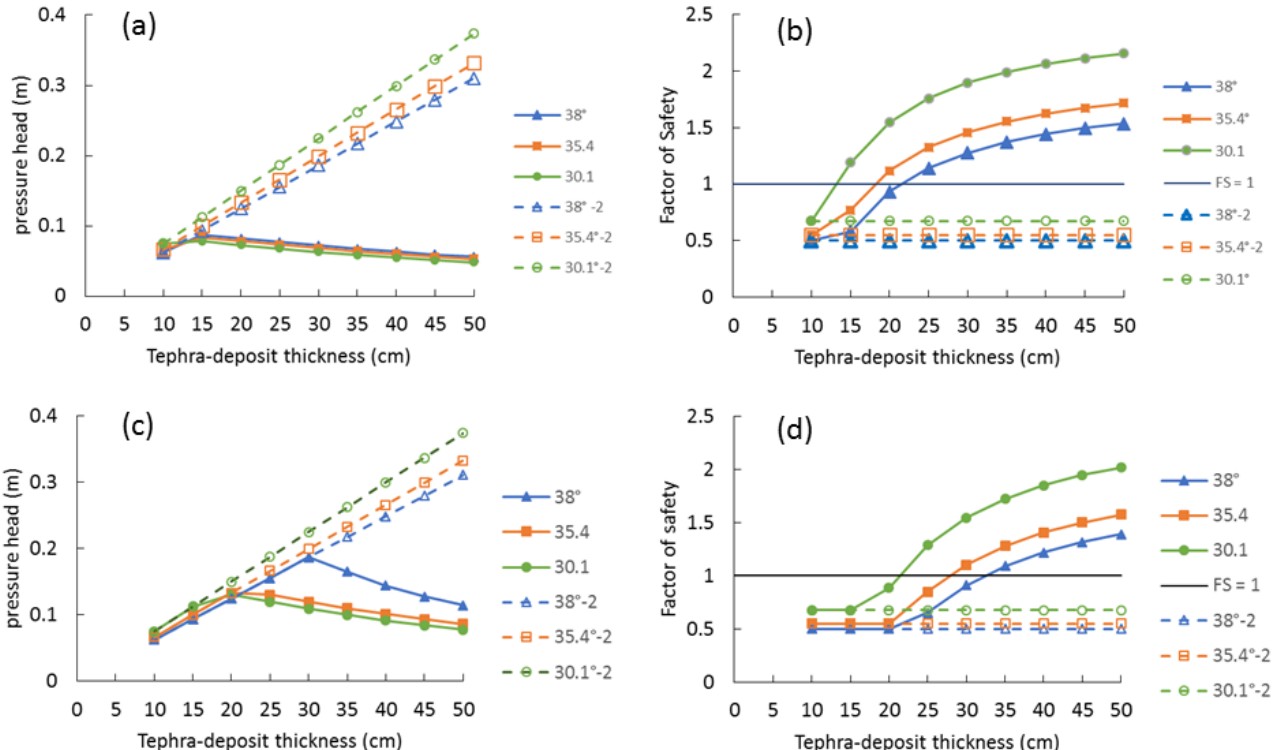

**Figure 12: Total pressure head and factor of safety versus tephra-fallout deposit thicknesses between 0.1 and 0.55 m for subplinian tephra-fallout deposits with $K_s = 1 \times 10^{-2}$ m s$^{-1}$ and $K_s = 6.8 \times 10^{-4}$ m s$^{-1}$ (dashed lines) for two different rainfall intensities and durations: A and B) 6.4 mm h$^{-1}$ with a duration of 5 hours. C) and D) 15.5 mm h$^{-1}$ with a duration of 3 hours (see also Table 6).**

