# Peer review of "Mapping the susceptibility of rain-triggered lahars at Vulcano island (Italy) combining field characterization, geotechnical analysis and numerical modelling"

_Natural Hazards and Earth System Sciences, 2019_

## Referee Comment (RC1) · Anonymous Referee #1 · 7 Jul 2019

**General comments**

In this study, the authors investigate lahar initiation at La Fossa Cone, Vulcano. Field studies are used to estimate permeability and soil-suction of 'primary' (i.e. tephra fall) deposits, with samples of both Tephra fall and lahar deposits collected for laboratory analysis. Both the field and laboratory data were used to constrain properties for an analysis of cascading hazard (tephra fall -> lahar) using two models (of tephra fall and slope stability).

Unstable areas at La Fossa are identifies using two scenarios, a vulcanian eruption scenario and subplinian scenario. The volume of potentially unstable tephra deposits in two catchments are identified through a shallow landslide slope stability model, and clarify critical tephra fall depths that may result in lahar initiation, for a limited range of rainfall scenarios.

The scientific methods used in this study are well established in previous work and are appropriately detailed and applied here, with some issues expanded on in 'Specific comments' section. However, while the manuscript presents a wealth of data and results, the scientific significance of this work is unclear. Interpretation of the results is limited to surface-level comparisons with other examples of lahar generation. This leads to conclusions that simply repeat results and don't appear substantial to the reader. The final conclusion ("..a comprehensive assessment of unstable volumes that could potentially trigger lahars ... requires dedicated numerical simulations combined with detailed field observations and geotechnical analysis"), is relevant and valuable, but is not supported by the manuscript's presentation, text or discussion.

To be acceptable for publication, I suggest the manuscript needs to be restructured in a way that (a) clearly identifies the importance and objectives of this study, (b) concisely presents data relevant to support the study, and (c) critically highlights results of importance and explains their significance to the reader. Some more detailed comments to be considered in the restructuring is highlighted in the following "specific comments" section.

**Specific comments**

Page 2, line 18: It is important to clarify that overland erosion and shallow landsliding is the most common mechanism (see e.g. Pierson and Major 2014), as opposed to the only two mechanisms for generating lahars.

Page 3, line 19-20: "Nonetheless, the use of physically-based models ... is necessary ...". I think this is intended to be the crucial aim/thrust of the manuscript, but is not justified from the text before it. What are the weaknesses in an approach such as Tierz et al.? Why is coupling a physical model preferable/necessary? Aspects of Volentik et al. (2009) and Galderisi et al. (2013) might help explain the benefits of such an approach.

Page 3, line 33 and on: "This approach provides the first integrated attempt ..." to page 4, line 6 "offers an innovative treatment of cascading effects". Both Galderisi et al. (2013) and Volentik et al. (2009) used a similar approach that integrated tephra fall models with a slope stability/shallow landslide model, so I don't necessarily agree with the statements in this paragraph. In combination with the previous comment, I think coupled approaches need to be discussed in more detail to highlight the significance of this work (in my opinion: better considers impact of rainfall on stability, identifies thresholds for initiation and provides fully quantitative estimates of lahar initiation).

**Field and laboratory methods and results** The field sampling is unclear to me in section 3.1. What is meant by collecting 8 samples "...to retain primary physical characteristics"? Is this where in-situ permeability/soil suction tests were done? On page 5, line 2 - A further 11 samples are collected, looking at Fig. 3 I presume from the Pal D deposit (1 sample), and then at two locations for the 1880-90 vulcanian eruption. This corresponds to samples V1, V2 and V3. However, line 3-7 then discuss location of samples (4 from S La Fossa, 2 from NW La Fossa, 2 from Palizzi valley), but that only sums to 8 samples.

Aided by confusion in sampling, the first two results sections (4.1, 4.2) are difficult to follow, presenting a large amount of information not directly relevant to the manuscript. The purpose of field and laboratory characterization is to identify parameters of Pal D and 1880-90 deposits for TRIGRS. While grainsize is a consideration for lahar generation and tephra fall, the authors do not use any of this information in their study. Tephra simulations use Biass et al. 2016 results, and no comparison between the chosen size

range in Biass et al. 2016 and the values found in this study. The sample naming scheme is hard to follow in context, switching from unit to location to site to unit in the tables, and unit to site to sample number in the figures.

I suggest simplifying both the description of sampling and presentation of results from field and laboratory analyses. In this study, we are interested in the Pal-D and 1880-90 tephra and lahar units. Individual data from each location (Tables 1-3) can be provided as supplementary material, and results should focus on how Geotechnical (table 4) and input parameters (table 5) are derived from your sampling campaign. I fail to see how the extensive study on grainsize is necessary here, beyond a few sentences, and would recommend shifting figures 5 and 6 to supplementary material.

Page 12, line 33: Were two upper catchments exactly the same size, or were they of similar size? How were catchment boundaries defined (i.e. from the slope or drainage networks/external data)?

Figure 9: It is better to make this figure greyscale compatible and easier to interpret. I would suggest something like using dashed lines for 25

Section 4.3.1 - This section seems to show that increasing the tephra thickness above a certain threshold will increase stability, nicely leading onto section 4.3.2. Figure 10 doesn't seem to add much to this discussion over table 7, so I would recommend removing it.

Page 14, line 2-4: It is unclear how deposit thickness was increased. Was this assuming a constant depth of deposit across the entire NW and S area, or was it applied as a proportion of the isopachs (either Tephra2 or observed)?

Page 15, line 1 and on: "... total pressure head has a higher maximum displaced to higher tephra fallout deposit thickness..." What does maximum displaced mean here?

Page 15, line 14 - page 16, line 2 : I do not understand the relevance to Cordon Caulle in the entire section, and it seems misguided. The friction angle of a granular material

is controlled by the distribution ($\sigma$) of grainsize, asperity and roughness; not mean grain size. Hydralic conductivity for these two different eruptions would be expected to differ, as the Pal D deposits are much coarser than Cordon Caulle lapilli. Some 'washing' of fines over time may occur, but if differences in measuring techniques cause a 2 orders of magnitude difference in conductivity, then the techniques are unreliable. This section is better served by starting with Page 16, line 3 (Table 8 ...).

Another consideration in section 5.1 is the volume of lahars. Lahar volumes for all the other examples in table 8 are quite large, in comparison to the smaller lahars at Vulcano.

Page 20, lines 6 - 8: How was the assessment of unstable areas found to be accurate? Without validation against a specific event (or set of events), a *methodology* has been shown to identify unstable areas.

**Technical corrections**

Note, given the need for extensive changes, I have not comprehensively identified minor technical corrections (some in the attached PDF), I would recommend a detailed proof read to identify further grammatical and typographic issues.

Page 2, line 13: A reference is probably necessary here.

Page 10, line 23 - 24: For such a small lahar, the errors in measurement with a hand-held GPS and thickness estimates would be significant. I suggest removing the sentence "The area of the front lobe ..." and simply explain two samples were taken from a recent lahar.

20   on the NW volcano flanks, in the Palizzi valley and in the Porto plain (Fig. 1). We described also a recent lahar deposit occurred in September 2017 on the La Fossa cone NW flank. Lahar matrix samples from these lahars were collected on the volcano flanks and ring plain for grainsize analyses.

**3.2 Laboratory analyses**

[revised manuscript text omitted]

**4 Results**

**4.1 Field characterization of tephra and lahar deposits**

*Pal D deposit*

We logged two sections of the Pal D primary deposit at outcrops located in the Palizzi Valley (point V3 and V4, Fig. 1). The

20  isopach map (Di Traglia, 2011) in Fig. 1b show the southward dispersal of Pal D primary deposit distribution on Vulcano island. The section is a 1 m-thick, massive, grain supported and well-sorted pumice deposit (Fig. 3 A). The Pal D deposit occurs between two sub-units of the Palizzi cycle. At the base, Pal C deposit (alternation of black lapilli and ash) crops out, while on top of Pal D the rhyolitic white ash of Rocche Rosse eruption from Lipari and the Breccia di Commenda are present (Di Traglia et al., 2011; Rosi et al., 2018). The mean saturated hydraulic conductivity of the Pal D deposit measured in the

25  field is $6.8 \times 10^{-4}\,\mathrm{m\,s^{-1}}$ (Table 1).

*1888-90 deposit*

Two stratigraphic sections of the 1888-90 primary deposit were logged in the upper part of the La Fossa cone S flank (V1; Fig. 3 C) and at the base of the NW flank (V2; Fig. 3 D). The isopach map for 1888-90 primary deposit shows its almost

30  circular dispersal (Fig. 1 c). The V1 section overlies several older tephra fallout and lahar units (Fig. 1). It is a 1 m-thick deposit consisting of an alternation of thin ash and lapilli layers overtopped by 0.2 m of reworked tephra. The whole sequence shows an inclination of 30°. The second section (V2) is a 0.5 m-thick deposit laying on the Commenda tephra

sequence and is overlaid by 0.3 m of reworked tephra. Seven soil suction measurements on the 1888-90 deposits located on the La Fossa cone in the upper catchment and lower part of the cone (Fig. 1) are comprised between 15 kPa and 27 kPa (Table 2). The mean saturated hydraulic conductivity measured in the field varies between 6.0 and $7.5 \times 10^{-5}$ m s$^{-1}$ (Table 1).

5 *Syn-eruptive lahars (1888-90 eruption)*

Here, we use *syn-eruptive* lahars to describe events that occurred during and shortly after the 1888-1990 eruption cycle. Stratigraphic sections in gullies and small channels on the NW flank of the la Fossa cone show several massive to laminated, remobilized deposits covering the 1888-90 primary tephra. Assessing the areal distribution and the correlation between different lahar deposits is difficult since no lahar map on Vulcano exist. The thickness of each lahar layer, representing

10 different flow pulses, varies between ~ 0.20 and 1 m. The lahars are massive to thinly laminated, matrix-supported deposit, with boulders immersed in a coarse ash and lapilli matrix. The observed boulders have diameter between 5 and 15 cm. Although the distinction among debris flow and hyperconcentrated flow emplacement mechanisms from the direct observation in the field was not possible due to the lack of information related to original water content, we speculate that most of the well-sorted, massive deposits were related to hyperconcentrated flows based on the definition of Pierson (2005).

15 11 samples (total number) of lahar matrices were sampled on the S cone flank (V5; Fig. 4a), on the NW cone flank (V8-V10; Fig 4d), in the Palizzi valley (V6; Fig 4b) and in the Porto Plain (V7-1-4; Fig. 4f), where the matrices of four lahars were sampled from distinctive episodes.

*September 2017 lahar*

20 We also described a small lahar event that occurred in September 2017, one month before our 2017 field work (Fig. 4e). The lahar source area was located on the NW cone flank in a funnel shaped area located above a small gully at an elevation of 159 m a.s.l. The lahar flowed into a gully with an average width of 2 meter and a depth varying between 0.4 and 0.8 m, formed levees on both sides and stopped on the La Fossa crater trail with a final runout of 120 m. The area of the front lobe deposit was measured with a handheld GPS (135.5 m$^2$) and approximate thickness estimated (0.3 m) in the field, which

25 resulted in a volume of ~40 m$^3$. A second lahar flow pulse deposited a small deposit confined within the channel (Fig. 5e). Two samples (V11 and V12) were taken from this recent lahar deposit in order to compare with the older syn-eruptive lahar deposits.

**4.2 Laboratory characterization of tephra and lahar deposits**

*Grain-size analyses*

30 Grain-size distributions of all samples are summarized in Table 3 and Figure 5. The Mdφ of the majority of samples is in the range of 2φ and -1φ and most deposits are well sorted. Grain-size distribution of primary deposits are presented in Figure 6a. The grain-size distribution of the top 30 cm of the Pal D primary deposit at section V3 shows Mdφ and σφ of -3.42 and 1.55, respectively (Table 3). Grain-size distributions of the top 30 cm of the 1888-90 primary deposit on the S flank of La Fossa

(V1) show Mdϕ of -0.88–0.08 and σϕ of 1.46–1.80, respectively (Table 3). At the base of the cone on the NW sector (V2), grain-size distributions are slightly finer (Mdϕ of 0.09 to 0.9) and with a poorer sorting (σϕ of 1.49 to 2.1). Generally, the grain-size distributions of the top 30 cm of sections V1 and V2 show a predominance of coarse ash (Fig. 3) but V2, located at the base of the cone, contains more fine ash. Grain-size distribution of lahar matrices are presented in Figure 6b. All

5    matrix samples have a low content of fine ash (i.e., 2.4–16%). The matrix is composed primarily of 60–83% coarse ash and 2–36% of lapilli. The Mdϕ vs σϕ diagram shows a finer grain-size distribution for lahar matrices located in the Palizzi valley and Porto di Ponente harbour (V6 and V7 samples, respectively, Fig. 5) than for those located on the La Fossa cone (V5 sample, Fig. 5). The grain-size distribution for older (V8–10 samples, Fig. 5) and more recent (V11–12 samples, Fig. 5) lahars are in the same range. In general, the grain-size distributions of the 1888-90 primary deposits is similar to their

10   remobilized counterparts (Figs. 5, 6; Table 3). Only three primary deposits are coarser than lahar matrices (V1C, V1D and V3; Fig. 5)

*Natural water content*

The natural water content ($w_n$) of the 1888-90 primary deposit varies from 2.64% to 3.65%, while Pal D primary deposit

15   exhibit higher $w_n$ (10.80–30.63%). The specific gravity ($G_s$) for the solid fraction was measured for both deposits and shows values of 2.57 and 2.42 for the 1888-90 and Pal D primary deposits, respectively.

*Shear strength*

Although the four specimens of 1888-90 primary deposit are consolidated at three different total stresses, all specimens

20   exhibit a slight hardening associated to a dilative behaviour. The shear stress envelope exhibit high friction angle ($\phi = 42°$) and nil cohesion (Table 4). For Pal D primary deposits, direct shear tests are performed on reconstructed specimens constituted only by sandy fraction, smaller than 20 mm. The specimen is reconstructed using air pluviation method into the shear box, i.e. pouring the dry deposit material with a spoon from nil drop height. The specimens exhibit high porosity (n equals about 0.72) closer than in-situ porosity. The tests are performed in dry condition and all specimens exhibited a

25   dilative behaviour. The friction angle at peak is high, and typical of lapilli clasts ($\phi = 54°$), while the cohesion is null (Table 4). The angle of dilatation ($\Psi$) is also evaluated and is about 13°. Thus, according to Taylor (1948) the friction angle is about 41°, but it will be reached at large deformation.

*Hydraulic conductivity*

30   The mean saturated hydraulic conductivity of the 1888-90 primary deposit measured in the laboratory ($8.50\times10^{-5}$ m s$^{-1}$) is similar to that obtained during field measurements (6.0 and $7.5\times10^{-5}$ m s$^{-1}$). The mean hydraulic saturated conductivity of the Pal D primary deposit could not be measured in the laboratory because the deposit grainsize (Mdφ = -3.4φ) is too coarse for the apparatus measuring hydraulic conductivity; instead we use here both values one from the literature for other coarse-grained volcanic soils and the second the value obtained through field measurements. A value of $1 \times 10^{-2}$ m s$^{-1}$ (Table 4) is

inferred from the hydraulic conductivity measured in the field on 2011 Cordon Caulle (Chile) eruption lapilli deposits (Baumann et al., 2018) and from the lower pumice deposit from Vesuvius (Crosta and Dal Negro, 2003). These values show a large discrepancy with our field measurements of $6.8 \times 10^{-4}$ m s$^{-1}$. These two end-members are hereafter referred to as *high* (i.e. $1 \times 10^{-2}$ m s$^{-1}$) and *low* (i.e. $6.8 \times 10^{-4}$ m s$^{-1}$) hydraulic conductivities.

*Soil Water Retention Curve*

The SWRC of the 1888-90 primary deposit exhibit air entry value equal to 1 kPa, while the water content at saturation ($\theta_s$) is 0.47 and the residual water content ($\theta_r$) is 0.04. The SWRC of Pal D deposit exhibit air entry value lower than 1 kPa and slightly high water content at saturation ($\theta_s = 0.72$) and low residual water content ($\theta_r = 0.03$). The data are interpolated using
10  both Gardner (1958) and Brooks and Corey (1962, 1964) equations (Table 4).

*Saturated soil diffusivity*

The saturated soil diffusivity of the 1888-90 and Pal D primary deposits are one order of magnitude higher than their saturated soil conductivity (Table 4). The 1888-90 primary deposit shows a value of $3.28 \times 10^{-4}$ m$^2$ s$^{-1}$ the Pal D primary
15  deposit a value of $6.59 \times 10^{-3}$ m$^2$ s$^{-1}$.

**4.3 Modelling**

Based on considerations in Section 2.2, TRIGRS simulation are run using only one very-high intensity rainfall scenario of 6.4 mm h$^{-1}$ over 5 hours (i.e., total of 32 mm). Such intensities have occurred twice in 2011 and have been witnessed to cause widespread floods in the Porto flood-plain. This scenario should be considered as a very high/torrential scenario used
20  to investigate the maximum unstable tephra volume. For the Pal D primary deposit, we used two different hydraulic conductivities in order to consider both end-members as described in laboratory analyses (Table 5).

[revised manuscript text omitted]

We have also plotted the total pore pressure and the lowest FS at the end of the rainfall event as a function of deposit thickness for three different slope angles (38°, 35.4° and 30.1°). Fig.12 presents results for the 1888-90 eruption primary deposit and results for the Pal D primary deposit, for both values of $K_s$ are seen in Fig. 13. We simulated two different rainfall events: a rainfall intensity of 6.4 mm h$^{-1}$ with a duration of 5 hours (total rainfall of 32 mm) and a rainfall intensity of

15   15.5 mm h$^{-1}$ with a duration of 3 hours (rainfall event recorded at Lentia station the 11 November 2017, with a total of 46.5 mm). The total pressure head for the 1888-90 eruption deposit has a maximum at 0.25 m for a 30.1° slope angle and a maximum of 0.3 m for a 38° and 35.4° slope angle (Fig 12a). We also show that there is an unstable window (FS < 1) for thicknesses between 12 cm and 50 cm with a slope of 38°, which with lower slopes is narrower; in the case of a slope of 30.1° the deposit is unstable for thicknesses between 20 and 27 cm (Fig. 12b). We also explored the pressure head and FS for

20   a higher rainfall intensity (I = 15.5 mm h$^{-1}$), representing the worst rainfall scenario for 2017 (Fig. 12c, d). We observed that the unstable window is larger for the three slope angles, with an upper thickness limit of 35 cm, 50 cm and more than 50 cm for the slope angles 30.1°, 35.4° and 38°, respectively. For the same tephra-fallout deposit the ratio of rainfall intensity and hydraulic conductivity (I/$K_s$) determines the time to reach the water table (located at the bottom of the deposit in our case study); the rate of rise of water table increases with an increase in I/$K_s$ ratio (Li et al., 2013). In the case of the 1888-90

25   tephra-fallout deposit, the upper critical thickness for instabilities increases with the increase of rainfall intensity and total rainfall.

For the Pal D primary deposit with high hydraulic conductivity, the pressure head increases with decreasing thickness and reaches a maximum of 0.07 m, 0.08 m and 0.087 m for a tephra thickness of 15 cm (Fig. 13a). In contrast, the total pressure head shows a monotonic increase with increasing deposit thickness for low hydraulic conductivity. In the case of the safety

30   factor for high hydraulic conductivity, only very shallow deposits are unstable and for slope angles of 38° the tephra-fallout deposit with a thickness larger than 20 m is stable. In contrast, the FS is constant in the case of low hydraulic conductivity and is below 1 for deposit thicknesses between 10 cm and 55 cm (Fig. 13b).

For a rainfall intensity of 15.5 mm h$^{-1}$ with a duration of 3 hour (Fig. 13c, d) and with a high hydraulic conductivity, the total

pressure head has a higher maximum displaced to higher tephra-fallout deposit thickness and the deposit thickness limit between stable and unstable area is higher (almost 32 cm for a slope angle of 38°) compared with the 6.4 mm h$^{-1}$ rainfall intensity (Fig. 13a, b). In contrast, in the case of the low hydraulic conductivity the total pressure head and the safety factor are the same as the results obtained for the lower rainfall intensity (6.4 mm h$^{-1}$). The constant factor of safety is the result of

5  the upper boundary of pressure head, which is physically limited at the beta-line (Iverson, 2000; Baum et. al., 2008). The total pore pressure cannot be above the values denoted by a water table at the ground surface (beta-line) and the model calculates the factor of safety with this value, which is the worst condition for instability.

**5. Discussion**

**5.1 Characteristics of lahar source deposits**

10  Grain-size distribution of pyroclastic material is one of the primary factors influencing the type of erosion or failure mechanism in the case of rainfall-triggered lahars (e.g., Manville et al., 2000, Pierson et al., 2013). In this study, critical characteristics of tephra-fallout deposits (i.e. grain-size, hydraulic conductivity and angle of friction) necessary as input for shallow landsliding process modelled with TRIGRS have been analysed for two different eruption scenarios (Vulcanian and subplinian). Overall, the tephra fallout deposits associated with both the 1888-90 Vulcanian eruptions and the subplinian Pal

15  D eruption are relatively internally homogeneous in terms of grain-size and geotechnical properties (Tables 1–5). In contrast, the tephra-fallout sequence associated with the climactic phase of the 2011 Cordón Caulle eruptions (Chile) that also generated post-eruption lahars is characterized by two different layers with contrasting grain-size: lapilli at the base (Units I and II) and ash on the top (Unit III) (Pistolesi et al., 2015). In terms of friction angle, Pal D deposits and lapilli layers I and II of Cordón Caulle eruption are very similar, 54° and 53°, respectively. However, the hydraulic conductivities measured in the

20  field are very different ($K_s$= 3.9x10$^{-2}$ m s$^{-1}$ and $K_s$= 6.8x10$^{-4}$ m s$^{-1}$, respectively). This significant difference in hydraulic conductivities can be due partly to the age and the geological setting of the primary deposits and partly to the use of different measuring techniques. In the case of 2011 Cordón Caulle lapilli (Units I and II) the hydraulic conductivity was measured by filling a plastic tube with an undisturbed sample and saturating the materiel with water. Instead, measurements for the Pal D primary deposit was performed on the outcrop with a single ring permeameter (see Appendix A). Besides, the 2011 Cordón

25  Caulle deposit had only a small layer (10 cm) on the top and 0.7% of fine ash. The Pal D deposit is older (AD 1200), with 7% fine ash and overtopped by almost two meters of younger pyroclastic deposits. The lower hydraulic conductivity of the Pal D deposit compared to the 2011 Cordón Caulle lapilli layers could be due to the migration of small particles from the top and a compaction due to the load of the deposits on the top, which reduced the porosity. In fact, the dry unit weight of the Pal D undisturbed sample and the same a reconstructed sample (without compaction) is 6.63 kN m$^{-3}$ (Table 4) and 6.18 kN m$^{-3}$

30  respectively, which means a greater porosity (6.7%) for the reconstructed sample. Although the 1888-90 primary deposits have a slightly greater friction angle than the ash layers (Unit III) from the 2011 Cordón Caulle eruption (41° and 38.4°, respectively), the hydraulic conductivities are in the same range. The 1888-90 primary deposit also has similar friction

angles and hydraulic conductivities with respect to the pyroclastic deposits (ash soil class B) from Vesuvius (Cascini et al., 2010).

Table 8 shows a detailed comparison of grainsize, hydraulic conductivity and infiltration capacity associated with various tephra-fallout and PDC deposits that have generated rain-triggered lahars; annual precipitation for the associated region is
5   also reported. It is important to note that infiltration capacity and hydraulic conductivity can be considered as similar parameters for the sake of this comparison. In fact, the infiltration capacity is a measure of the rate at which soil is able to absorb water (Horton, 1945), while the hydraulic conductivity measures the ease with which water will pass through a porous media (Darcy, 1856). Infiltration capacity typically decreases through time and converges to a constant value, which is the hydraulic conductivity. Infiltration capacity is more easily measured in the field, while hydraulic conductivity is more
10  easily measured in the laboratory.

Examples of lahar generation enhanced by fine-grained deposits include Mt. St Helen 1980 (Leavsley et al., 1989), Chaitén 2008 (Pierson et al., 2013) and Cordon Caulle 2011 (Pistolesi et al., 2015) (Table 8). In contrast, the grainsize of the 1888-90 eruption deposit is closer to Mt. Unzen PDC and Pinatubo tephra. Hydraulic conductivity associated with the 1888-90 eruption deposit is 44 times higher than the infiltration capacity of the PDC in Shultz Creek (Mt. St. Helen). Infiltration
15  capacity is low also in the case of Mt. Unzen, but is higher for the PDCs of Mt. Pinatubo. An important difference between the deposits studied in this paper and the other deposits is the climatic conditions (Table 8): semi-arid, poorly vegetated regions with non-permanent streams and limited annual rainfall (500 mm), such as in the case of Vulcano, versus forested areas with permanent streams draining the volcano flanks and annual precipitation between 1000 mm and until 4300 mm for all other cases.

20  Grain-size of tephra-fallout deposits directly affect the hydraulic conductivity and infiltration capacity of the deposits. The highest hydraulic conductivities ($1 \times 10^{-2}$) for Md$\phi$ < -1$\phi$ (lapilli) and 
[revised manuscript text omitted]

5   D eruption as the reference event). The great novelty of this work is the assessment of compounding hazards (tephra-fallout deposits and lahar triggering) based on both numerical modelling and field characterization. We demonstrate that an accurate assessment of unstable areas can only be obtained based on a combination of dedicated numerical simulations and detailed field and geotechnical studies. In fact, volumes of tephra-fallout deposit that could be remobilized by rain-triggered lahars were analyzed using a slope stability model (TRIGRS) in combination with field observations and geotechnical tests. In

10   particular, we have considered 12 probabilistic isopach maps with different eruption duration and probabilities of occurrence of 25% and 75% in the case of the Vulcanian eruptive scenario. We have also considered 4 probabilistic isopach maps with two different VEI (2 and 3) and the same probabilities of occurrence in the case of subplinian eruptive scenario. In addition, a parametric analysis was performed with TRIGRS to determine the tephra fallout thickness thresholds required to trigger lahars for a given rainfall event. Two basins of same area were identified on the NW and S flank of the volcano to analyse

15   the effect of different morphology and of different accumulation related to the prevailing wind direction. The results of unstable volumes for the two basins located show that:

1) for the Vulcanian scenario, the largest unstable volume is reached for an eruption duration of 18 months and a 25% probability of occurrence scenario, with a volume of 1,105 m3 for the S basin and 990 m3 for the NW basin;

2) for the subplinian scenario, the largest unstable volume (2,455 m3) resulted for the VEI3 (25% of occurrence) with

20   a Ks= 6.8 × 10-4 m s-1 (i.e. hydraulic conductivity measured in the field) in the case of the NW basin;

3) for the subplinian scenario with Ks= 1 × 10-2 m s-1 (i.e. hydraulic conductivity estimated from literature) the largest unstable volume (563 m3) was found for the NW basin with a scenario VEI2 (25% of occurrence).

The parametric analysis with variable tephra-fallout thickness and slope and two rainfall events of 6.4 mm h$^{-1}$ for 5 hours and 15. 5mm h-1 for 3 hour shows that:

25   4) for a tephra-fallout deposit with features associated with a Vulcanian eruption, the thickness generating the largest instability is between 20 and 25 cm for 6.4 mm h$^{-1}$ and between 20 and 35 cm for 15.5 mm h$^{-1}$;

5) for a tephra-fallout deposit with features associated with a subplinian eruption with $K_s$= 1 × 10$^{-2}$ m s$^{-1}$, the unstable area percentage decreases rapidly with an increase in deposit thickness, being all the area almost stable beyond a 45 cm thickness;

30   6) for a tephra-fallout deposit with features associated with a subplinian eruption with $K_s$= 6.8 × 10$^{-4}$ m s$^{-1}$, almost all the lahar source area results unstable (98%) between 0.1 and 0.65 m deposit thickness; beyond 0.65 m the fraction of unstable area decreases with the increase in tephra thickness increase for the rainfall of 6.4 mm h$^{-1}$;

7) for a tephra-fallout deposit with features associated with a Vulcanian eruption we observe an unstable window with

a minimum thickness for instability and suggest that cohesion leads to a critical minimum landslide depth dependent on the slope angle. Instead, cohesionless material such as the primary Pal D deposit is unstable for all small deposit thickness values and a low thickness limit was not reached; the tephra-fallout deposit becomes stable with thickness increase depending on rainfall and slope angle.

The results modelled with TRIGRS show that shallow landsliding is an effective process for eroding both Vulcanian-type and subplinian-type (with $K_s = 6.8 \times 10^{-4}$ m s$^{-1}$) tephra-fallout deposits in combination with high-intensity rainfall events with short duration, such as those occurring in Vulcano every year. Nonetheless, the occurrence of shallow landsliding is a complex process (e.g., Table 7 and Fig. 10 and 13) and the tephra fallout deposit thickness threshold strongly depends on

10  rainfall intensity, tephra-fallout deposit characteristics and geomorphology features. Both eruptive scenarios (e.g., plume height, erupted mass, eruption duration) and prevailing wind direction are, therefore, crucial to the generation of rain-triggered lahars, having a first order control on tephra-fallout deposit thickness. Physical characteristics of tephra-fallout deposits (e.g. hydraulic conductivity, grainsize and cohesion) and geomorphological features (e.g. flank slopes), the characteristics of soil at the base of the deposits and vegetation are also important parameters to consider as they have a first

15  order control on slope instability. We can conclude that deposit thickness and rainfall intensity alone are not sufficient to derive thresholds for lahar triggering; a comprehensive assessment of unstable volumes that could potentially trigger lahars, in fact, requires dedicated numerical simulations combined with detailed field observations and geotechnical analysis as we did in this study.

**Data availability**

20  Most data is made available in tables. Additional data is available upon request, based on a collaborative agreement.

**Appendix**

**Appendix A: Field strategies**

**Sampling of undisturbed deposit for geotechnical tests**

25  Undisturbed tephra-fallout deposit is sampled for testing the properties in the laboratory, without disturbing structure texture, density, natural water content and stress condition (Figs A1a and b). Sampling was performed by inserting a steel tube 3 mm thick with a height of 30 cm and a diameter of 10 cm into the ground (Fig. A1a). After that, we cleaned all the deposit around the tube to extract it with a minimum disturbance. Then, a support was inserted at the base of the cylinder, and the

tube was extracted from the deposit. Finally, the tube was covered on both ends with a plastic cover and plastic tape to preserve the deposit for disturbance during the transport (Fig. A1b).

**Soil suction measurement**

Soil suction measurements were carried out in situ on the 1888-90 tephra-fallout deposit with a soil moisture probe ("Quick Draw" Model 2900FI) (Fig. A2b). The first step in taking a reading with the probe is to core a hole pushing the coring tool into the soil (Fig. A2a). After removing the coring tool we have a proper sized hole to insert the probe. The second step is to insert the probe in the soil and wait approximately one minute (equalization time assessed for such soils). Finally, the suction can be read on the dial gauge (Fig. A2b). The soil suction is created by water capillary pressure that each soil particle applies into the soil. The moisture probe has a porous ceramic sensing tip at the end of the tube. The soil suction reading is obtained when a small amount of water transfers between the sensing tip of the probe and the soil.

**Saturated hydraulic conductivity measurement**

The saturated hydraulic conductivity was estimated in the field with a single-ring permeameter for both deposits (Figs A3a and b). The apparatus for the measurements consists of a steel ring with diameter of 21 cm and height of 12 cm, a plastic cover with a hole to insert a Mariotte bottle (Fig. A3b). In the field, we put the ring on a horizontal plane surface on the tephra-fallout deposit. Then, the first 6 cm were pushed into the ground. Finally, we filled the Mariotte bottle with water and inserted it on the tape turned upside-down. The water first formed a 1 cm layer on the tephra-fallout deposit and then started to infiltrate into the deposit. For the infiltration rate measurements the readings were done every minute in the case of 1888-90 eruption deposit and every 30 seconds in the case of Pal-D deposit. The duration of the measurements was 40 minutes for 1888-90 deposit and 3.2 minutes for the pal-D deposit.

[Figure]

[Figure]

**Figure A1: a) Sampling the 1888-90 AD tephra sequences with a 30 cm steel tube. We clean all the deposit around the tube to extract it with a minimum disturbance. B) Sampling the Pal-D deposit with a 30 cm steel tube. The tube is covered with a plastic cover and plastic tape before to extract the tube from the deposit with a basal support.**

[Figure]

[Figure]

[Figure]

**Figure A2: a) Coring tool into the soil before soil suction measurement b) Soil suction measurement on the 1888-90 primary deposit on the NW volcano flank.**

[Figure]

**Figure A3: a) Ring infiltrometer, 6 cm are buried into the 1888-90 primary deposit. b) Ring infiltrometer during the infiltration measurements. With the bottle turned upside-down with water infiltrating in the ground.**

**Competing interests**

10   The authors declare that they have no conflict of interest.

**Acknowledgments**

The authors are grateful to Irene Manzella, Michel Jaboyedoff, and Mario Sartori for constructive discussion and to Corine Frischknecht, Mauro Rosi and Ramon Arrowsmith for discussion and assistance in the field. 
[revised manuscript text omitted]

[Figure]

[Figure]

**Tables**

Table 1: Hydraulic conductivity measured in the field

| unit | test | $Ks$ (m s$^{-1}$) |
|---|---|---|
| **Pal D** | 1 | $6.8 \cdot 10^{-4}$ |
| **V 88-90** | 1 | $6.0 \cdot 10^{-5}$ |
| **V 88-90** | 2 | $7.5 \cdot 10^{-5}$ |

Table 2: Suction measured in the field on the
1888-90 tephra-fallout deposit and Md$\phi$ for Su5
and Su7 (Fig. 1). NA = Not available

| Location | Suction (kPa) | Md$\phi$ |
|---|---|---|
| Su1 | 25 | NA |
| Su2 | 24 | NA |
| Su3 | 27 | NA |
| Su4 | 14 | NA |
| Su5/V01 | 15 | -0.27 |
| Su6 | 15 | NA |
| Su7/V08 | 20 | 0.90 |

[Figure]

[Figure]

Table 3: Summary of the physical characteristics of the tephra-fallout and lahar samples analysed. 30*: slope is measured on GIS (Geographic Information System). *Thick.* refers to the total deposit thickness. *Unit* refers to the 1888-1890 eruption (1888-90), the Palizzi D eruption (Pal D) and to the September 2017 lahars. *T* refers to the primary fallout and *L* to the associated lahars. For V1 and V2, the horizon shows the sampled section interval. *F1* and *F2* refers to the weight sample fraction < 1mm and < 63 µm, respectively.

[revised manuscript text omitted]

**Long-lasting Vulcanian eruption scenario**

| Probability (%) | Duration (months) | NW catchment Thickness (cm) | Volumes (m³) Total | Unstable | S catchment Thickness (cm) | Volumes (m³) Total | Unstable |
|---|---|---|---|---|---|---|---|
| 25 | 9 | 11-13 | 574 | 246 | 15-16 | 734 | 585 |
| 25 | 12 | 14-16 | 748 | 601 | 20-21 | 959 | 952 |
| 25 | 18 | 20-24 | 1061 | 991 | 28-30 | 1353 | 1105 |
| 25 | 24 | 25-30 | 1326 | 861 | 35-37 | 1689 | 95 |
| 75 | 9 | 6.5-7.5 | 333 | 0 | 9 | 427 | 0 |
| 75 | 12 | 9-10 | 417 | 0 | 11 | 533 | 21 |
| 75 | 18 | 9.5-11.5 | 489 | 47 | 13-14 | 627 | 231 |
| 75 | 24 | 10-12 | 507 | 74 | 13.5-14.5 | 649 | 319 |

**Plinian eruption scenario**

| Probability (%) | VEI | NW catchment Thickness (cm) | Volumes (m³) Total | Unstable | S catchment Thickness (cm) | Volumes (m³) Total | Unstable |
|---|---|---|---|---|---|---|---|
| 25 | 2 | 11-15 | 589 | 563 | 32-40 | 1663 | 0 |
| 25 | 3 | 37-58 | 2455 | 0 | 101-112 | 5112 | 0 |
| 75 | 2 | 1.2-2 | 77 | 77 | 3.2-3.6 | 187 | 185 |
| 75 | 3 | 2-8 | 245 | 244 | 13-16 | 736 | 401 |
| 25 | 2* | 11-15 | 589 | 604 | 32-40 | 1663 | 1636 |
| 25 | 3* | 37-58 | 2455 | 2455 | 101-112 | 5112 | 0 |
| 75 | 2* | 1.2-2 | 77 | 77 | 3.2-3.6 | 187 | 187 |
| 75 | 3* | 2-8 | 245 | 245 | 13-16 | 736 | 731 |

[Figure]

[Figure]

Table 8: Measured median grain size, hydraulic conductivity and infiltration capacity on tephra-fallout (TF) and PDC deposits near volcanic vents

| Eruption | Deposit type | Median grain size, Mdφ | Annual precipitation (mm) | Hydraulic conductivity (m/s) | Post-eruption infiltration capacity (m/s) | Data source |
|---|---|---|---|---|---|---|
| Vulcano (1888-90 Vulcanian) | TF | -0.90 - 1 | 500 | $8.5 \times 10^{-5}$ | ND | this study |
| Vulcano (Pal D subplinian) | TF | -3.42 | 500 | $1 \times 10^{-2}$ | ND | this study |
| Cordón Caulle 2011 | TF-ash (unit III) | 2.15 | 2500-3000 | $5 \times 10^{-5}$ | ND | a |
| Cordón Caulle 2011 | TF-lapilli (unit I-II) | -1.5 | 2500-3000 | $3.9 \times 10^{-2}$ | ND | a |
| Chaitén 2008 | TF | < 3 | 2600-4300 | ND | ND | b |
| Mt Pinatubo 1990 | PDC | 0-3 | | ND | $1 \times 10^{-4}$ | c |
|  | TF | -1.1- 1.7 | 1950 | ND | ND | c |
| Mt Unzen 1990-95 | PDC | -1-1 | 3100 | ND | $1.25 \times 10^{-5}$
$5 \times 10^{-6}$ | d |
| Mt St Helen 1980 | TF | 1.73 | 1200 | ND | $1.9 \times 10^{-6}$
$1.13 \times 10^{-6}$ | e |

ND = No data. References: a) Baumann et al., 2018 b) Pierson et al., 2013 c) Daag, 2003 d) Yamakoshi and Suwa, 2000 e)
5 Leavesley et al., 1989.

[Figure]

**Figures**

[revised manuscript text omitted]

---

## Referee Comment (RC2) · Jenni Barclay (Referee) · 16 Jul 2019

This is an interesting paper that tackles an archetype of a cascading hazard (rainfall-induced remobilisation of volcanic deposits) in a specific contect (Vulcano).

As a strength it deploys a wide range of analytical and modelling techniques from across volcanology and geotechnical studies to understand the likely susceptibility of the generation of rain-triggered lahars following an eruption of Vulcano. The paper is generally well written and each individual element of data is carefully described and the

approach justified, and limitations carefully thought through. It provides some useful primary data on hydraulic conductivity and other relevant characteristics of the deposits and promises to 'provide a new innovative treatment of the cascading effect between tephra fallout and lahar susceptibility'.

The weakness in this paper comes in the integration of these elements and in considering whether there is sufficient information to consider the conclusions of the parametric analysis to be robust. I have several broader comments to make here, in the hope that addressing these might improve the integration of data and thus the robustness of the conclusions.

One thing that may particularly help with some of the issues around choices to be made with parametrisation and applicability of the system is via the inclusion of any contemporary accounts of lahar activity from the 1888 eruption. Observations of lahars (or even immediate geomorphological change) are not included in the paper (just the discussion of existing 'undisturbed' lahar deposits left on the island now). Even if the population was evacuated it may be that there are some contemporary accounts which might help to validate some of the choices you have made in terms of analytical focus and also provide something against which to measure your conclusions, and might also help with the interpretation of those deposits you have encountered. Similarly, do you have contemporary accounts of weather patterns following this eruption?

Introduction: A distinction between lahar archetypes that is more generally made is that described in Pierson et al., (2014) – which is between 'primary' and 'secondary' lahars – this more obviously has a similarity in mechanism than the examples you give. I'd suggest you would use this more commonly used typology here and throughout – it might make things less confusing. I realise that you regard these as 'syn-eruptive' but really these are secondary lahar generated via syn-eruptive rainfall? (see also lines 0-7 on p. 4).

2.1 and Figure 1. Eruptive history. To allow an evaluation of the validity of your choice

in eruptive scenarios a summary figure of the last 1000 years of activity as a function of eruption size and duration would be useful (in a similar way to how you have synthesised typical rainfall in Figure 2). 2.3 Recent lahars. This and the description September 2017 lahar (p.10) are a little distracting/confusing. They describe the cumulative morphological change and the nature of flows some 120+ years after the last eruption. It would be more salient here to describe some of the observations of any laharic activity or immediate morphological changes following the 1888 eruption, as mentioned above. A critical feature here of the eruption is the extent to which 30cm of deposit is representative of different eruptive phases. If the triggering mechanism is landsliding, then different eruptive stages might supply significant mechanical discontinuity. Section 3.3 – would be easier to evaluate the 'likelihood' of the scenarios used here with a easier to read eruptive history than currently provided in 2.1 3.4 TRIGRS model. . .. This would be easier to follow to the uninitiated reader if the content of paras 2 and 3 was earlier than paragraph one. Section 4,3 modelling. I was not sure of the justification for only one rainfall scenario here. It was also not clear to me the extent to which you had evaluated the influence of deposits from differing eruptive phases (discontinuities in deposits mechanical characteristics). You do discuss in detail the influence of a fine-grained layer on the infiltration (and thus likelihood of instability controlled lahar initiation) but there are more insights available here from the experimental set upo of Jones et al., (2017) – this would point to the influence of antecedent rainfall (in generating armouring conditions for subsequent intense rainfall too) – to go with the observational record from other lahar episodes. This would suggest that the overland runoff via Hortonian flow may be significant even immediately following an eruption. This could be compared for example to observed rainfall patterns which then may help assess the likelihood of the failure mechanism you explore (whether the possibility of both means lahars are even more likely). I think the nature of this analysis such that a summary diagram, considering the interacting dimensions of tephra thickenss, antecedent and contemporaneous rainfall and hydraulic conductivity (as a function of grainsize, porosity and saturation) may be more important in 'suggesting windows' of

eruptive behaviour and rainfall where lahar generation may become a significant hazard. The conclusions drawn may then be more robust as given all the caveats and unknowns I am not sure the thresholds and % you present here are fully defensible.

References Used Pierson, T.C. et al (2014) J App Volc. 3:16 Jones, R et al., (2017) Geomorphology 282:39-51

---

## Author Response (AR1)

**FACULTÉ DES SCIENCES**

SECTION DES SCIENCES DE LA TERRE
ET DE L'ENVIRONNEMENT
**DÉPARTEMENT DES SCIENCES DE LA TERRE**

Costanza Bonadonna
Professeure

Natural Hazard and
Earth System Science

Geneva, August 17, 2019

Dear Editor,

Please find enclosed the revised version of our manuscript "Mapping the susceptibility of rain-triggered lahars at Vulcano island (Italy) combining field characterization, geotechnical analysis and numerical modelling"  by Valérie Baumann, Costanza Bonadonna, Sabatino Cuomo, Mariagiovanna Moscariello, Sebastien Biass, Marco Pistolesi and Alessandro Gattuso addressing all comments of the two Reviewers.

We believe that our contribution will not only provide fundamental insights for a better comprehension of rainfall-induced lahars but could also suggest new ways to evaluate lahar hazard. Lahars represent one of the most significant hazards to people living in volcanic areas and the characterization of volumes of pyroclastic material that can be potentially remobilized by water is crucial to accurate hazard assessments. Nonetheless, most of the research on lahar hazard is related to lahar inundation based on numerical models of variable complexity.

In this study we analyse the spatial distribution of the potential lahar sources in association with two eruptive scenarios that have characterized the activity of La Fossa cone (Vulcano, Italy): a long-lasting Vulcanian cycle and a subplinian eruption. In particular, we explore the spatial distribution of the potential lahar sources in association with two eruptive scenarios that have characterized the activity of La Fossa cone (Vulcano, Italy): a long-lasting Vulcanian cycle and a subplinian eruption. Our analysis is based on a combination of tephra-fallout probabilistic modelling (with TEPHRA2), slope stability modelling (with TRIGRS), field observations and geotechnical tests.Tephra-fallout properties (hydraulic conductivity, friction angle, cohesion, total unit weight of the soil, saturated and residual water content) required by numerical models and obtained for the primary tephra-fallout deposits on the La Fossa cone represent the first data derived for this region.

The comments of the reviewers have helped us strengthen the manuscript. In fact, it has been substantially restructured and rewritten based on comments of both reviewers in the attempt to clearly identifies the importance and objectives of this study, concisely presents data relevant to support the study, and critically highlights results of importance and explains their significance to the reader. We hope that these substantial modifications (see responses to reviewers here below) have made the manuscript suitable for publication.

Sincerely,

Costanza Bonadonna
on behalf of all authors

Rue des Maraîchers 13, CH-1205 Genève, Suisse
costanza.bonadonna@unige.ch

Tél. +41 22 379 30 55 - Fax +41 22 379 32 10
http://www.unige.ch/hazards/About/People/Bonadonna.html

**Detailed response to reviewers.**

We thank Reviewer 1 for valuable and constructive comments. All specific and technical comments (including the provided PDF) have been addressed (see detailed description below with our comments in red).

1) To be acceptable for publication, I suggest the manuscript needs to be restructured in a way that (a) clearly identifies the importance and objectives of this study, (b) concisely presents data relevant to support the study, and (c) critically highlights results of importance and explains their significance to the reader. Some more detailed comments to be considered in the restructuring is highlighted in the following "specific comments" section

   The paper has been substantially restructured and rewritten based on comments of both reviewers in the attempt to clearly identifies the importance and objectives of this study, concisely presents data relevant to support the study, and critically highlights results of importance and explains their significance to the reader.

2) Page 2, line 18: It is important to clarify that overland erosion and shallow landsliding is the most common mechanism (see e.g. Pierson and Major 2014), as opposed to the only two mechanisms for generating lahars

   We thank the reviewer for this comment, however, in our statement in the introduction, we already say that "sheet and rill erosion" and "infiltration of slope surface by rainfall" are the main two mechanisms that trigger lahars in association with heavy rain. As a result, we think that the statement is sufficiently inclusive of all possible mechanisms.

3) Page 3, line 19-20: "Nonetheless, the use of physically-based models ... is necessary...". I think this is intended to be the crucial aim/thrust of the manuscript, but is not justified from the text before it. What are the weaknesses in an approach such as Tierz et al.? Why is coupling a physical model preferable/necessary? Aspects of Volentik et al. (2009) and Galderisi et al. (2013) might help explain the benefits of such an approach.

   We agree with the reviewer that this part was not very clear, so we have expanded it and added some reference to Volentik et al. 2009 and Galderisi et al. 2013 (introduction).

4) Page 3, line 33 and on: "This approach provides the first integrated attempt ..." to page 4, line 6 "offers an innovative treatment of cascading effects". Both Galderisi et al. (2013) and Volentik et al. (2009) used a similar approach that integrated tephra fall models with a slope stability/shallow landslide model, so I don't necessarily agree with the statements in this paragraph. In combination with the previous comment, I think coupled approaches need to be discussed in more detail to highlight the significance of this work (in my opinion: better considers impact of rainfall on stability, identifies thresholds for initiation and provides fully quantitative estimates of lahar initiation).

   We agree that our work is not the first attempt to couple lahar triggering with probabilistic modelling of tephra sedimentation (see previous comment). However, our work certainly represents the first "integrated attempt to quantify the source volume of lahars as a function of probabilistic hazard assessment for tephra fallout (with TEPHRA2), numerical modelling of lahar triggering (with TRIGRS), field observations (including primary tephra-fallout deposits, geology, geomorphology and precipitations), and geotechnical tests of source deposits." We have expanded the description of the Volentik

et al. 2009, Galderisi et al. 2013 and Tierz et al. 2017 studies in order to clarify this point (introduction).

5) **Field and laboratory methods and results** The field sampling is unclear to me in section 3.1. What is meant by collecting 8 samples "...to retain primary physical characteristics"? Is this where in-situ permeability/soil suction tests were done? On page 5, line 2 - A further 11 samples are collected, looking at Fig. 3 I presume from the Pal D deposit (1 sample), and then at two locations for the 1880-90 vulcanian eruption. This corresponds to samples V1, V2 and V3. However, line 3-7 then discuss location of samples (4 from S La Fossa, 2 from NW La Fossa, 2 from Palizzi valley), but that only sums to 8 samples.

We agree that the sampling description was unclear. We have rewritten this whole part.

6) Aided by confusion in sampling, the first two results sections (4.1, 4.2) are difficult to follow, presenting a large amount of information not directly relevant to the manuscript. The purpose of field and laboratory characterization is to identify parameters of Pal D and 1880-90 deposits for TRIGRS. While grainsize is a consideration for lahar generation and tephra fall, the authors do not use any of this information in their study. Tephra simulations use Biass et al. 2016 results, and no comparison between the chosen size range in Biass et al. 2016 and the values found in this study.

It is correct that Biass et al did not use this grainsize because the input of TEPHRA2 is the total grainsize distribution and not the distribution at individual locations. As a result, the TGSD used in TEPHRA2 and the GS analysed in this paper cannot be compared. However, the grainsize of tephra-fallout deposit was analyzed to compare it with that of lahar deposits in order to relate the source area with eh remobilized material. In addition, grain size distribution of source area is required to estimate the Soil Water Retention Curve of soil using the Kovac and Aubertin model necessary for TRIGRS (see section 3.2). In fact, this model requires grainsize data including the diameter corresponding to 10% and 60% passing on the grain size curve (i.e., D10 and D60), and the liquid limit. We have, therefore, kept the Mdphi vs sorting plot in the main text and moved all tables and figures of grainsize distribution in the supplementary material.

7) The sample naming scheme is hard to follow in context, switching from unit to location to site to unit in the tables, and unit to site to sample number in the figures. I suggest simplifying both the description of sampling and presentation of results from field and laboratory analyses. In this study, we are interested in the Pal-D and 1880-90 tephra and lahar units. Individual data from each location (Tables 1-3) can be provided as supplementary material, and results should focus on how Geotechnical (table 4) and input parameters (table 5) are derived from your sampling campaign. I fail to see how the extensive study on grainsize is necessary here, beyond a few sentences, and would recommend shifting figures 5 and 6 to supplementary material.

We reorganized the table 3 and put tables 1-3 in supplementary material. Figure 6 was moved to supplementary material and we have kept Fig 5 in the main text to illustrate the grainsize similarity between primary and lahars deposits.

8) Page 12, line 33: Were two upper catchments exactly the same size, or were they of similar size? How were catchment boundaries defined (i.e. from the slope or drainage networks/external data)?

Yes, they are of same size and we have added an explanation on how we have defined them (section 4.3 Modelling).

9) Figure 9: It is better to make this figure greyscale compatible and easier to interpret. I would suggest something like using dashed lines for 25

Good suggestion! We have made the figure greyscale compatible.

10) Section 4.3.1 - This section seems to show that increasing the tephra thickness above a certain threshold will increase stability, nicely leading onto section 4.3.2. Figure 10 doesn't seem to add much to this discussion over table 7, so I would recommend removing it.

We agree with this comment. However, we also feel that Fig. 10 helps visualize results of Table 7. Given the importance of these results, we feel that this Figure should be kept.

11) Page 14, line 2-4: It is unclear how deposit thickness was increased. Was this assuming a constant depth of deposit across the entire NW and S area, or was it applied as a proportion of the isopachs (either Tephra2 or observed)?

Thanks for this comment. We have explained in the text that the thickness was considered constant over the whole source area (section 4.3.2)

12) Page 15, line 1 and on: "... total pressure head has a higher maximum displaced to higher tephra fallout deposit thickness..." What does maximum displaced mean here?

We agree that this statement was confusing. We have rewritten this all section.

13) Page 15, line 14 - page 16, line 2 : I do not understand the relevance to Cordon Caulle in the entire section, and it seems misguided. The friction angle of a granular material is controlled by the distribution (_) of grainsize, asperity and roughness; not mean grain size. Hydralic conductivity for these two different eruptions would be expected to differ, as the Pal D deposits are much coarser than Cordon Caulle lapilli. Some 'washing' of fines over time may occur, but if differences in measuring techniques cause a 2 orders of magnitude difference in conductivity, then the techniques are unreliable. This section is better served by starting with Page 16, line 3 (Table 8 ...).

We agree with these comments and this part has been removed.

14) Another consideration in section 5.1 is the volume of lahars. Lahar volumes for all the other examples in table 8 are quite large, in comparison to the smaller lahars at Vulcano.

We agree with this comment and added some lahar volumes for the volcanoes listed in the table and discussed it in section 5.1.

15) Page 20, lines 6 - 8: How was the assessment of unstable areas found to be accurate? Without validation against a specific event (or set of events), a methodology has been shown to identify unstable areas.

We have removed this statement

We thank Prof. Barclay (Reviewer 2) for valuable and constructive comments. All comments have been addressed (see detailed description below with our comments in red).

1) One thing that may particularly help with some of the issues around choices to be made with parametrisation and applicability of the system is via the inclusion of any contemporary accounts of lahar activity from the 1888 eruption. Observations of lahars (or even immediate geomorphological change) are not included in the paper (just the discussion of existing 'undisturbed' lahar deposits left on the island now). Even if the population was evacuated it may be that there are some contemporary accounts which might help to validate some of the choices you have made in terms of analytical focus and also provide something against which to measure your conclusions, and might also help with the interpretation of those deposits you have encountered. Similarly, do you have contemporary accounts of weather patterns following this eruption?

Unfortunately, there is no accurate description of lahars in any of the available chronicles. Lahars clearly occurred as we see them in the deposit and because they occur even at present day (even if in small size given that the original material has almost all gone). Mercalli (1891) do not describe any mud flow, probably because he was not interested in this phenomenon. De Fiore (1922) describes erosion of the 1888-90 deposit which was still loose in 1921 when he observed it. Interesting to notice also that the eruption started on August 3, ie during the dry season, so when lahars are not expected to form. De Fiore 1922 also mentions that he installed a rain station that did not work and also the station in Sicily were not working at that time apparently. But he mentions that the most intense rain typically happens in November, while December is the months with the most frequent rain events. July is the month with the least intense and frequent rain events. All this is in agreement with the recent observations we describe in our paper (Fig. 2). So, it seems that the weather pattern in this region is pretty constant. We have added the information of De Fiore (1922) to the text to clarify this points.

2) Introduction: A distinction between lahar archetypes that is more generally made is that described in Pierson et al., (2014) – which is between 'primary' and 'secondary' lahars – this more obviously has a similarity in mechanism than the examples you give. I'd suggest you would use this more commonly used typology here and throughout – it might make things less confusing. I realise that you regard these as 'syn-eruptive' but really these are secondary lahar generated via syn-eruptive rainfall? (see also lines 0-7 on p. 4).

Thanks for this comment. However, as we mention in the introduction, we follow the definition of Sulpizio et al. (2006), in agreement with Vallance and Iverson (2015) and Gudmunsson (2015), that syn-eruptive lahars occur during or just after the eruption. The modelled lahars are expected to initiate during the Vulcanian cycle (and therefore syn-eruptive to the cycle) or just after the subplinian eruption. In contrast, recent lahars can be clearly considered as post-eruptive lahars.

3) 2.1 and Figure 1. Eruptive history. To allow an evaluation of the validity of your choice in eruptive scenarios a summary figure of the last 1000 years of activity as a function of eruption size and duration would be useful (in a similar way to how you have synthesised typical rainfall in Figure 2).

We agree with the reviewer and we completely rewrote section 2.1, in order to better describe the stratigraphy of the selected period of activity. We also present the extrapolated recurrence of the eruptive scenarios (new Table 1). To help with this, we

also redrew Figure 1 which is now accompanied by a synthetic log of the last 1000 years of activity.

4) 2.3 Recent lahars. This and the description September 2017 lahar (p.10) are a little distracting/confusing. They describe the cumulative morphological change and the nature of flows some 120+ years after the last eruption. It would be more salient here to describe some of the observations of any laharic activity or immediate morphological changes following the 1888 eruption, as mentioned above. A critical feature here of the eruption is the extent to which 30cm of deposit is representative of different eruptive phases. If the triggering mechanism is landsliding, then different eruptive stages might supply significant mechanical discontinuity.

We agree with this comment. This part has been moved to supplementary material.

5) Section 3.3 – would be easier to evaluate the 'likelihood' of the scenarios used here with a easier to read eruptive history than currently provided in 2.1

Following the reviewer's comment and our strategy used for comment #3, we insert a table (Table 1) in which we show the number of events and the recurrence time for each type of activity.

6) 3.4 TRIGRS model. This would be easier to follow to the uninitiated reader if the content of paras 2 and 3 was earlier than paragraph one.

We have restructured this section following this comment

7) Section 4,3 modelling. I was not sure of the justification for only one rainfall scenario here.

We have strengthened the explanation of the use of one rainfall scenario (section 4.3). In fact, the selected scenario (heavy-torrential precipitation) represents the most intense precipitation scenario based on available data and is used to investigate the maximum unstable tephra volume. We also note that the parametrization analysis was carried out using two rainfall scenarios in order to investigate the effect of variable rainfall (see Figs 11 and 12 and Table 6).

8) It was also not clear to me the extent to which you had evaluated the influence of deposits from differing eruptive phases (discontinuities in deposits mechanical characteristics). You do discuss in detail the influence of a fine-grained layer on the infiltration (and thus likelihood of instability controlled lahar initiation) but there are more insights available here from the experimental set up of Jones et al., (2017) – this would point to the influence of antecedent rainfall (in generating armouring conditions for subsequent intense rainfall too) – to go with the observational record from other lahar episodes. This would suggest that the overland runoff via Hortonian flow may be significant even immediately following an eruption. This could be compared for example to observed rainfall patterns which then may help assess the likelihood of the failure mechanism you explore (whether the possibility of both means lahars are even more likely). I think the nature of this analysis such that a summary diagram, considering the interacting dimensions of tephra thickenss, antecedent and contemporaneous rainfall and hydraulic conductivity (as a function of grainsize, porosity and saturation) may be more important in 'suggesting windows' of eruptive behaviour and rainfall where lahar generation may become a significant hazard. The conclusions drawn may then be more

robust as given all the caveats and unknowns I am not sure the thresholds and % you present here are fully defensible.

In this work we did not consider the influence of discontinuities in deposit mechanical characteristics because this cannot be described in TRIGRS. However, PAL D deposits are massive and the thickness of individual layers for the 1888-90 deposit is very small (mostly < 1 cm) and, therefore, we assume that they are also massive.
We added the rainfall simulation experimental results from Jones et al. 2017 on fine grained and coarse-grained tephra fallout deposits in section 5.1. However, our outcomes cannot be directly compared with the outcomes of Jones et al. (2017) because, unfortunately, hydraulic conductivity was not measured in this experiments and antecedent rain could not be investigated with TRIGRS. Regarding the armouring conditions, we have not observed any of this even at present days. This is also in agreement with the observations of De Fiore 1922. We consider then that this has a negligible effect on lahar triggering for 1888-90 and Pal D deposit. Nonetheless, we have added a discussion on this in section 5.1.
Finally, we have added a summary table (Table 6) to better describe the outcomes of Fig 11 and 12 and simplified the description of this part in the main text.

**Mapping the susceptibility of  rain-triggered lahars at Vulcano island (Italy) combining field characterization, geotechnical analysis and numerical modelling**

Valérie Baumann[1], Costanza Bonadonna[1], Sabatino Cuomo[2], Mariagiovanna Moscariello[2], Sebastien Biass[3], Marco Pistolesi[4], Alessandro Gattuso[5]

[1] Department of Earth Sciences, University of Geneva, Rue des Maraîchers 13, 1205 Geneva, Switzerland

[2] Laboratory of Geotechnics, University of Salerno, Via Giovanni Paolo II 132, 84081 Fisciano Salerno, Italy

[3] Earth Observatory of Singapore, Nanyang Technological University, Singapore, Singapore

[4] Dipartimento di Scienze della Terra, Università di Pisa, Pisa, Italy

[5] Istituto Nazionale di Geofisica e Vulcanologia, Sezione Palermo, Italy

**Abstract.** The characterization of triggering dynamics and of remobilised volumes is crucial to the assessment of associated lahar hazard. We propose an innovative treatment of the cascading effect between tephra fallout and lahar hazards based on probabilistic modelling that also accounts for a detailed description of source  sediments. As an example, we have estimated the volumes of tephra-fallout deposit that could be remobilised by rainfall-triggered lahars in association with two eruptive scenarios that have characterized the activity of La Fossa cone (Vulcano, Italy) in the last 1000 years: a long-lasting Vulcanian cycle and a subplinian eruption. The spatial distribution and volume of  deposits that could potentially trigger lahars were analysed based on a combination of tephra-fallout probabilistic modelling (with TEPHRA2), slope stability modelling (with TRIGRS), field observations and geotechnical tests.  Model input data (were obtained from both geotechnical tests and field measurements (e.g. hydraulic conductivity, friction angle, cohesion, total unit weight of the soil, saturated and residual water content) ~~were obtained from both geotechnical tests and field measurements. In particular, hydraulic conductivity plays an important role on the stability of tephra-fallout deposits. Our parametric analysis has shown that the tephra-fallout critical thickness required to trigger a lahar for the considered rainfall event is between 20-25 cm for the Vulcanian scenario, and between 10-65cm or <10cm for a subplinian event depending on the hydraulic conductivity. The scenario remobilizing the largest unstable volumes by rain-triggered lahars is, therefore, that associated with a Vulcanian cycle with duration of 18 months and a subplinian eruption of VEI 3 (for low hydraulic conductivity). TRIGRS simulations show that shallow landsliding is an effective process for eroding the primary tephra-fallout deposits in combination with high-intensity rainfall events with short duration, such as those occurring on Vulcano every year. Our results provide a new innovative treatment of the cascading effect between tephra fallout and lahar susceptibility that also accounts for detailed characterization of source sediments based on field observations and geotechnical tests~~). 
[revised manuscript text omitted]

15  *September 2017 lahar*
~~We also described a small lahar event that occurred in September 2017, one month before our 2017 field work (Fig. 4e). The lahar source area was located on the NW cone flank in a funnel shaped area located above a small gully at an elevation of 159 m a.s.l. The lahar flowed into a gully with an average width of 2 meter and a depth varying between 0.4 and 0.8 m, formed levees on both sides and stopped on the La Fossa crater trail with a final runout of 120 m. The area of the front lobe deposit~~

20  ~~was measured with a handheld GPS (135.5 m²) and approximate thickness estimated (0.3 m) in the field, which resulted in a volume of ~40 m³. A second lahar flow pulse deposited a small deposit confined within the channel (Fig. 5e). Two samples (V11 and V12) were taken from this recent lahar deposit in order to compare with the older syn-eruptive lahar deposits.~~

[revised manuscript text omitted]
 results for the Pal D primary deposit, for both values of $K_s$, are seen in Fig. 13. We simulated two  rainfall intensities have been considered (6.4 mm h$^{-1}$ with a duration of 5 hours

35 and 15.5 mm h$^{-1}$ with a duration of 3 hours. In particular, the rainfall intensity of 15.5 mm h$^{-1}$ represents the worst rainfall scenario for 2017 (rainfall event recorded at Lentia station the 11 November 2017, with a total of 46.5 mm). In summary, Table 6 shows how a lapilli-rich tephra-fallout deposit with a low hydraulic conductivity is unstable

40 for slopes >30° regardless of the associated thickness (deposit features of Pal D eruption); nonetheless, Fig. 10 shows that the deposit becomes stable for thickness values >65 cm. In fact, a constant FS (dashed lines in Fig. 12b,d) is the result of the upper boundary of pressure head, which is physically limited at the beta-line

(Iverson, 2000; Baum et. al., 2008). The total pore pressure cannot be above the values denoted by a water table at the ground surface (beta-line) and the model calculates FS with this value, which is the worst condition for instability. In contrast, the same lapilli-rich tephra-fallout deposit is unstable at all observed slopes only for thickness values <10-20 cm in case of high hydraulic conductivity. Finally, a tephra-fallout deposit dominated by coarse ash is unstable at slopes <35.4° mostly for thickness values between about 10-40 cm (deposit features of the 1888-90 eruption); for a slope >38° the same tephra-fallout deposit is unstable for thickness values larger than 13 cm. For the same tephra-fallout deposit the ratio of rainfall intensity and hydraulic conductivity ($I/K_s$) determines the time to reach the water table (located at the bottom of the deposit in our case study); the rate of rise of water table increases with an increase in $I/K_s$ ratio (Li et al., 2013). In the case of the 1888-90 tephra-fallout deposit, the upper critical thickness for instabilities increases with the increase of rainfall intensity and total rainfall. It is also important to note how the maximum value of total pore pressure, and, therefore, the potential for triggering lahars, is shifted toward larger values of tephra-fallout deposit thickness when rainfall intensity is increased. As an example, the maximum value of pore pressure for 38° slope angle (blue solid line in Figs 12a,c) is reached at 15 cm and 30 cm for a rainfall intensity of 6.4 mm h⁻¹ and 15.5 mm h⁻¹, respectively.

~~For the Pal D primary deposit with high hydraulic conductivity, the pressure head increases with decreasing thickness and reaches a maximum of 0.07 m, 0.08 m and 0.087 m for a tephra thickness of 15 cm (Fig. 13a). In contrast, the total pressure head shows a monotonic increase with increasing deposit thickness for low hydraulic conductivity. In the case of the safety factor for high hydraulic conductivity, only very shallow deposits are unstable and for slope angles of 38° the tephra-fallout deposit with a thickness larger than 20 m is stable. In contrast, the FS is constant in the case of low hydraulic conductivity and is below 1 for deposit thicknesses between 10 cm and 55 cm (Fig. 13b).~~

~~For a rainfall intensity of 15.5 mm h⁻¹ with a duration of 3 hour (Fig. 13c, d) and with a high hydraulic conductivity, the total pressure head has a higher maximum displaced to higher tephra-fallout deposit thickness and the deposit thickness limit between stable and unstable area is higher (almost 32 cm for a slope angle of 38°) compared with the 6.4 mm h⁻¹ rainfall intensity (Fig. 13a, b). In contrast, in the case of the low hydraulic conductivity the total pressure head and the safety factor are the same as the results obtained for the lower rainfall intensity (6.4 mm h⁻¹). The constant factor of safety~~

 the factor of safety with this value, which is the worst condition for instability.

**5. Discussion**

**5.1 Characteristics of lahar source deposits**

~~Grain-size distribution of pyroclastic material is one of the primary factors influencing the type of erosion or failure mechanism in the case of rainfall-triggered lahars (e.g., Manville et al., 2000, Pierson et al., 2013). In this study, critical characteristics of tephra-fallout deposits (i.e. grain-size, hydraulic conductivity and angle of friction) necessary as input for shallow landsliding process modelled with TRIGRS have been analysed for two different eruption scenarios (Vulcanian and subplinian). Overall, the tephra fallout deposits associated with both the 1888-90 Vulcanian eruptions and the subplinian Pal D eruption are relatively internally homogeneous in terms of grain-size and geotechnical properties (Tables 1-5). In contrast, the tephra-fallout sequence associated with the climactic phase of the 2011 Cordón Caulle eruptions (Chile) that also generated post-eruption lahars is characterized by two different layers with contrasting grain-size: lapilli at the base (Units I and II) and ash on the top (Unit III) (Pistolesi et al., 2015). In terms of friction angle, Pal D deposits and lapilli layers I and II of Cordón Caulle eruption are very similar, 54° and 53°, respectively. However, the hydraulic conductivities measured in the field are very~~

~~different ($K_s$ = 3.9x10$^{-2}$ m s$^{-1}$ and $K_s$ = 6.8x10$^{-4}$ m s$^{-1}$, respectively). This significant difference in hydraulic conductivities can be due partly to the age and the geological setting of the primary deposits and partly to the use of different measuring techniques. In the case of 2011 Cordón Caulle lapilli (Units I and II) the hydraulic conductivity was measured by filling a plastic tube with an undisturbed sample and saturating the materiel with water. Instead, measurements for the Pal D primary deposit was performed on the outcrop with a single ring permeameter (see Appendix A). Besides, the 2011 Cordón Caulle deposit had only a small layer (10 cm) on the top and 0.7% of fine ash. The Pal D deposit is older (AD 1200), with 7% fine ash and overtopped by almost two meters of younger pyroclastic deposits. The lower hydraulic conductivity of the Pal D deposit compared to the 2011 Cordón Caulle lapilli layers could be due to the migration of small particles from the top and a compaction due to the load of the deposits on the top, which reduced the porosity. In fact, the dry unit weight of the Pal D undisturbed sample and the same a reconstructed sample (without compaction) is 6.63 kN m$^{-3}$ (Table 4) and 6.18 kN m$^{-3}$ respectively, which means a greater porosity (6.7%) for the reconstructed sample. Although the 1888-90 primary deposits have a slightly greater friction angle than the ash layers (Unit III) from the 2011 Cordón Caulle eruption (41° and 38.4°, respectively), the hydraulic conductivities are in the same range. The 1888-90 primary deposit also has similar friction angles and hydraulic conductivities with respect to the pyroclastic deposits (ash soil class B) from Vesuvius (Cascini et al., 2010).~~

[revised manuscript text omitted]

Table 3: Summary of the physical characteristics of the tephra-fallout and lahar samples analysed. 30*: slope is measured on GIS (Geographic Information System). *Thick.* refers to the total deposit thickness. *Unit* refers to the 1888-1890 eruption (1888-90), the Palizzi D eruption (Pal D) and to the September 2017 lahars. *T* refers to the primary fallout and *L* to the associated lahars. For V1 and V2, the horizon shows the sampled section interval. *F1* and *F2* refers to the weight sample fraction < 1mm and < 63 µm, respectively.

| Site | Name | Thick. (cm) | Horizon | Unit | Slope (°) | Mdφ | σφ | F1 | F2 |
|------|------|-------------|---------|------|-----------|-----|-----|-----|-----|
| S La Fossa cone | V1A | 100 | 0-6 | 1888-90 T | 30 | -0.37 | 1.80 | 35.13 | 3.23 |
| | V1B | 100 | 6-12 | 1888-90 T | 30 | 0.08 | 1.46 | 39.99 | 2.60 |
| | V1C | 100 | 12-18 | 1888-90 T | 30 | -0.88 | 1.63 | 24.03 | 1.22 |
| | V1D | 100 | 18-24 | 1888-90 T | 30 | -0.41 | 1.68 | 32.81 | 0.84 |
| | V1E | 100 | 24-30 | 1888-90 T | 30 | -0.27 | 1.71 | 35.26 | 0.83 |
| NW La Fossa cone base | V2A | 50 | 0-6 | 1888-90 T | 10 | 0.14 | 1.49 | 40.81 | 5.63 |
| | V2B | 50 | 6-12 | 1888-90 T | 10 | 0.09 | 1.78 | 44.97 | 4.13 |
| | V2C | 50 | 12-18 | 1888-90 T | 10 | 0.85 | 1.51 | 62.09 | 7.86 |
| | V2D | 50 | 18-24 | 1888-90 T | 10 | 0.66 | 1.80 | 54.66 | 8.68 |
| | V2E | 50 | 24-30 | 1888-90 T | 10 | 0.90 | 2.10 | 60.41 | 9.71 |
| Pallizi valley | V3 | 25 | - | Pal D T | 10 | -3.42 | 1.55 | 6.31 | 2.83 |
| | V6 | 15 | - | 1888-90 L | 5 | 2.07 | 1.78 | 84.12 | 12.38 |
| S La Fossa cone | V5 | 15 | - | 1888-90 L | 30 | -0.12 | 1.83 | 37.46 | 5.73 |
| Porto di Ponente | V7-1 | 26 | - | 1888-90 L | 0-3 | 0.58 | 2.04 | 51.65 | 7.71 |
| | V7-2 | 11 | - | 1888-90 L | 0-3 | 0.90 | 1.75 | 58.91 | 7.87 |
| | V7-3 | 10 | - | 1888-90 L | 0-3 | 1.38 | 1.73 | 72.86 | 10.31 |
| | V7-4 | 6 | - | 1888-90 L | 0-3 | 1.29 | 1.76 | 68.65 | 8.92 |
| NW La Fossa cone | V8 | 30 | - | 1888-90 L | 30* | -0.29 | 1.67 | 33.27 | 2.51 |
| | V9 | 40 | - | 1888-90 L | 30 | -0.27 | 1.69 | 34.04 | 3.29 |
| | V10 | 30 | - | 1888-90 L | 30 | -0.24 | 1.68 | 34.02 | 2.85 |
| | V11 | 20 | - | 2017 L | 25* | -0.40 | 1.82 | 31.68 | 1.89 |
| | V12 | 20 | - | 2017 L | 25* | -0.02 | 1.82 | 39.42 | 3.16 |

Table 2: Geotechnical parameters for the subplinian tephra-fallout deposit (Pal D) and the Vulcanian tephra-fallout deposit associated with the 1888-90 eruption (Vulc).

| unit | $K_s$ | $D_0$ * | $\varphi´$ | $\Upsilon_s$ wet | $\Upsilon_s$ dry | $c´$ | $\theta_s$ | $\theta_r$ | $\alpha$ | $G_s$ | n | e |
|---|---|---|---|---|---|---|---|---|---|---|---|---|
| | $m\ s^{-1}$ | $m^2\ s^{-1}$ | deg | $kN\ m^{-3}$ | $kN\ m^{-3}$ | kPa | % | % | $kPa^{-1}$ | - | - | - |
| Pal D | $1 \cdot 10^{-2}$ | $6.59 \cdot 10^{-3}$ | 54.00 | 13.70 | 6.64 | 0.00 | 0.72 | 0.04 | 0.93 | 2.42 | 0.72 | 2.57 |
| Vulc | $8.50 \cdot 10^{-5}$ | $3.28 \cdot 10^{-4}$ | 40.98 | 17.00 | 13.51 | 0.00 | 0.47 | 0.03 | 0.28 | 2.57 | 0.47 | 0.87 |

c': cohesion; $\varphi$': friction angle; $\gamma_s$: total unit weight of the soil; $K_s$: saturated conductivity; $D_0$: saturated diffusivity; $\theta_s$: saturated water content; $\theta_r$: residual water content; $\alpha$: Gardner parameter; Gs: specific gravity; n: porosity; e: void ratio. *
15  Diffusivity was evaluated using the procedure proposed in the paper: Rossi et al. (2013).

Table 5: Input parameters for subplinian ($K_s$ from literature), subplinian *($K_s$ measured in the field) and the 1888-90 Vulcanian eruption scenarios used for simulation with TRIGRS.

| unit | $K_s$ | $D_0$ | $\varphi´$ | $\Upsilon_s$ | $c´$ | $\theta_s$ | $\theta_r$ | $\alpha$ |
|---|---|---|---|---|---|---|---|---|
| | $(m\ s^{-1})$ | $(m^2\ s^{-1})$ | (deg) | $(kN\ m^{-3})$ | (kPa) | (%) | (%) | $(kPa^{-1})$ |
| vulcanian | 8.50x10-5 | 1 x 10-4 | 41.00 | 17.00 | 0.5 | 0.47 | 0.029 | 0.28 |
| subplinian | $1x\ 10^{-2}$ | $6.59x10^{-3}$ | 54.00 | 13.70 | 0.00 | 0.72 | 0.04 | 0.93 |
| subplinian * | $6.8x10^{-4}$ | $6.59x10^{-3}$ | 54.00 | 13.70 | 0.00 | 0.72 | 0.04 | 0.93 |

20  c': cohesion; $\varphi$': friction angle; $\gamma_s$: total unit weight of the soil; $K_s$: saturated conductivity; $D_0$: saturated diffusivity; $\theta_s$: saturated water content; $\theta_r$: residual water content; $\alpha$: Gardner parameter;

[revised manuscript text omitted]

**Figure 87: Probabilistic isopach maps (converted from the probabilistic isomass maps of Biass et al. (2016) based on deposit density) and corresponding instability maps compiled with TRIGRS for a Vulcanian eruption with: A) an eruption duration of 6 months and a probability of occurrence of 25%; B) an eruption duration of 12 months and a probability of occurrence of 25%; C)) an eruption duration of 18 months and a probability of occurrence of 25%. D) an eruption duration of 24 months and a probability of occurrence of 25%. The rainfall intensity is 6.4 mm h[-1] with a duration of 5h for all the scenarios and the parameters for the 1888-90 Vulcanian deposits are listed in table 5Table 3.**

[Figure]

**Figure 98: Unstable tephra-fallout volume for the S and NW upper catchments obtained with TRIGRS for eruption durations of 3, 6, 9, 12, 18 and 24 and for probabilities of occurrence of 25% and 75%. UC: upper catchment (see Fig. 7b).**

[Figure]

**Figure 9: Unstable tephra-fallout volume for the S and NW upper catchments obtained with TRIGRS for: 1) the subplinian scenarios VEI 2 and VEI 3 with a $K_s$ = 1x10$^{-2}$ m s$^{-1}$ (from literature) and VEI2, VEI3 with a $K_s$= 6.8x10$^{-4}$ m s$^{-1}$ (as measured in the field) and the probabilities of occurrence 25% and 75%.**

[Figure]

[Figure]

**Figure 10: Percentage of unstable area for the NW and S lahar source areas simulated with TRIGRS for tephra-fallout deposit thicknesses between 0.1-1.1 m and a rainfall intensity of 6.4 mm h$^{-1}$ with a duration of 5 hours and parameters for: Vulcanian tephra-fallout deposits (red squares); subplinian tephra-fallout deposits with $K_s$ = 1x10$^{-2}$ m s$^{-1}$ (value from literature; orange diamonds) and subplinian tephra-fallout deposits with $K_s$= 6.8x10$^{-4}$ m s$^{-1}$ (value measured in the field; green circles).**

[Figure]

**Figure 11: Total pressure head and factor of safety  versus tephra-fallout deposit thicknesses between 0.1 and 0.55 m for Vulcanian tephra-fallout deposits (Table 3) and a rainfall intensity of:  A) and B) 6.4 mm h⁻¹ with a duration of 5 hours (I/Kₛ = 0.02); C) and D) 15.5 mm h⁻¹ with a duration of 3 hours (I/Kₛ = 0.05)  (see also Table 6).**

[Figure]

[Figure]

**Figure 12: Total pressure head and factor of safety  versus tephra-fallout deposit thicknesses between 0.1 and 0.55 m for subplinian tephra-fallout deposits with $K_s = 1 \times 10^{-2}$ m s$^{-1}$ and $K_s = 6.8 \times 10^{-4}$ m s$^{-1}$ (dashed lines) for two different rainfall intensities and durations:  A and B) 6.4 mm h$^{-1}$ with a duration of 5 hours. C) and D) 15.5 mm h$^{-1}$ with a duration of 3 hours. (see also Table 6).**

**Page 35: [1] Formatted**          **Costanza Bonadonna**          **8/13/2019 3:08:00 PM**

English (United Kingdom)

**Page 35: [1] Formatted**          **Costanza Bonadonna**          **8/13/2019 3:08:00 PM**

English (United Kingdom)

**Page 35: [2] Formatted**          **Costanza Bonadonna**          **8/13/2019 3:08:00 PM**

English (United Kingdom), Not Superscript/ Subscript

**Page 35: [2] Formatted**          **Costanza Bonadonna**          **8/13/2019 3:08:00 PM**

English (United Kingdom), Not Superscript/ Subscript

**Page 35: [3] Formatted**          **Costanza Bonadonna**          **8/13/2019 3:08:00 PM**

English (United Kingdom)

**Page 35: [3] Formatted**          **Costanza Bonadonna**          **8/13/2019 3:08:00 PM**

English (United Kingdom)

**Page 35: [4] Formatted**          **Costanza Bonadonna**          **8/13/2019 3:08:00 PM**

English (United Kingdom), Not Superscript/ Subscript

**Page 35: [4] Formatted**          **Costanza Bonadonna**          **8/13/2019 3:08:00 PM**

English (United Kingdom), Not Superscript/ Subscript

**Page 35: [5] Formatted**          **Costanza Bonadonna**          **8/13/2019 3:08:00 PM**

English (United Kingdom), Not Superscript/ Subscript

**Page 35: [5] Formatted**          **Costanza Bonadonna**          **8/13/2019 3:08:00 PM**

English (United Kingdom), Not Superscript/ Subscript

**Page 35: [6] Formatted**          **Costanza Bonadonna**          **8/13/2019 3:08:00 PM**

English (United Kingdom), Not Superscript/ Subscript

**Page 35: [6] Formatted**          **Costanza Bonadonna**          **8/13/2019 3:08:00 PM**

English (United Kingdom), Not Superscript/ Subscript

**Page 35: [7] Formatted**          **Costanza Bonadonna**          **8/13/2019 3:08:00 PM**

English (United Kingdom)

**Page 35: [7] Formatted**          **Costanza Bonadonna**          **8/13/2019 3:08:00 PM**

English (United Kingdom)

| Page 35: [8] Formatted | Costanza Bonadonna | 8/13/2019 3:08:00 PM |

English (United Kingdom), Not Superscript/ Subscript

| Page 35: [8] Formatted | Costanza Bonadonna | 8/13/2019 3:08:00 PM |

English (United Kingdom), Not Superscript/ Subscript

| Page 35: [9] Formatted | Costanza Bonadonna | 8/13/2019 3:08:00 PM |

English (United Kingdom), Not Superscript/ Subscript

| Page 35: [9] Formatted | Costanza Bonadonna | 8/13/2019 3:08:00 PM |

English (United Kingdom), Not Superscript/ Subscript

| Page 35: [9] Formatted | Costanza Bonadonna | 8/13/2019 3:08:00 PM |

English (United Kingdom), Not Superscript/ Subscript

| Page 35: [9] Formatted | Costanza Bonadonna | 8/13/2019 3:08:00 PM |

English (United Kingdom), Not Superscript/ Subscript

| Page 35: [10] Formatted | Costanza Bonadonna | 8/13/2019 3:08:00 PM |

English (United Kingdom), Not Superscript/ Subscript

| Page 35: [10] Formatted | Costanza Bonadonna | 8/13/2019 3:08:00 PM |

English (United Kingdom), Not Superscript/ Subscript

| Page 35: [10] Formatted | Costanza Bonadonna | 8/13/2019 3:08:00 PM |

English (United Kingdom), Not Superscript/ Subscript

| Page 35: [10] Formatted | Costanza Bonadonna | 8/13/2019 3:08:00 PM |

English (United Kingdom), Not Superscript/ Subscript

| Page 35: [11] Formatted | Costanza Bonadonna | 8/13/2019 3:08:00 PM |

French (Switzerland)

| Page 35: [11] Formatted | Costanza Bonadonna | 8/13/2019 3:08:00 PM |

French (Switzerland)

| Page 35: [11] Formatted | Costanza Bonadonna | 8/13/2019 3:08:00 PM |

French (Switzerland)

| Page 35: [11] Formatted | Costanza Bonadonna | 8/13/2019 3:08:00 PM |

French (Switzerland)

| **Page 35: [11] Formatted** | **Costanza Bonadonna** | **8/13/2019 3:08:00 PM** |

French (Switzerland)

| **Page 35: [11] Formatted** | **Costanza Bonadonna** | **8/13/2019 3:08:00 PM** |

French (Switzerland)

| **Page 35: [11] Formatted** | **Costanza Bonadonna** | **8/13/2019 3:08:00 PM** |

French (Switzerland)

| **Page 35: [11] Formatted** | **Costanza Bonadonna** | **8/13/2019 3:08:00 PM** |

French (Switzerland)

| **Page 35: [11] Formatted** | **Costanza Bonadonna** | **8/13/2019 3:08:00 PM** |

French (Switzerland)

| **Page 35: [11] Formatted** | **Costanza Bonadonna** | **8/13/2019 3:08:00 PM** |

French (Switzerland)

| **Page 35: [11] Formatted** | **Costanza Bonadonna** | **8/13/2019 3:08:00 PM** |

French (Switzerland)

| **Page 35: [11] Formatted** | **Costanza Bonadonna** | **8/13/2019 3:08:00 PM** |

French (Switzerland)